

# Periodically, quasi-periodically, and randomly driven conformal field theories (II): Furstenberg's theorem and exceptions to heating phases

Xueda Wen[1,2], Yingfei Gu[3], Ashvin Vishwanath[1] and Ruihua Fan[1]

**1** Department of Physics, Harvard University, Cambridge, MA 02138, USA
**2** Department of Physics, University of Colorado, Boulder, CO 80309, USA
**3** California Institute of Technology, Pasadena, CA 91125, USA

## Abstract

In this sequel (to [Phys. Rev. Res. 3, 023044(2021)], arXiv:2006.10072), we study randomly driven $(1+1)$ dimensional conformal field theories (CFTs), a family of quantum many-body systems with soluble non-equilibrium quantum dynamics. The sequence of driving Hamiltonians is drawn from an independent and identically distributed random ensemble. At each driving step, the deformed Hamiltonian only involves the energy-momentum density spatially modulated at a single wavelength and therefore induces a Möbius transformation on the complex coordinates. The non-equilibrium dynamics is then determined by the corresponding sequence of Möbius transformations, from which the Lyapunov exponent $\lambda_L$ is defined. We use Furstenberg's theorem to classify the dynamical phases and show that except for a few *exceptional points* that do not satisfy Furstenberg's criteria, the random drivings always lead to a heating phase with the total energy growing exponentially in the number of driving steps $n$ and the subsystem entanglement entropy growing linearly in $n$ with a slope proportional to central charge $c$ and the Lyapunov exponent $\lambda_L$. On the contrary, the subsystem entanglement entropy at an exceptional point could grow as $\sqrt{n}$ while the total energy remains to grow exponentially. In addition, we show that the distributions of the operator evolution and the energy density peaks are also useful characterizations to distinguish the heating phase from the exceptional points: the heating phase has both distributions to be continuous, while the exceptional points could support finite convex combinations of Dirac measures depending on their specific type. In the end, we compare the field theory results with the lattice model calculations for both the entanglement and energy evolution and find remarkably good agreement.

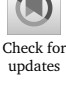

# 1 Introduction

Our understanding of quantum phases of matter has been deeply enriched thanks to the recent studies on the time-dependent driven many-body systems. Novel phases that have no equilibrium analog have been proposed and partly realized experimentally, such as Floquet

topological phases [1–14] and time crystals [15–22]. Non-equilibrium phenomena, including localization-thermalization transitions, prethermalization, dynamical localization, dynamical Casimir effect, are analyzed using models with periodic drivings [23–36].

We are interested in the low energy physics of a critical quantum system that can be described by conformal field theory (CFT) at $(1 + 1)$-dimension [37, 38], where the conformal invariance is particularly helpful in tracing the non-equilibrium dynamics. In Part 1 of this series [39], a general framework has been established for drivings with a single wavelength modulation on the CFT Hamilton, under which the Heisenberg evolution of local operators, as well as the energy and entanglement evolution, are captured by a sequence of Möbius transformation. The Part 1 has focused on periodic and quasi-periodic drivings and found rich non-equilibrium dynamical phase diagrams. However, in experiments, it is inevitable to have the noise during the driving, and therefore it is desirable to understand the fate of a driven quantum many-body system with randomness, which is the main theme of this paper.

Let us recall the general setup discussed in the Part 1, the driving Hamiltonian has the following form

$$H = \int_0^L \frac{dx}{2\pi} \left( f(x)T(x) + g(x)\overline{T}(x) \right),$$ (1)

where $f(x)$ and $g(x)$ are two independent smooth real functions, dubbed deformation functions, and $L$ is the length of the system. Here $T(x)$ and $\overline{T}(x)$ are the chiral and anti-chiral energy-momentum density, namely, $T + \overline{T}$ is the energy density and $T - \overline{T}$ the momentum density. The ordinary homogeneous CFT Hamiltonian, denoted as $H_0$, corresponds to $f(x) = g(x) = 1$. For general deformation function $(f, g)$, the operator evolution under the deformed Hamiltonian (1) can be characterized by a conformal transformation [39–44]:

$$\boxed{\text{Operator evolution}} \Longleftrightarrow \boxed{\text{Conformal maps}}.$$ (2)

Furthermore, when the modulation only involves a single wavelength (i.e. $SL_2$ deformation such as the sine-square deformation), the conformal transformation reduces to a Möbius transformation [40–42, 45–49]. In summary, for the Hamiltonian given in the form of (1), we have

$$\text{Operator evolution under general deformations} \Longleftrightarrow \text{Circle maps},$$
$$\text{Operator evolution under } SL_2 \text{ deformations} \Longleftrightarrow \text{Möbius maps}.$$ (3)

One can also consider non-unitary time-dependent driving, such as imaginary time evolution or using non-Hermitian driving Hamiltonians, to generate operator evolution that is described by a more general conformal map.

With the driving Hamiltonians specified, let us introduce the driving protocol. For simplicity, the initial state $|\Psi_0\rangle$ is chosen to be the ground state of the homogeneous Hamiltonian $H_0$. In the $j$-th step, we drive the system with a sequence of deformed Hamiltonian $\{H_j\}_{j=1\ldots n}$ with certain $(f_j, g_j)$ for a time period $T_j$. The resulting state after $n$ steps is

$$|\Psi_n\rangle = U_n \cdots U_2 \cdot U_1 |\Psi_0\rangle, \quad \text{with} \quad U_j = e^{-iH_j T_j}.$$ (4)

For the periodic and quasi-periodic drivings discussed in the Part 1 [39], the sequences of unitaries $\{U_j\}_{j=1,\ldots,n}$ are deterministic. They share some common features in the heating phases: (1) The entanglement entropy grows linearly in time, and the total energy grows exponentially in time; (2) There are emergent spatial (stable and unstable) fixed points for the operator evolution, which results in the quantum entanglement pumps as well as the formation of energy-momentum peaks in space. Although the studies in this series are based on $SL_2$ deformed Hamiltonian, both features persist in the general deformation [43]. In this part 2, we

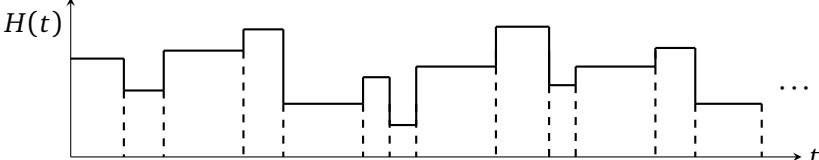

Figure 1: Schematic illustration of a random driving with randomly chosen driving Hamiltonians and random time durations. Here we consider the one parameter family of driving Hamiltonians, and the $y$ axis corresponds to the values of this parameter.

are interested in a *random* sequence. [1] More concretely, each $U_j$ is drawn independently from the ensemble $\{(u_k, p_k)\}_{k=1,\cdots,m}$, where $u_k = e^{-iH_k T_k}$ is the unitary matrix and $p_k$ is the corresponding probability, with $\sum_k p_k = 1$. A typical setup of randomly driven CFTs is schematically illustrated in Fig. 1, where both the time duration and the driving Hamiltonians can be chosen in a random way. Our goal in this paper is to determine the dynamical phases based on the protocols of the random driving.

## 1.1 Random drivings and Furstenberg's theorem

In this work and [39], we consider the driving Hamiltonians with modulations of single wavelength, i.e. we choose the following deformation function $f(x)$ in (1) of the form

$$f(x) = \sigma^0 + \sigma^+ \cos\frac{2\pi q x}{L} + \sigma^- \sin\frac{2\pi q x}{L}, \quad q \in \mathbb{Z}^+, \tag{5}$$

with $\sigma^0, \sigma^+, \sigma^- \in \mathbb{R}$, and similarly for the anti-chiral deformation function $g(x)$. With this deformation, one can find that the driving Hamiltonian only contains three Virasoro generators $\{L_0, L_{\pm q}\}$, which generate the finite-dimensional $SL_2$ algebra, thus the name of $SL_2$ deformation. Then for each driving step, the operator evolution of the primary field $\mathcal{O}$ on the $z$-Riemann surface is determined by

$$U_n^\dagger \mathcal{O}(z, \bar{z}) U_n = \left(\frac{\partial z'}{\partial z}\right)^h \left(\frac{\partial \bar{z}'}{\partial \bar{z}}\right)^{\bar{h}} \mathcal{O}(z', \bar{z}'), \tag{6}$$

where $U_n = e^{-iH_n T_n}$, and $h$ ($\bar{h}$) are the conformal dimensions of operator $\mathcal{O}$. The coordinate $(z, \bar{z})$ arise from a conformal map $z = e^{\frac{2\pi q}{L} w}$ that maps the $w = \tau + ix$-cylinder to a $q$-sheet $z$-Riemann surface, as shown in Fig. 2.

For the $SL_2$ deformation in (5), the operator evolution in (6) has a very simple form of Möbius transformation, with

$$z' = \begin{pmatrix} \alpha & \beta \\ \beta^* & \alpha^* \end{pmatrix} \cdot z = \frac{\alpha z + \beta}{\beta^* z + \alpha^*} =: M(z), \tag{7}$$

where $\alpha, \beta \in \mathbb{C}$, $|\alpha|^2 - |\beta|^2 = 1$. That is, the matrix $M$ as defined above is a $SU(1, 1)$ matrix, which is isomorphic to $SL(2, \mathbb{R})$. The parameters $\alpha$ and $\beta$ in (7) are functions of *both* the deformed function $f_j(x)$ in the driving Hamiltonian $H_j$ and the driving time $T_j$. Therefore, for the random sequence of unitary operators $\{U_j\}$ in (4), we have a random sequence of $SU(1, 1)$

---

[1]Some initial numerical studies on the effects of randomness as a perturbation were done in Ref. [42], where it was found that the randomness can destroy the non-heating phase and result in a heating phase with linear growth of entanglement entropy and exponential growth of total energy. In this current work, we will give a more systematic and rigorous study of the randomly driven CFT.

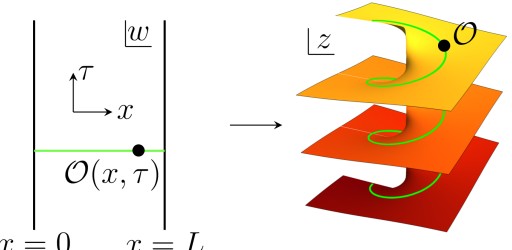

Figure 2: Conformal map $z = e^{\frac{2\pi q w}{L}}$ from the $w$-cylinder/strip to the $q$-sheet $z$-Riemann surface. For the coordinate $w = \tau + ix$, $x = L$ and $x = 0$ are either identified or imposed with conformal boundary conditions for a cylinder or strip, respectively.

matrices $\{M_j\}$. The time evolution of operators in the randomly driven CFT is determined by the random product of SU(1, 1) matrices

$$\Pi_n = M_1 \cdot M_2 \cdots M_{n-1} \cdot M_n \,. \tag{8}$$

Then the phase diagram is determined by the behavior of norm growth in the random products of matrices, which is characterized by the so-called Lyapunov exponent defined as follows

$$\lambda_L := \lim_{n \to \infty} \frac{1}{n} \mathbb{E}\left(\log ||M_n \cdots M_1||\right), \tag{9}$$

where $\mathbb{E}(\cdot)$ represents averaging over the ensamble, [2] and $|| \cdot ||$ is a matrix norm.[3] Now we have reduced a physical problem of diagnosing dynamical phases to a mathematical problem which is the main theme of Furstenberg's theorem [50][4].

**Theorem 1.1** (Furstenberg's theorem). *Let $\{M_j, j \geqslant 1\}$ be independent and identically distributed (i.i.d.) random variables with a probability measure $\mu$, taking values in $\mathrm{SL}(n, \mathbb{R})$. Let $G_\mu$ be the smallest closed subgroup of $\mathrm{SL}(n, \mathbb{R})$ containing the support of the distribution of $M_j$, and assume that $\mathbb{E}(\log ||M_j||) < \infty$. In addition, assume that $G_\mu$ is not compact, and there exists no $G_\mu$-invariant finite set of unit vectors in $\mathbb{R}^n$. Then there exists a positive constant $\lambda_L$ (i.e., Lyapunov exponent) such that with probability one*

$$\lambda_L = \lim_{n \to \infty} \frac{1}{n} \log ||M_n \cdots M_1|| > 0 \,. \tag{10}$$

In our randomly driven CFTs, since the driving time within each driving step is finite, the condition $\mathbb{E}(\log ||M_j||) < \infty$ is always satisfied for each individual $M_j \in \mathrm{SU}(1, 1) \cong \mathrm{SL}(2, \mathbb{R})$. Then to ensure a positive Lyapunov exponent from Furstenberg's theorem, $G_\mu \in \mathrm{SL}(n, \mathbb{R})$ should satisfy the two conditions stated in the theorem, which we will call *Furstenberg's criteria* hereafter:

1. Non-compactness.

   This condition is natural: a positive Lyapunov exponent implies an unbounded growth of the matrix norm. Therefore, $G_\mu$ has to be non-compact.

---

[2]We will sometimes use $\overline{\cdots}$ and $\mathbb{E}(\cdots)$ interchangeably to denote the ensemble average.

[3]The specific choice of norm $||\cdots||$ is not essential for our purpose. We will choose Frobenius norm in this paper, i.e., $||M||_F := \left(\sum_{i,j} |M_{ij}|^2\right)^{1/2}$.

[4]For reviews, see also Ref. [51, 52].

2. No $G_\mu$-invariant finite set of unit vectors in $\mathbb{R}^2$.

Translating to the CFT context, our transformation matrix $M_j$ belongs to SU(1, 1) which is isomorphic to the SL(2, $\mathbb{R}$) that fits into the Furstenberg's theorem. The isomorphism is established by Cayley transformation (see appendix A for more details), and a unit vector in $\mathbb{R}^2$ corresponds to a complex number of modulus 1. Therefore, the second criterion is equivalent to saying that there is no *finite* subset $\mathcal{F} \in \partial\mathbb{D}$ such that $M(\mathcal{F}) = \mathcal{F}$ for all $M \in G_\mu$. Here $\mathbb{D} = \{z \in \mathbb{C}, |z| \leqslant 1\}$ is the unit disk, and $\partial\mathbb{D} = \{z \in \mathbb{C}, |z| = 1\}$ is the boundary of the unit disk. The SU(1, 1) matrix $M$ acts on $\mathcal{F} \subset \partial\mathbb{D}$ as a Möbius transformation, i.e., for $z \in \mathcal{F}$, $M(z)$ is defined in Eq.(7).

This condition is also known as *strong irreducibility*.[5]

Here are several remarks regarding Furstenberg's theorem and its application to randomly driven CFTs:

1. Furstenberg's theorem gives a sufficient but not necessary condition. When the criteria in Furstenberg's theorem are not satisfied, it does not guarantee $\lambda_L = 0$ and one needs to check the Lyapunov exponent $\lambda_L$ explicitly.

2. The Lyapunov exponent defined in (9) is an ensemble average over random products of matrices. Furstenberg's theorem gives a stronger result in the sense that (10) holds for each random sequence with probability 1.

It turns out that most choices of random drivings satisfy Furstenberg's criteria and ensure $\lambda_L > 0$. The exceptional random drivings that violate Furstenberg's criteria only have zero measure in the parameter space, and the behaviors of $\lambda_L$ have to be checked explicitly case by case. Depending on whether Furstenberg's criteria are satisfied, we categorize the possible phases in a randomly driven CFT as follows

$$\begin{cases} \text{Heating phase :} & \text{Furstenberg's criteria are satisfied}, \\ \text{Exceptional point :} & \text{Furstenberg's criteria are violated}. \end{cases} \tag{11}$$

We emphasize that certain types of exceptional points also have a positive Lyapunov exponent. They distinguish themselves from the heating phase by other physical quantities such as the energy-momentum density and the spatial distribution of Heisenberg operators. In other words, the Lyapunov exponent does *not* provide a complete characterization of the heating phase, and it is necessary to examine other observables as we will show in the following sections.

## 1.2 Summary of main results

In this work, we classify and characterize random drivings drawn from independent and identically distributed random ensembles of CFTs with $SL_2$ deformations. In general, there are heating phases and different types of exceptional points, with the features of the time evolution of various physical observables summarized in Table. 1. The results are briefed as follows

1. *Phase diagrams:*

   The heating phases are where Furstenberg's criteria are satisfied, and there are three types of exceptional points when the criteria are failed. All types of exceptional points are

---

[5]It is noted that this condition is stronger than irreducibility. More concretely, given a subset $S$ of SL($d$, $\mathbb{R}$), we say that $S$ is irreducible if there does not exist a proper linear subspace $V$ of $\mathbb{R}^d$ such that $M(V) = V$ for any $M \in S$. $S$ is strongly irreducible if there does not exist a finite union of proper linear subspace of $\mathbb{R}^d$, $V_1$, $V_2, \cdots, V_k$ such that $M(V_1 \cup V_2 \cdots \cup V_k) = V_1 \cup V_2 \cdots \cup V_k$ for any $M \in S$.

Table 1: Features in the heating phase and at different types of exceptional points. The growth rates $\lambda_S$ and $\lambda_E$ are for entanglement entropy (linearly) and energy (exponentially) as defined in (13). $\nu_O$ and $\nu_E$ denote the distributions of the operator evolution and the energy-momentum density peaks. $s$ and $u$ in sub-type-2 of type III exceptional points represent the strength of the stable and unstable fixed points (that coincide in real space) respectively. $\delta$ represents a finite convex combination of Dirac measures (i.e., $\sum_j p_j \delta(x - x_j)$ with $p_j > 0$ and $\sum_j p_j = 1$). For all the three cases with $\lambda_S = 0$, the entanglement entropy grows in time as $\sqrt{n}$.

| | $\lambda_S$ | $\lambda_E$ | $\nu_O$ | $\nu_E$ |
|---|---|---|---|---|
| Heating phase | $> 0$ | $> 0$ | continuous | continuous |
| type I | $= 0$ | $> 0$ | $\delta$ | $\delta$ |
| type II | $= 0$ | $> 0$ | $\delta$ | $\delta$ |
| type III (sub-type 1) | $> 0$ | $> 0$ | $\delta$ | continuous |
| type III (sub-type 3) | $> 0$ | $> 0$ | continuous | $\delta$ |
| type III (sub-type 4) | $> 0$ | $> 0$ | $\delta$ | continuous |
| type III (sub-type 5) | $> 0$ | $> 0$ | continuous | $\delta$ |
| type III (sub-type 2) $s > u$ | $> 0$ | $> 0$ | $\delta$ | continuous |
| type III (sub-type 2) $s < u$ | $> 0$ | $> 0$ | continuous | $\delta$ |
| type III (sub-type 2) $s = u$ | $= 0$ | $> 0$ | continuous | continuous |

distinguished from the heating phase by their behaviors in the time evolution of certain physical observables (See Table. 1). In general, the exceptional points have measure zero in the parameter space, and the phase diagram is dominated by the heating phase. See, e.g., the phase diagram in Fig. 3.

2. *Entanglement and energy growth:*

   In the heating phase where Furstenberg's criteria are satisfied, the ensemble-averaged entanglement entropy and total energy grows in time as[6]

   $$\mathbb{E}(S_A(n)) \sim \frac{\lambda_S \cdot c}{3} n, \quad \mathbb{E}(E(n)) \sim g_E \cdot \frac{c}{l} \cdot e^{2\lambda_E \cdot n}, \quad \text{as } n \to \infty, \tag{12}$$

   where $n$ denotes the number of driving steps, and the subsystem $A$ is chosen as a 'unit cell' $A = [kl + \delta, (k+1)l + \delta]$ where $k \in \mathbb{Z}$, $l = L/q$ is the wavelength of deformation, and $\delta \in [0, l)$ is arbitrarily chosen and will not affect the result. $\lambda_S$ and $\lambda_E$ are two positive real numbers characterising the growth rate of entanglement entropy (linearly) and energy (exponentially). The prefactor $g_E$ for the energy growth is a dimensionless coefficient that depends on the details of the driving. Expression (12) can also be viewed as the definition of the growing rates of the ensemble-averaged entanglement entropy and energy:

   $$\lambda_S := \lim_{n \to \infty} \frac{1}{n} \cdot \frac{3}{c} \cdot \mathbb{E}(S_A(n)), \quad \lambda_E := \lim_{n \to \infty} \frac{1}{n} \cdot \frac{1}{2} \cdot \log \mathbb{E}(E(n)), \tag{13}$$

   which will be used in characterizing different types of exceptional points. In particular, in the heating phase, we can prove that

   $$\lambda_S = \lambda_L, \tag{14}$$

---

[6]When we talk about the evolution time, usually we use the number of driving steps $n$ instead of the real time $t$. It is understood that $t = n \cdot \mathbb{E}(T)$, where $\mathbb{E}(T)$ stands for the ensemble average of driving time in a single driving step.

where $\lambda_L$ is the Lyapunov exponent in (9) for products of transform matrices, which according to Furstenberg's theorem is actually a fixed number that every sequence in the ensemble converges to with probability 1 in the long time limit. That is to say, we can strengthen the first result in (12) to

$$S_A(n) \sim \frac{c}{3} \cdot \lambda_L \cdot n, \quad \text{as } n \to \infty, \tag{15}$$

with probability 1. For the energy growth, it is found that in general $\lambda_E \geqslant \lambda_L$ in a randomly driven CFT. This is different from the features in periodically/quasi-periodically CFTs where $\lambda_E = \lambda_S = \lambda_L$. [39]

Interestingly, at type I, II, and sub-type-2 (when the strengths of the coincident stable and unstable fixed points are the same) of type III exceptional points, the Lyapunov exponent is zero, and so is $\lambda_S$ as defined in (13). In these three cases, we find that

$$\mathbb{E}(S_A(n)) \sim g_S \cdot c \sqrt{n}, \quad \mathbb{E}(E(n)) \sim g_E \cdot \frac{c}{l} \cdot e^{2\lambda_E \cdot n}, \quad \text{as } n \to \infty. \tag{16}$$

That is, although the total energy still grows exponentially in time, the entanglement entropy grows as a square root of time. Here $g_S$ and $g_E$ are dimensionless coefficients that depend on the details of the driving. We provide a physical picture of why the entanglement entropy grows slower than linear: The 'Einstein-Podolsky-Rosen (EPR) pairs' that carry the quantum entanglement are pumped in and out of the subsystem during the random driving, which results in a partial cancellation of the entanglement entropy. This partial cancellation makes the entanglement entropy grow slower than linear. In particular, by choosing the entanglement cuts appropriately, there could be a complete cancellation of entanglement entropy growth, and thus the entanglement entropy may oscillate in time without any growing.

For other sub-types in type III exceptional points, the time evolution of entanglement entropy and total energy are still described by (12). In contrast to the heating phase, we have $\lambda_E = \lambda_S = \lambda_L$ for sub-type 1, 3, 4, and 5 exceptional points.

To verify the field theory calculation, we also provide lattice simulation using a free-fermion lattice model at the critical point and find remarkable agreement in the entanglement and energy evolution.

3. *Distributions of operator evolution and energy-momentum density peaks:*

The distributions of operator evolution and energy-momentum density peaks provide two finer characterizations of the spatial features of a randomly driven CFT. More explicitly, in the context of this paper, the operator evolution is equivalent to the underlying conformal map. In randomly driven CFTs, these conformal maps develop fixed points that are stable or unstable. The stable points capture the locations that an operator at a random initial position will be sent to in the long time limit. We use $\nu_O$ to denote the distribution of stable points and refer as the "distribution of operator evolution". On the other hand, the unstable fixed points are where the energy-momentum density peaks will be. We use $\nu_E$ to denote the distribution of unstable fixed points and refer as the "distribution of energy-momentum density peaks".

In the heating phase, both $\nu_O$ and $\nu_E$ are *continuous*. However, there is a subtle and important difference: the stable fixed points for a given random sequence converge in the long time limit. Therefore the distribution $\nu_O$ is solely due to the ensemble average. However, for the unstable fixed points, the locations fluctuate within a random sequence. Therefore the distribution $\nu_E$ is a consequence of both time and ensemble average. Nevertheless, the two distributions, $\nu_O$ and $\nu_E$ are closely related in the heating phase.

At the exceptional points, the distributions of operator evolution and energy-momentum density peaks exhibit different features. At type I and type II exceptional points, we find that within each wavelength of deformation both $\nu_O$ and $\nu_E$ are a *finite* convex combination of Dirac measures in the long time limit. More concretely, $\nu_O = \nu_E = \frac{1}{2}(\delta(x - x_0) + \delta(x - x_1))$ for the chiral (or anti-chiral) components, where $x_0$ and $x_1$ denote two emergent fixed points (within each wavelength of deformation) in the operator evolution. Here is a schematic illustration to show the difference for $\nu_O$ at type I/II exceptional points and in the heating phase

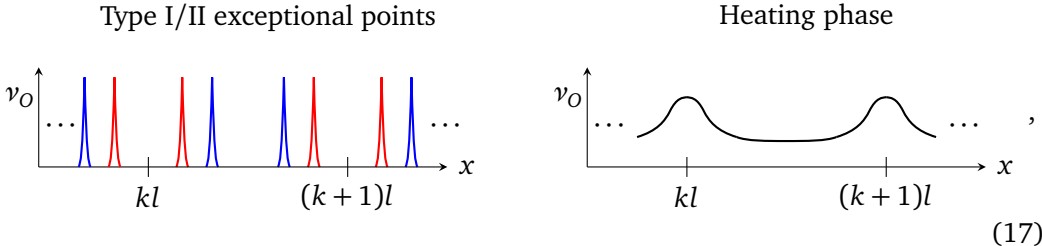

$$(17)$$

where the chiral and anti-chiral components of $\nu_O$ are colored in red and blue respectively. The distribution $\nu_E$ is similar. We also remark that the distribution of operator evolution and energy-momentum peaks at type I/II exceptional points may look similar to those in the heating phase of periodically/quasi-periodically driven CFTs. However, here is a subtle difference: in periodically/quasi-periodically CFTs, there is a single stable fixed point for the (chiral) operator evolution within each wavelength, i.e., $\nu_O = \delta(x - x_\bullet)$. At type I/II exceptional points, the stable and unstable fixed points will switch with each other during the random driving, which yields $\nu_O = \nu_E = \frac{1}{2}(\delta(x - x_0) + \delta(x - x_1))$.

For type III exceptional points, there are five different sub-types (see Table 1). Four sub-types of the five (sub-type 1,3,4,5) have both $\lambda_S > 0$ and $\lambda_E > 0$, i.e., the entanglement entropy grows linearly in time and the energy grows exponentially in time. In these four sub-types, either $\nu_O$ or $\nu_E$ has a Dirac measure $\delta(x - x_*)$ within each wavelength of deformation, where $x_*$ corresponds to the common stable (or unstable) fixed point in the random driving. For the residual sub-type (sub-type 2), its feature depends on the relative strengths of the stable and unstable fixed points which coincide with each other in space. Interestingly, when the relative strengths of the two coincident fixed points are the same, the entanglement entropy grows in time as $\sqrt{n}$, and both $\nu_O$ and $\nu_E$ are continuous. When the relative strengths are different, either $\nu_O$ or $\nu_E$ is continuous and the other is a Dirac measure.

Based on the above features on entanglement/energy evolution and the distributions of operator evolution $\nu_O$ and peaks of energy-momentum density $\nu_E$, we can distinguish the heating phase from all types of exceptional points, as seen in Table 1.

There are two additional remarks:

1. In the heating phase, as we approach the exceptional point and the trivial point (where there is effectively no driving) respectively, we observe different scaling behaviors of the Lyapunov exponents. This difference can be intuitively visualized by studying the group walking of the random products of SU(1, 1) matrices. It is observed that the group walking near the exceptional point and near the trivial point exhibit qualitatively different features. This interesting relation deserves a future study.

2. There is a one-to-one correspondence between the physical properties in randomly driven CFTs and related mathematical theorems on random matrix products. For example, the distribution of operator evolution $\nu_O$ in the heating phase corresponds to the $\mu$-invariant measure in Theorem 2.5. The distribution of operator evolution $\nu_O$ at type I/II exceptional points correspond to the *common* invariant measure in Theorem A.6.

Here is an outline for the rest of this paper: In Section 2, we study in detail various properties of a randomly driven CFT in both the heating phase and at different types of exceptional points based on the field theory approach. The properties we examine include the time evolution of entanglement entropy and total energy, the distribution of operator evolution, and the distribution of energy-momentum density peaks. In Section 3, we compare the lattice simulations and the CFT calculations on both the entanglement and energy evolution. Then we conclude and discuss some future problems in Section 4, including the possible generalization of randomly driven CFTs, and the accidental exceptional points as recently studied in mathematical literature [53]. There are also two appendices. In Appendix A, we provide some details on the basics of Furstenberg's theorem as well as the properties of exceptional points. We also present some further details on the entanglement/energy evolution as well as group walking in a randomly driven CFT in Appendix B.

# 2 Randomly driven CFTs

## 2.1 Preliminaries

In this subsection, we introduce the general formulas for the time evolution of entanglement entropy and energy in $\mathrm{SL}_2$ deformed CFTs, which will be used to characterize different dynamical phases in a randomly driven CFT. Some related details can also be found in our prior work [39].

Let us first classify the types of driving Hamiltonians and their distinct effects on the operator evolution. For the deformations in (1) and (5), one can find that depending on the sign of the quadratic Casimir $c^{(2)} := -(\sigma^0)^2 + (\sigma^+)^2 + (\sigma^-)^2$, there are in total three types of $\mathrm{SL}_2$ deformed driving Hamiltonians:

$$
\begin{cases}
c^{(2)} < 0 : & \text{elliptic type}, \\
c^{(2)} = 0 : & \text{parabolic type}, \\
c^{(2)} > 0 : & \text{hyperbolic type}.
\end{cases}
\tag{18}
$$

Different types of driving Hamiltonians will give different types of Möbius transformations in the operator evolution. More explicitly, by evolving the system for time $T$, one can obtain different types of SU(1,1) matrices $M$ in (7), with the following matrix elements: [39,47]

1. Elliptic ($c^{(2)} < 0$): $|\mathrm{Tr}(M)| < 2$

$$
\alpha = \cos\left(\frac{\pi \mathcal{C} T}{l}\right) + i\frac{\sigma^0}{\mathcal{C}} \sin\left(\frac{\pi \mathcal{C} T}{l}\right), \quad \beta = i\frac{\sigma^+ + i\sigma^-}{\mathcal{C}} \sin\left(\frac{\pi \mathcal{C} T}{l}\right).
\tag{19}
$$

2. Parabolic ($c^{(2)} = 0$): $|\mathrm{Tr}(M)| = 2$

$$
\alpha = 1 + i\frac{\sigma^0 \pi T}{l}, \quad \beta = i\frac{(\sigma^+ + i\sigma^-)\pi T}{l}.
\tag{20}
$$

3. Hyperbolic ($c^{(2)} > 0$): $|\mathrm{Tr}(M)| > 2$

$$
\alpha = \cosh\left(\frac{\pi \mathcal{C} T}{l}\right) + i\frac{\sigma^0}{\mathcal{C}} \sinh\left(\frac{\pi \mathcal{C} T}{l}\right), \quad \beta = i\frac{\sigma^+ + i\sigma^-}{\mathcal{C}} \sinh\left(\frac{\pi \mathcal{C} T}{l}\right).
\tag{21}
$$

Here, $l := L/q$ is the wavelength of the deformation in (5) and $\mathcal{C} := \sqrt{|-(\sigma^0)^2 + (\sigma^+)^2 + (\sigma^-)^2|}$. For a finite driving time $T < \infty$, $|\text{Tr}(M)| < \infty$ and the condition $\mathbb{E}(\log\|M\|) < \infty$ in Furstenberg's theorem is always satisfied. As reviewed in appendix A, on the unit circle $\partial\mathbb{D} := \{z \in \mathbb{C}, |z| = 1\}$, the elliptic, parabolic, and hyperbolic matrix has 0, 1, and 2 fixed points, respectively. Here the fixed points of $M \in \text{SU}(1,1)$ on the unit circle $\partial\mathbb{D}$ are defined by $M(z) = z$ for $z \in \partial\mathbb{D}$. Note the one-to-one correspondence between the types of driving Hamiltonians and types of SU(1,1) matrices (Möbius transformations):

$$
\boxed{\begin{array}{l} \text{Driving with hyperbolic/parabolic/elliptic Hamiltonians for a finite time} \\ \Longleftrightarrow \quad \text{SU}(1,1) \text{ matrices of hyperbolic/parabolic/elliptic types.} \end{array}} \tag{22}
$$

As introduced in Section 1.1, the operator evolution after $n$ steps of random drivings can be represented by:

$$
U_1^\dagger \cdot U_2^\dagger \cdots U_n^\dagger \, \mathcal{O}(z,\overline{z}) \, U_n \cdots U_2 \cdot U_1 = \left(\frac{\partial z_n}{\partial z}\right)^h \left(\frac{\partial \overline{z}_n}{\partial \overline{z}}\right)^{\overline{h}} \mathcal{O}(z_n,\overline{z}_n), \tag{23}
$$

where each step of driving is associated to a matrix $M_j \in \text{SU}(1,1)$ (See (7)). Let us denote the random products of SU(1,1) matrices as

$$
\Pi_n = M_1 \cdot M_2 \cdots M_{n-1} \cdot M_n = \begin{pmatrix} \alpha_n & \beta_n \\ \beta_n^* & \alpha_n^* \end{pmatrix}, \tag{24}
$$

where $\alpha, \beta \in \mathbb{C}$ and $|\alpha_n|^2 - |\beta_n|^2 = 1$. Then in the operator evolution in (23), $z_n$ is related to $z$ through the Möbius transformation

$$
z_n = \Pi_n(z).
$$

Once we know the operator evolution, one can obtain the time evolution of correlation functions and deduce the entanglement entropy. The entanglement entropy $S_A(n)$ of a subsystem with arbitrary length has a complicated expression. However, when we choose the subsystem $A$ to have length $l$, where $l = L/q$ is the wavelength of the deformation in (5), then $S_A(n)$ has a simple form. For example, for $A = [(k-1/2)l, (k+1/2)l]$, where $k \in \mathbb{Z}$, the time-dependent entanglement entropy is (See Appendix B.1)

$$
S_A(n) - S_A(0) = \frac{c}{3}\Big(\log\big|\alpha_n - \beta_n\big| + \log\big|\alpha_n' - \beta_n'\big|\Big), \tag{25}
$$

where $c$ is the central charge, and the first (second) term comes from the contribution of the chiral (anti-chiral) component. See Appendix B.1 for the general choice of $A = [kl + \delta, (k+1)l + \delta]$ with arbitrary shift $\delta \in [0, l]$.

Furthermore, the time evolution of the stress-energy tensor (a quasi-primary operator) is governed by

$$
U_1^\dagger \cdot U_2^\dagger \cdots U_n^\dagger \, T(z) \, U_n \cdots U_2 \cdot U_1 = \left(\frac{\partial z'}{\partial z}\right)^2 T(z') + \frac{c}{12}\text{Sch}(z', z), \tag{26}
$$

where the last term represents the Schwarzian derivative. Then one can obtain the time-dependent energy-momentum density as

$$
\frac{1}{2\pi}\langle T(x,n)\rangle = -\frac{q^2\pi c}{12L^2} + \frac{\pi c}{12L^2}\cdot(q^2-1)\cdot\frac{1}{|\alpha_n e^{\frac{2\pi i x}{l}} + \beta_n|^4}, \quad \text{where } l = L/q. \tag{27}
$$

For the anti-chiral component $\frac{1}{2\pi}\langle \overline{T}(x,n)\rangle$, the expression is the same as above by replacing $\alpha_n(\beta_n) \to \alpha'_n(\beta'_n)$ and $e^{\frac{2\pi i x}{l}} \to e^{-\frac{2\pi i x}{l}}$. If the driven CFT is in a heating phase ($\lambda_L > 0$), or more generally the norm of $\Pi_n$ grows to infinity as $n \to \infty$, then based on (27) one can find that the energy-momentum density is peaked at

$$x_{\text{peak}} = \frac{l}{2\pi i} \log\left(-\frac{\beta_n}{\alpha_n}\right) \mod l, \quad \text{where } |\alpha_n|, |\beta_n| \gg 1. \tag{28}$$

For $x$ away from $x_{\text{peak}}$, the energy-momentum density will be suppressed. In the random driving, we are interested in the ensemble-averaged distribution of these energy-momenum density peaks $x_{\text{peak}}$, which is denoted as $\nu_E$. It is emphasized that $\nu_E$ is *not* the distribution of the energy-momentum density $\langle T(x,n)\rangle$ itself.

The total energy of the system $E(n) = \frac{1}{2\pi}\int_0^L \langle T(x,n)+\overline{T}(x,n)\rangle dx$ has the following explicit expression:

$$E(n) = -\frac{q^2\pi c}{6L} + \frac{\pi c}{12L}(q^2-1)\cdot(|\alpha_n|^2 + |\beta_n|^2 + |\alpha'_n|^2 + |\beta'_n|^2). \tag{29}$$

Note both the entanglement entropy (25) and the total energy (29) are solely determined by the SU(1,1) matrix $\Pi_n$ (8). In the random driving, the quantities we frequently use are the expectation value $\mathbb{E}(S_A(n))$ and $\mathbb{E}(E(n))$, where the average is performed over the ensemble. It is also convenient to consider the limit $L \to \infty$ and consider the energy in one 'unit cell' $E(n) = \frac{1}{2\pi}\int_0^l \langle T(x,n) + \overline{T}(x,n)\rangle dx$, which has the expression

$$E(n) = \frac{\pi c}{12l}(|\alpha_n|^2 + |\beta_n|^2 + |\alpha'_n|^2 + |\beta'_n|^2), \quad L \to \infty. \tag{30}$$

Later in Section 3, we will make a comparison of the CFT calculation and the numerical simulation on a lattice system. In the lattice simulation, to approximate the field theory calculation, it is required that $l = L/q \gg a$ where $a$ is the lattice constant. To have an efficient simulation (which means we consider the total number of lattice sites as small as possible) on the lattice, we choose $q = 1$ and consider open boundary conditions. On the lattice, it is also convenient to only deform the Hamiltonian density, by choosing $f(x) = g(x)$ in (1). In this case, one can find the time evolution of the entanglement entropy as follows: [40]

$$S_A(n) - S_A(0) = \frac{c}{3}\log|\alpha_n - \beta_n|, \quad \text{where } A = [0, L/2]. \tag{31}$$

The expectation value of the chiral energy-momentum density becomes: [42]

$$\frac{1}{2\pi}\langle T(x,n)\rangle = -\frac{\pi c}{12L^2} + \frac{\pi c}{16L^2}\cdot\frac{1}{|\alpha_n e^{\frac{2\pi x}{L}} + \beta_n|^4}. \tag{32}$$

The anti-chiral part $\langle \overline{T}(x,n)\rangle/2\pi$ has the same expression as above by replacing $e^{i\frac{2\pi x}{L}}$ with $e^{-i\frac{2\pi x}{L}}$. Then one can obtain the total energy of the system as

$$E(n) = \frac{\pi c}{8L}(|\alpha_n|^2 + |\beta_n|^2) - \frac{\pi c}{6L}. \tag{33}$$

If there are no drivings, (29) and (33) reduce to $E = -\frac{\pi c}{6L}$ and $-\frac{\pi c}{24L}$, which correspond to the Casimir energies of the CFT of length $L$ with periodic and open boundary conditions respectively.

## 2.2 Phase diagram

We determine the phase diagram of a randomly driven CFT in two steps: (1) Use Furstenberg's criteria and (11) to distinguish the heating phase and the exceptional points; (2) If the driven CFT happens to be at the exceptional point, we then further identify the specific type of each exceptional point.

### 2.2.1 Furstenberg's criteria

Recall that in Section 1.1, the Furstenberg's criteria are briefly summarized as

1. $G_\mu$ is non-compact;

2. $G_\mu$ is strongly irreducible .

where $G_\mu$ is the smallest closed subgroup of $SU(1,1)$ that is generated by the random matrices $\{M_j\}$, and $\mu$ denotes the probability measure of $\{M_j\}$.

Let us begin with the first criterion. Our setup of random driven CFTs always has at least two non-commuting driving Hamiltonians. The corresponding unitary evolution and $SU(1,1)$ matrices in (7) are also non-commuting for generic parameters.[7] Then the first Furstenberg's criterion is always satisfied based on the following theorem [54, 55]:

**Theorem 2.1.** *Let $M_A$, $M_B \in SU(1,1)$ be two non-commuting matrices, then the subgroup $G_\mu$ generated by $\{M_A, M_B\}$ must be non-compact .*

Therefore, we only need to examine the second criterion. In principle, we have to exclude the existence of any invariant *finite* subset $\mathcal{F} \subset \partial\mathbb{D}$. Note that the subgroup generated by the driving Hamiltonians is always non-compact, the second criterion is simplified to the following one [51, 55]:

**Theorem 2.2.** *For a non-compact subgroup $G_\mu \subset SU(1,1)$, Furstenberg's second criterion is equivalent to: there is no finite set $\mathcal{F} \subset \partial\mathbb{D}$ with cardinality 1 or 2 such that $M(\mathcal{F}) = \mathcal{F}$ for all $M \in G_\mu$.*

Namely, for non-compact subgroups, we do not have to examine finite sets with more than two elements.

### 2.2.2 Classification of exceptional points

For simplicity, let us start with two Hamiltonians $H_A$ and $H_B$ that generate two non-commuting transformation matrices $M_A$ and $M_B$ and discuss the more general case with multiple Hamiltonians later. Following Theorem 2.2, we can show that all kinds of random drivings satisfy the second criterion except for three cases [54]

1. Type I: $M_A$ and $M_B$ are reflection matrices. Here $M \in SU(1,1)$ is called a reflection matrix if $M$ is traceless, i.e. $\text{Tr}(M) = 0$.[8] One important property of the reflection matrix $M$ is $M^2 = -\mathbb{I}$, where $\mathbb{I}$ is the identity matrix.

2. Type II: One of $M_A$ and $M_B$ is hyperbolic, and the other is reflection. The reflection matrix permutes the two fixed points of the hyperbolic matrix, in the sense that $M(e^{i\theta_{0(1)}}) = e^{i\theta_{1(0)}}$ where $e^{i\theta_0}$ and $e^{i\theta_1}$ are the two fixed points of the hyperbolic matrix and $M$ is the reflection matrix.

3. Type III: $M_A$ and $M_B$ are both non-elliptic and share one common fixed point .

---

[7] There might be special choice of driving times $T_0$ and $T_1$, where the unitary evolution operators $e^{-iH_0 T_0}$ and $e^{-iH_1 T_1}$ commute while $H_0$ and $H_1$ do not commute. When this happens, it means that either there is no driving effectively or the driving reduces to a single quantum quench. Both are trivial for the purpose of this work and will not be included in our discussion. As an example, see the 'trivial point' in Fig. 3.

[8] A reflection matrix $M$ in this context has eigenvalues $\pm i$ instead of $\pm 1$ which is the usual definition for reflection, therefore the name of 'reflection' may be a slight misnomer. However, since $M$ and $iM$ gives the same Möbius transformations, we will use this terminology.

Here we only list the main properties of these exceptional points that appears in the randomly driven CFTs. We refer readers to Appendix A.2 for detailed and rigorous proof. Mathematical study of this problem can be found in Ref. [54].

A few comments are followed. Apparently, a reflection matrix must be elliptic while the product of two non-commuting reflection matrices must be hyperbolic, thus Typy I still satisfies the first Furstenberg's criterion (non-compactness). The elliptic matrix (19) that appears in the random driven CFTs becomes a reflection matrix if and only if the time period $T$ satisfies the quantization condition $T = (n + \frac{1}{2})\frac{l}{C}$ where $n \in \mathbb{Z}$. In addition, one can show that (see Appendix A.3) each type I exceptional point with subgroup $G_\mu \subset \mathrm{SU}(1,1)$ can be mapped to a type II exceptional point with the same $G_\mu$, and vice versa. Thus, they are equivalent in terms of the subgroup $G_\mu$. But their physical behaviors (such as the detailed features of entanglement entropy evolution) can have some quantitative difference since the driving protocol depends on the time order of the drivings.

By Theorem 2.2, each of these exceptional points has an invariant finite set $\mathcal{F} \in \partial \mathbb{D}$ that has one or two elements. Let us list them below [54]:

1. $\mathcal{F} = \{e^{i\theta_0}, e^{i\theta_1}\} \subset \partial \mathbb{D}$, where $e^{i\theta_0}$ and $e^{i\theta_1}$ are the two fixed points of the hyperbolic matrix $M_C = M_A M_B$, where $M_A$ and $M_B$ are two non-commuting reflection matrices.

2. $\mathcal{F} = \{e^{i\theta_0}, e^{i\theta_1}\} \subset \partial \mathbb{D}$, where $e^{i\theta_0}$ and $e^{i\theta_1}$ are the two fixed points of the hyperbolic matrix in $\{M_A, M_B\}$.

3. $\mathcal{F} = \{e^{i\theta_0}\} \subset \partial \mathbb{D}$, where $e^{i\theta_0}$ is the unique common fixed point of the non-elliptic matrices $M_A$ and $M_B$.

Later in Section 2.4, we will discuss the physical meaning of $\mathcal{F}$: they determine the distributions of operator evolution and energy-momentum density peaks.

Now we consider the random driving with $N$ ($N \geqslant 2$) driving Hamiltonians. Any two of the corresponding $N$ matrices ($M_j \in \mathrm{SU}(1,1)$ with $j = 1, \cdots, N$) do not commute with each other. There are still three types of exceptional points where Furstenberg's second criterion is not satisfied:

1. Type I: All of the $N$ driving Hamiltonians are elliptic. The corresponding $\mathrm{SU}(1,1)$ matrices are all reflection matrices. If $N = 2$, there is no constraint on these two reflection matrices. If $N > 2$, these $N$ matrices are constrained as follows. Pick any two non-commuting reflection matrices, say, $M_A$ and $M_B$. Then all the reflection matrices $M_j$ ($j = 1, \cdots, N$) are required to permute the two fixed points of the hyperbolic matrix $M_C = M_A M_B$.

2. Type II: One of the $N$ driving Hamiltonians is hyperbolic and the others are elliptic. All the elliptic matrices are reflection matrices, which are required to permute the two fixed points of the hyperbolic matrix.

3. Type III: All of the $N$ driving Hamiltonians are non-elliptic (i.e., either parabolic or hyperbolic). The corresponding $\mathrm{SU}(1,1)$ matrices share *only one* common fixed point.[9]

Similar to the case of $N = 2$, one can find that for $N > 2$, type I and type II exceptional points are equivalent to each other. That is, for each type I exceptional point where $G_\mu \in \mathrm{SU}(1,1)$ is generated by $\{M_1, \cdots, M_N\}$, one can map it to a type II exceptional point with the same $G_\mu$, and vice versa.

---

[9]It is noted that if two parabolic matrices share a common fixed point, this means these two matrices commute with each other. Here we are interested in the non-commuting case, and therefore we do not include this trivial case.

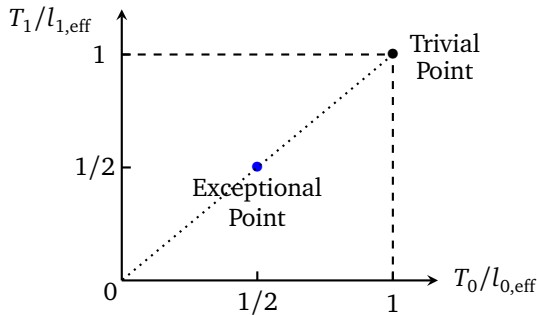

Figure 3: Phase diagram of a randomly driven CFT with two arbitrarily chosen non-commuting *elliptic* Hamiltonians $H_0$ and $H_1$. The phase diagram is periodic in $T_0/l_{0,\text{eff}}$ and $T_1/l_{1,\text{eff}}$. The randomly driven CFT is in the heating phase everywhere in $0 < T_0/l_{0,\text{eff}}, T_1/l_{1,\text{eff}} < 1$ except at the exceptional point where $T_0/l_{0,\text{eff}} = T_1/l_{1,\text{eff}} = 1/2$.

We emphasize again that violation of Furstenberg's second criterion does not imply a vanishing Lyapunov exponent $\lambda_L$. In other words, Furstenberg's criteria are sufficient but not necessary conditions to ensure a positive Lyapunov exponent. We have to check the behavior of $\lambda_L$ at the exceptional points explicitly.

### 2.2.3   Example of phase diagram

In this work, we will mainly consider the following two examples of random driving protocols:

1. *Protocol 1*: There are only two driving Hamiltonians $H_0$ and $H_1$, with the fixed driving time $T_0$ and $T_1$ respectively.[10] During the driving process, we pick $H_0$ and $H_1$ randomly with probabilities $p_0 > 0$ and $p_1 > 0$, where $p_0 + p_1 = 1$.

2. *Protocol 2*: There are more than two driving Hamiltonians. The driving Hamiltonians are randomly chosen from $\{H_j\}$ with a certain probability distribution. Here $H_j$ is characterized by the real parameters $\{\sigma_j^0, \sigma_j^+, \sigma_j^-\}$ in (5). The driving time $T_j$ for each Hamiltonian $H_j$ is fixed, here $j \in \mathcal{J}$ with $\mathcal{J}$ an index set that can be finite or infinite.

The protocol 1 is designed to demonstrate the simplest $N = 2$ case, while in protocol 2, we will include a continuous family of Hamiltonians as shown momentarily.

The randomness is only in the driving Hamiltonians and the the phase diagram is in the dimensionless parameter space spanned by $\{T_j/l\}$. There are certainly other possible protocols one can consider, such as introducing randomness in both the driving Hamiltonians $H_j$ and the driving time $T_j$ (See Fig. 1). We will see that our choice here is already able to capture all the interesting cases including the heating phase and all types of exceptional points introduced in the previous subsection.

In the rest of this subsection, we give explicit examples to illustrate the two protocols introduced above, and describe the corresponding phase diagrams. We will mainly focus on the elliptic driving Hamiltonians and the type I exceptional point in this subsection. Other Hamiltonian types and exceptional points can be found in Section 2.5 and Appendix B.2, where it is found that the type II exceptional points can form a line rather than an isolated point in the parameter space.

---

[10]For simplicity, one of the Hamiltonian will be chosen in the original undeformed form, hence the label $H_0$.

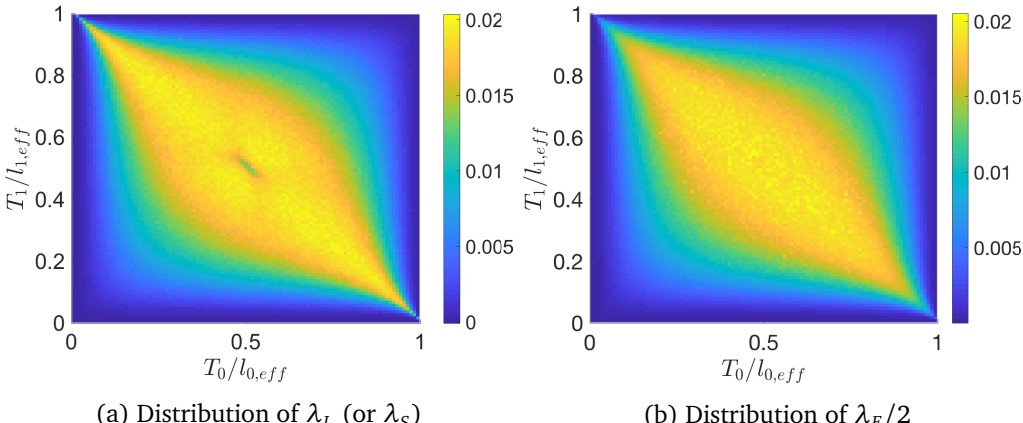

(a) Distribution of $\lambda_L$ (or $\lambda_S$)  (b) Distribution of $\lambda_E/2$

Figure 4: Distribution of Lyapunov exponents $\lambda_L$ (left), or equivalently $\lambda_S$ which characterize the entanglement entropy growth, and $\lambda_E$ (right) which characterize the total energy growth (see (13)), as a function of $T_0/l_{0,\text{eff}}$ and $T_1/l_{1,\text{eff}}$. We choose two random driving Hamiltonians $H_{\theta_0=0}$ and $H_{\theta_1=0.2}$ in (34) with probabilities $1/2$ and $1/2$ respectively. We perform ensemble average over $N_{\text{sample}} = 10^3$ ($N_{\text{sample}} = 2 \times 10^4$) in the calculation of $\lambda_L$ ($\lambda_E$). The type I exceptional point is located at $T_0/l_{0,\text{eff}} = T_1/l_{1,\text{eff}} = 1/2$. See also the schematic plot of phase diagram in Fig. 3.

Let us give some general statements about the phase diagrams without specifying the detailed form of the Hamiltonian. In protocol 1, we consider two non-commuting elliptic Hamiltonians $H_0$ and $H_1$ with driving time $T_0$ and $T_1$, which generates elliptic Möbius transformation (19) with $\mathcal{C}_0$ and $\mathcal{C}_1$. The phase diagram must be periodic in $\pi\mathcal{C}_0 T_0/l$ and $\pi\mathcal{C}_1 T_1/l$. Therefore, we only need to consider the parameter regime $0 < T_0/l_{0,\text{eff}}, T_1/l_{1,\text{eff}} < 1$, where $l_{0(1),\text{eff}} := l/\mathcal{C}_{0(1)}$ is the effective length of the deformed CFT. The type I exceptional point is at $T_0/l_{0,\text{eff}} = T_1/l_{1,\text{eff}} = 1/2$, where the two Möbius transformation matrices become reflection. The heating phase occupies the rest of the phase diagram.[11] See Fig. 3 for a sketch of the structure of the phase diagram. Later in Section 2.4, we will further show that this type I exceptional point has a zero Lyapunov exponent.

The above discussion can be generalized to protocol 2 with $N > 2$ elliptic driving Hamiltonians. When $T_0/l_{0,\text{eff}} = \cdots = T_{N-1}/l_{N-1,\text{eff}} = 1/2$, the corresponding Möbius transformation matrices become reflection. However, it needs to satisfy one more condition to be a type I exceptional point, *i.e.*, all the reflection matrices must permute the two fixed points of the hyperbolic matrix which is obtained by multiplying two arbitrary reflection matrices.

Now, let us specify some concrete Hamiltonians to illustrate the above general picture with some numerical results. The random Hamiltonian will be drawn from the following one-parameter family by choosing random $\theta$:

$$H_\theta = \int_0^L \left(1 - \tanh(2\theta) \cdot \cos\frac{2\pi q x}{L}\right) T_{00}(x)dx, \quad q \in \mathbb{Z}^+, \theta \geqslant 0. \tag{34}$$

$H_{\theta=0}$ is the uniform CFT Hamiltonian and $H_{\theta=\infty}$ is the sine-square deformed Hamiltonian [56–70]. Denoting the driving time with $H_\theta$ as $T_\theta$, then the corresponding Möbius transfor-

---

[11]There is also a trivial point at $T_0/l_{0,\text{eff}} = T_1/l_{1,\text{eff}} = 1$, where the corresponding SU(1,1) matrices are identities (up to a global minus sign) and there is effectively no driving.

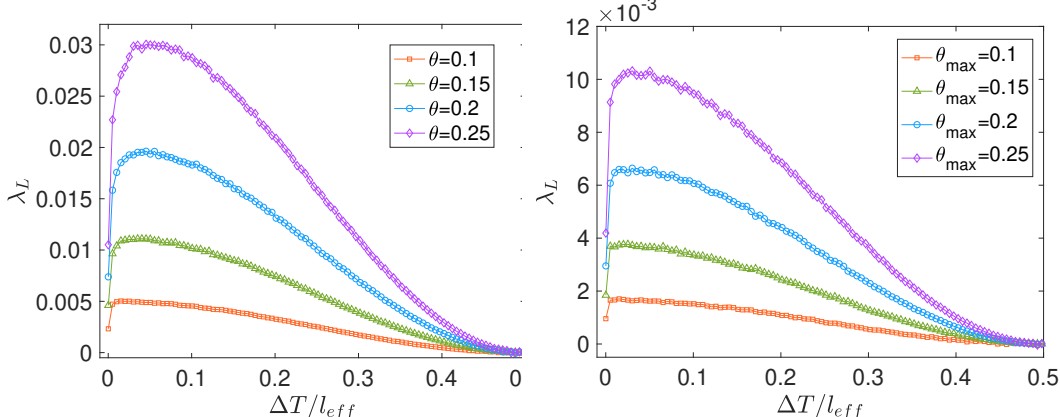

Figure 5: Lyapunov exponents $\lambda_L$ as a function of $\Delta T/l_{\text{eff}} := T_\theta/l_{\theta,\text{eff}} - 1/2$ with $N = 2$ (left) and $N = \infty$ (right) driving Hamiltonians, respectively. For $N = 2$, we choose the two driving Hamiltonians as $H_{\theta=0}$ and $H_{\theta\neq0}$, with $\theta = 0.1$, 0.15, 0.2, and 0.25, respectively. The probabilities are $p_0 = p_1 = 1/2$. For $N = \infty$, we choose the Hamiltonians as $H_\theta$ with $\theta$ uniformly distributed in $[0, \theta_{\text{max}}]$, where $\theta_{\text{max}} = 0.1$, 0.15, 0.2, and 0.25, respectively. For both cases, we take the ensemble average over $N_{\text{sample}} = 5000$.

mation $M(H_\theta, T_\theta)$ has the form in (7) with

$$\alpha = \cos\left(\frac{\pi T_\theta}{l_{\theta,\text{eff}}}\right) + i\cosh(2\theta)\cdot\sin\left(\frac{\pi T_\theta}{l_{\theta,\text{eff}}}\right), \quad \beta = -i\sinh(2\theta)\cdot\sin\left(\frac{\pi T_\theta}{l_{\theta,\text{eff}}}\right),$$

where $l_{\theta,\text{eff}} = l\cosh(2\theta)$ is the effective length of the deformed CFT. One can check that the above SU(1,1) matrix $M(H_\theta, T_\theta)$ is always elliptic, except at $T_\theta = nl_{\theta,\text{eff}}$ ($n \in \mathbb{Z}$) where $M$ becomes an identity matrix (up to a global minus sign).

Now we consider the driving protocol 1 by randomly choosing two driving Hamiltonians $H_0 = H_{\theta=0}$ and $H_1 = H_{\theta\neq0}$ with probabilities 1/2 and 1/2 respectively. A numerical calculation of the distribution of Lyapunov exponents $\lambda_L$ is shown in Fig. 4 (a), where one can observe a dip at the type I exceptional point at $T_0/l_{0,\text{eff}} = T_1/l_{1,\text{eff}} = 1/2$. We also plot the distribution of the energy growth rate $\lambda_E$ (defined in (13)) in Fig. 4 (b), where there is no dip at the type I exceptional point. Thus, energy growth is not able to distinguish the type I exceptional point from the heating phase.

Now let us take a closer look at the distribution of $\lambda_L$, and in particular how the Lyapunov exponent $\lambda_L$ approaches zero near the exceptional point and near the trivial point respectively. In Fig. 5 (left plot), we consider the driving protocol 1 with only two driving Hamiltonains, and study the Lyapunov exponent along the line $T_0/l_{0,\text{eff}} = T_1/l_{1,\text{eff}} =: \frac{1}{2} + \Delta T/l_{\text{eff}}$ in Fig. 4. It is found that the Lyapunov exponent is positive everywhere except at the exceptional point at $\Delta T/l_{\text{eff}} = 0$ and trivial point at $\Delta T/l_{\text{eff}} = 1/2$. The Lyapunov exponents $\lambda_L$ changes continuously from $\lambda_L = 0$ at the exceptional point to $\lambda_L = 0$ at the trivial point.

One can also consider the driving protocol 2 with $N = \infty$ driving Hamiltonians, by taking $\theta$ randomly distributed in $[0, \theta_{\text{max}}]$ in (34). The infinite dimensional parameter space spanned by $\{T_\theta/l_{\theta,\text{eff}}\}$ always has a type I exceptional point at $T_\theta/l_{\theta,\text{eff}} = 1/2$ for all $\theta \in [0, \theta_{\text{max}}]$. As shown in Fig. 5 (right plot), similar to the case of $N = 2$, the Lyapunov exponent is positive everywhere except at the exceptional point at $\Delta T/l_{\text{eff}} := T_\theta/l_{\theta,\text{eff}} - 1/2 = 0$ and the trivial point at $\Delta T/l_{\text{eff}} = 1/2$.

In both plots in Fig. 5, $\lambda_L$ is a continuous function of the parameter $\Delta T/l_{\text{eff}}$. This continuous property of $\lambda_L$ is mathematically proved in Ref. [71]. Also see Theorem A.2 in the

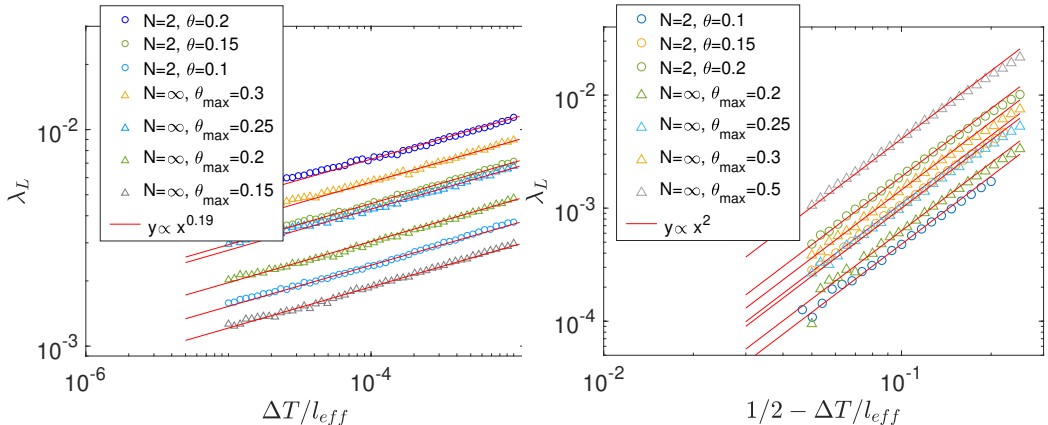

Figure 6: Scaling behavior of $\lambda_L$ near the exceptional point at $\Delta T/l_{\text{eff}} := T_\theta/l_{\theta,\text{eff}} - 1/2 = 0$ (left) and the trivial point at $\Delta T/l_{\text{eff}} = 1/2$ (right) in Fig. 5. The fitting lines (in red) are $y \propto x^{0.19}$ on the left and $y \propto x^2$ on the right, respectively. We study both the driving protocols with $N = 2$ and $N = \infty$ random Hamiltonians.

appendix.

We further study the scaling behavior of $\lambda_L$ near the exceptional point and near the trivial points for both protocols. As seen in Fig. 6, the scaling behavior of $\lambda_L$ near the exceptional point is different from that near the trivial point. More explicitly, we find $\lambda_L \propto (\Delta T/l_{\text{eff}})^{0.19}$ near the exceptional point, and $\lambda_L \propto (1/2 - \Delta T/l_{\text{eff}})^2$ near the trivial point. This difference indicates the fine structures of random product of matrices are different near the exceptional point and near the trivial point. This difference is further revealed by the group walking in the random product of matrices, which essentially tells us how the matrix elements of $\Pi_n = M_1 \cdots M_n$ evolves in time (see Appendix B.3). As seen in Fig. 29 in the appendix, indeed one can observe distinct features of group walking near the exceptional point and near the trivial point. Understanding the relation between the scaling behavior of $\lambda_L$ and the features of group walking is an interesting problem and is left for future study.

As a remark, we also study the scaling behavior of $\lambda_L$ near the type II exceptional points in Appendix B.2. As seen in Fig. 28 in the appendix, It is found that $\lambda_L \propto (\Delta T/l_{\text{eff}})^{0.2}$, where the scaling exponent is close to that near the type I exceptional point. This is somehow as expected, since one can show that type I and type II exceptional points are equivalent as discussed in Appendix A.3.

## 2.3 Heating phase

The heating phase refers to the parameter regime that satisfies the Furstenberg's criteria and has a positive Lyapunov exponent. In this section, we give the details of two main features that have been summarized in Table. 1

1. The entanglement entropy grows linearly and the total energy grows exponentially in time;

2. The distributions of the operator evolution and the averaged energy-momentum density peaks are *continuous* in space in the long time driving limit $n \to \infty$.[12]

---

[12]For heating phase with $\lambda_L > 0$, the long time limit may be understood as $n \gg 1/\lambda_L$.

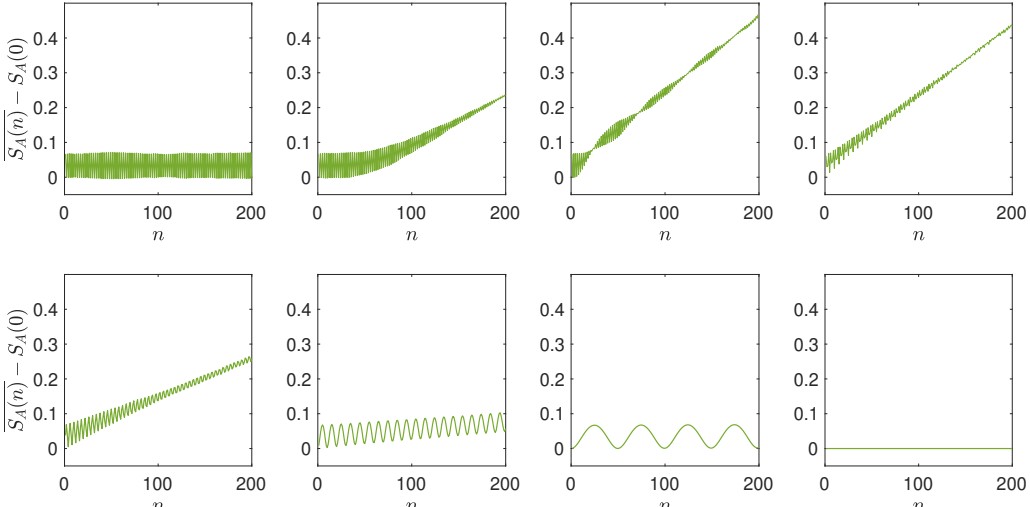

Figure 7: Ensemble-averaged entanglement entropy evolution in a randomly driven CFT, with $A = [kl - l/2, kl + l/2]$ where $k \in \mathbb{Z}$. We consider the driving protocol 2 by choosing $\theta$ randomly distributed in $[0, 0.2]$ in (34). From left to right (and then top to bottom), we choose $\Delta T / l_{\text{eff}} := T_\theta / l_{\theta, \text{eff}} - 1/2 = 0$, 0.0005, 0.01, 0.1, 0.25, 0.4, 0.48, and 0.5, respectively. It is noted that $\Delta T / l_{\text{eff}} = 0$ corresponds to the type I exceptional point and $\Delta T / l_{\text{eff}} = 0.5$ corresponds to the trivial point. In this plot, for comparison we show the time evolution for a fixed time window, which may be too small to see the linear slope for cases with small growth rate, e.g. the plot for $\Delta T / l_{\text{eff}} = 0.48$ is indeed growing when we zoom out the time window. Other parameters are $N_{\text{sample}} = 10^4$, $c = 1$ and $l = 1$.

### 2.3.1 Entanglement entropy growth with $\lambda_S = \lambda_L$

As shown in Fig. 7 for an explicit example, the ensemble-averaged entanglement entropy grows linearly in the long time limit in heating phase. In general, we show that for a subsystem $A$ with length $l = L/q$, (e.g. $A = [kl + \delta, (k+1)l + \delta]$ where $\delta \in [0, l)$ and $k \in \mathbb{Z}$), the entropy formula has a simple form (12)

$$\mathbb{E}(S_A(n)) - S_A(0) = \frac{c}{3} \cdot \lambda_S \cdot n \quad \text{as } n \to \infty, \quad \text{where } \lambda_S > 0. \tag{35}$$

In addition, the growth rate of entanglement $\lambda_S$ is equal to the Lyapunov exponent $\lambda_L$

$$\lambda_S = \lambda_L. \tag{36}$$

To prove the above equality, let us first choose the shift $\delta = -1/2$, i.e., the subsystem $A = [kl - l/2, kl + l/2]$ where $k \in \mathbb{Z}$. Following our notation in (24), let $M_j$ denote the SU(1, 1) Möbius transformation matrix for each driving step and $\Pi_n = M_1 \cdots M_n$ the total one for $n$ steps. The growth rate $\lambda_S$ in (35) can be written as (only the chiral part)

$$\lambda_S = \lim_{n \to \infty} \frac{1}{n} \mathbb{E}(\log |\alpha_n - \beta_n|), \tag{37}$$

where $\alpha_n, \beta_n \in \mathbb{C}$ are the matrix elements of $\Pi_n = M_1 \cdots M_n$ in (24). The proof relies on the following theorem in the random products of SL(2, $\mathbb{R}$) matrices [51].

**Theorem 2.3.** *Let $\{Y_n; n \geqslant 1\}$ be independent and identically distributed (i.i.d.) random matrices in* $\mathrm{SL}(2, \mathbb{R})$. *If Furstenberg's criteria are satisfied, then for any vector* $\vec{x} = (x_1, x_2)^T \neq 0$, *we have*

$$\lambda_L := \lim_{n \to \infty} \frac{1}{n} \log \|Y_n \cdots Y_1\| = \lim_{n \to \infty} \frac{1}{n} \log \|Y_n \cdots Y_1 \vec{x}\|, \tag{38}$$

*with probability 1.*

In order to apply this theorem, we introduce an isomorphism between the $\mathrm{SU}(1, 1)$ matrix $M$ and $\mathrm{SL}(2, \mathbb{R})$ matrix $Y$, $M = QYQ^{-1}$, where $Q = \frac{1}{\sqrt{2}} \begin{pmatrix} 1 & -i \\ 1 & i \end{pmatrix}$ is the Cayley map discussed in Appendix A. Then, we define $M_i = QY_{n+1-i}Q^{-1}$ and have $\Pi_n = Q \cdot Y_n \cdots Y_1 \cdot Q^{-1}$. By denoting the matrix product as $Y_n \cdots Y_1 =: \begin{pmatrix} a_n & b_n \\ c_n & d_n \end{pmatrix}$ and plugging in the expression of $Q$, one can find that $|\alpha_n - \beta_n| = \sqrt{b_n^2 + d_n^2}$, which is also what appears on the right-hand side of (38) if we choose $\vec{x} = (0, 1)^T$. Thus, we have shown

$$\lambda_S = \lim_{n \to \infty} \frac{1}{n} \log |\alpha_n - \beta_n| = \lim_{n \to \infty} \frac{1}{n} \log(b_n^2 + d_n^2)^{1/2} = \lim_{n \to \infty} \frac{1}{n} \log \|Y_n \cdots Y_1 \vec{x}\| = \lambda_L, \tag{39}$$

which proves the result for this special choice of subsystem. By noting that Eq. (38) holds even without averaging, we actually obtain a stronger result

$$S_A(n) - S_A(0) = \frac{c}{3} \cdot \lambda_L \cdot n, \quad n \to \infty. \tag{40}$$

Namely, for a single random sequence, the entanglement entropy almost surely linearly grows and its growth rate $\lambda_S$ is equal to the Lyapunov exponent $\lambda_L$. A sample plot of the entanglement entropy evolution in a single random sequence in the heating phase can be found in Fig. 24 in the appendix.

The above proof can be generalized to the subsystem $A = [kl + \delta, (k+1)l + \delta]$ where $\delta \in [0, l)$ and $k \in \mathbb{Z}$, of which the entanglement entropy is (see (B.4) in Appendix B.1)

$$S_A(n) - S_A(0) = \frac{c}{3} \Big( \log \Big| \alpha_n \cdot e^{\frac{2\pi i \delta}{l}} + \beta_n \Big| \Big). \tag{41}$$

The previous procedures still hold as long as $\vec{x} = (0, 1)^T$ is replaced by $\vec{x} = (-\cos \frac{\pi \delta}{l}, \sin \frac{\pi \delta}{l})^T$. This proves $\lambda_S = \lambda_L$ for arbitrary subsystems of length $l = L/q$.

As a numerical verification of $\lambda_S = \lambda_L$, we calculate the ensemble-averaged time evolution of the entanglement entropy as well as $\log \|\Pi_n\|$ in the heating phase. For both quantities, it is found that they grow linearly in $n$, and the growing rates (which correspond to $\lambda_S$ and $\lambda_L$) are the same. See, e.g., Fig. 25 in the appendix.

Let us conclude this subsection with a remark. There is no necessary relation between the Lyapunov exponents $\lambda_L$ and the growth rate of energy $\lambda_E$ in (16). Because the ensemble-averaged energy is an average over $|\alpha_n|^2 + |\beta_n|^2$ in $\Pi_n$. The Lyapunov exponent, however, is an average over $\log \|\Pi_n\|$. It is emphasized that *the average does not commute with the logarithm*. Therefore, it is possible that the Lyapunov exponent is zero but the total energy grows exponentially in time. For example, as we will see later, at the type I and type II exceptional points, the Lyapunov exponent is zero but the total energy grows exponentially in time as $\mathbb{E}(E(n)) \propto e^{2\lambda_E \cdot n}$ where $\lambda_E > 0$.

### 2.3.2 Energy growth with $\lambda_E \geqslant \lambda_L$

In the heating phase, we find that the ensemble-averaged energy in (30) always grows exponentially in time as

$$\mathbb{E}(E(n)) - E(0) \sim g_E \cdot \frac{c}{l} \cdot e^{2\lambda_E \cdot n}, \quad \text{as } n \to \infty. \tag{42}$$

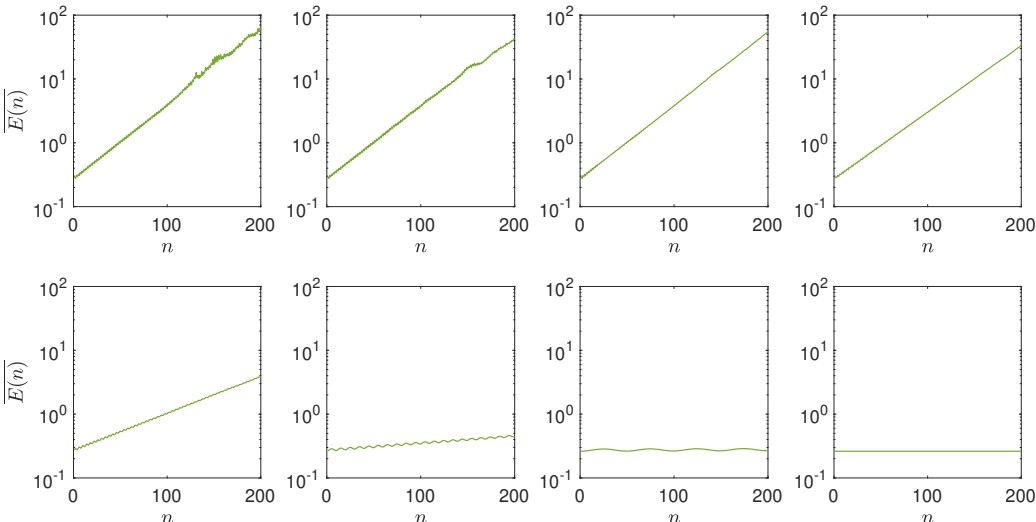

Figure 8: Time evolution of the ensemble-averaged energy in (30) in a randomly driven CFT. The driving protocol and parameters are the same as those in Fig. 7. Here we choose $N_{\text{sample}} = 10^5$.

A sample plot of the energy growth is shown in Fig. 8. To make a comparison for different driving parameters, we plot all the energy growths within a small time window. It is noted that for small $\lambda_E$, one needs to take more driving steps to observe the exponential growth.

In the heating phases of periodically and quasi-periodically driven CFTs, the growth rates of entanglement and energy are both equal to the Lyapunov exponent $\lambda_S = \lambda_E = \lambda_L$. [13] However, this relation does not hold in general in the randomly driven CFT. In general, one can apply Jensen's inequality to prove $\lambda_E \geqslant \lambda_L$.[14] By comparing Fig. 8 and Fig. 7, one can actually find that $\lambda_E > \lambda_S$ (near the type I exceptional point), which can be seen more clearly later in Fig. 11. This may be understood based on the continuity of $\lambda_E$ and $\lambda_S$. First, we have proved $\lambda_E > \lambda_S = 0$ at the type I exceptional points (see Section 2.4). Second, we can see that $\lambda_S$ and $\lambda_E$ are continuous functions of driving parameters near the exceptional point.[15] It is then natural to expect $\lambda_E > \lambda_S$ to hold near the exceptional points.

### 2.3.3 Operator evolution and energy-momentum density peak distribution

In this subsection, we examine the fine structures in the heating phase. In particular, we want to explore the following two aspects:

1. Distributions of operator evolution. The position of an operator, when mapped onto the $q$-sheet Riemann surface, moves on the unit circle $\partial \mathbb{D}$ randomly under the Möbius transformation. We are curious (1) whether its position has a well-defined long-time limit, and (2) what the distribution of the position in the long-time limit $v_O$ is. We call them the distribution of operator evolution.

---

[13]As a reminder, in periodically and quasi-periodically driven CFTs, $\lambda_S$ and $\lambda_E$ are defined in (13) without taking ensemble average, and $\lambda_L$ is defined in (10).

[14]Based on (13) and (29), we have $\lambda_E = \lim_{n\to\infty} \frac{1}{2n} \log \mathbb{E}(||\Pi_n||^2)$, where $\Pi_n$ is the random product of SU(1,1) matrices in (8). Then by using the Jensen's inequality, we have $\log\left(\frac{\sum_j x_j}{n}\right) \geqslant \frac{1}{n}\sum_j \log x_j$, where $x_j$ are positive numbers. It follows that $\lambda_E \geqslant \lim_{n\to\infty} \frac{1}{n}\mathbb{E}(\log||\Pi_n||) = \lambda_L$, where $\lambda_L$ is defined in (9).

[15]That $\lambda_L$ ($\lambda_S$) is a continuous function of driving parameters has been proved mathematically [71]. The continuity of $\lambda_E$ is a numerical observation.

2. Energy-momentum density. We are curious (1) whether the energy-momentum density develops peaks as what it does in the periodic and quasi-periodic driving case, and if so, (2) what the distribution of the energy-momentum density peak $\nu_E$ is.

The two aspects are also closely related. Notice that the Möbius transformation has both stable fixed points and unstable fixed points. The two aspects are exactly probing the (distribution of) stable and unstable fixed point respectively [42]. These questions, especially ones about operator evolution, can be studied with the concept called the *invariant measure on* $\partial\mathbb{D}$ with respect to $G \subset \mathrm{SU}(1,1)$, which is the probability distribution on $\partial\mathbb{D}$ that is invariant under $\mathrm{SU}(1,1)$ transformations in $G$. In the following, we will first introduce the precise definition of this concept and relevant theorem, then apply them to answer these questions raised above. Readers can also directly jump to the physics conclusions and refer to the mathematics theorem when necessary.

Let $\mathcal{M}(\partial\mathbb{D})$ denote the set of probability measures $\nu$ on $\partial\mathbb{D}$. Given $M \in \mathrm{SU}(1,1)$ and $\nu \in \mathcal{M}(\partial\mathbb{D})$, we define $M\nu \in \mathcal{M}(\partial\mathbb{D})$ by

$$\int f(z)\,d(M\nu)(z) = \int f(Mz)\,d\nu(z), \tag{43}$$

where $z \in \partial\mathbb{D}$, and $M$ acts on $z$ by a Möbius transformation in (7). Moreover, denoting $\mu$ as the probability measure of $M \in \mathrm{SU}(1,1)$, we define the convolution $\mu * \nu \in \mathcal{M}(\partial\mathbb{D})$ by

$$\int_{\partial\mathbb{D}} f(z)\,d(\mu * \nu)(z) = \int_{\mathrm{SU}(1,1)} \int_{\partial\mathbb{D}} f(Mz)\,d\mu(M)\,d\nu(z). \tag{44}$$

In particular, if $\mu * \nu = \nu$, then $\nu$ is called $\mu$-invariant measure.[16] The $\mu$-invariant measure in the heating phase of randomly driven CFT has the following property: [51]

**Lemma 2.4.** *Let $\mu$ be the probability distribution of* $\mathrm{SU}(1,1)$*. If Furstenberg's criteria are satisfied, then the $\mu$-invariant measure on $\partial\mathbb{D}$ is unique and continuous.*

Now let us consider the distribution of operator evolution both in a single random sequence and in the ensemble average. Its behavior can be understood based on the following theorem: [51]

**Theorem 2.5.** *Let $\{M_j, j \geqslant 1\}$ be a bounded sequence in $\mathrm{SU}(1,1)$ with a probability measure $\mu$ such that Furstenberg's criteria are satisfied.*

1. *Then there exists $z^\bullet \in \partial\mathbb{D}$ s.t. for any $z \in \mathbb{D}$, $M_1 M_2 \cdots M_n \cdot z$ converges to $z^\bullet$ as $n \to \infty$.*

2. *If $\nu$ is a continuous distribution on $\partial\mathbb{D}$, $M_1 \cdots M_n \nu$ converges weakly to the Dirac measure $\delta(z - z^\bullet)$.*

3. *The distribution of $z^\bullet$ is the unique $\mu$-invariant continuous distribution on $\partial\mathbb{D}$.*

---

[16]As the simplest case, in Appendix A.2 (see Theorem A.5), we have summarized the invariant measure for a *single* matrix $M \in \mathrm{SU}(1,1)$. One can find that only the matrix of elliptic type has a continuous invariant measure. For more than one matrices, one can also consider the *common invariant measure*, which is the intersection of invariant measure for each matrix. As summarized in Theorem A.6, one can find that if these matrices are non-commuting, then the common invariant measure can only be a discrete measure (which is a convex combination of the Dirac measure). By definition, the common invariant measure is automatically a $\mu$-invariant measure. However, the opposite is not true, i.e., a $\mu$-invariant measure is not necessarily a common invariant measure. In fact, in the heating phase, the $\mu$-invariant measure is always not a common invariant measure of the matrices in $G_\mu$.

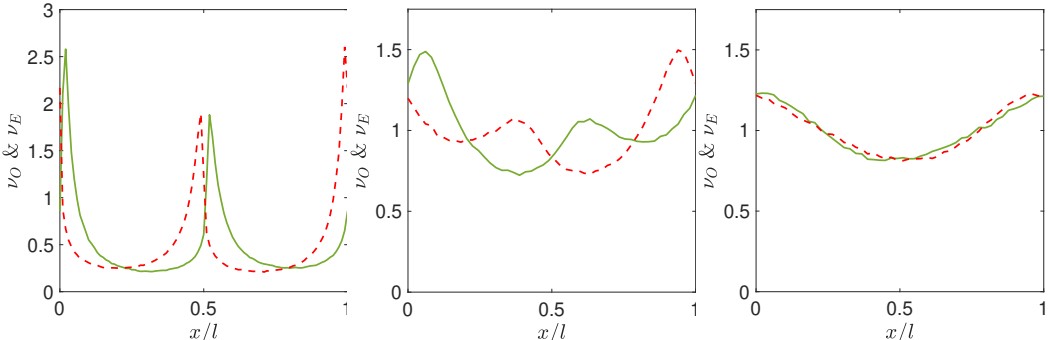

Figure 9: Distribution of the (chiral) operator evolution $\nu_O$ (green solid lines) within one wavelength of deformation. The protocol and driving parameters are the same as those in Fig. 7 and Fig. 8. From left to right, we take $\Delta T/l_{\text{eff}} = 0.0005$, $0.01$, and $0.1$ respectively. The red dashed lines are the distribution of (chiral) energy-momentum density peaks $\nu_E$. We take $N_{\text{sample}} = 10^6$ in the ensemble average.

The property 2 in the above theorem is usually called the contraction property, i.e., a continuous distribution on $\partial \mathbb{D}$, acted by $M_1 \cdots M_n \cdots$, is always contracted to a Dirac measure. However, the location of this Dirac measure is not predictable.

With the above theorems, let us give mathematically rigorous statements of the properties of operator evolution in the heating phase of a randomly driven CFT:

1. *Operator evolution in a single random sequence:* The property 2 in Theorem 2.5 describes exactly the operator evolution in a single random sequence. That is, although the driving is random, the operators starting from arbitrary initial positions will flow to a stable fixed point in the long time driving limit. A sample plot of the operator evolution can be found in Fig. 22 in Appendix A.4. Recall that the operator evolution is considered in the Heisenberg picture. In the Shrödinger picture, it means the degrees of freedom, which carry quantum entanglement, flow away from this fixed point. Therefore, the emergent stable fixed point in the operator evolution actually corresponds to the location of source of generating quantum entanglement [39, 43]. In other words, during the random driving, the location where the quantum entanglement is generated becomes stable in the long time driving limit.

2. *Ensemble-averaged operator evolution:*

   The property 3 in Theorem 2.5 tells us the feature of the ensemble-averaged operator evolution $\nu_O$. That is, the ensemble average of the stable fixed points of operator evolution in the long time driving limit corresponds to the $\mu$-invariant measure as defined near (44). Based on Lemma 2.4, this distribution $\nu_O$ is *continuous*. A sample plot of the ensemble-averaged distribution of operator evolution with different driving parameters in the heating phase can be found in Fig. 9.

   As a remark, one can see some interesting features in the distribution of operator evolution in Fig. 9. In particular, there are some peak structures near $\Delta T/l_{\text{eff}} := T_\theta/l_{\theta,\text{eff}} - 1/2 = 0$. The reason is that, as we will see later in Section 2.4.3, the distribution of operator evolution becomes a convex combination of Dirac measure (which are located at $x/l = 0$ and $1/2$ for the parameters in Fig. 9) at the exceptional point. The peak structure in Fig. 9 inherits the

   feature of $\nu_O$ at the exceptional point.

Now let us consider the energy-momentum density peaks:

1. *Energy density peaks in a single random sequence:*

   First, it is noted that in the heating phase, the total energy of the system grows exponentially in time (see Section 2.3.2). This indicates that the energy density always forms peaks in space in the long time driving limit (This can be understood based on (27) where $\alpha_n$, $\beta_n \to \infty$). The locations of these peaks are determined by (28) within each wavelength of deformation. Second, different from the operator evolution, it is found that the locations of these energy density peaks keep changing during the random driving. A sample plot of the time evolution of locations of energy density peaks can be found in Appendix A.4. Namely, although the random product of SU(1, 1) matrices has a unique stable fixed point in the long time driving limit, the position of the unstable fixed point keeps oscillating.

2. *Distribution of the energy density peaks:*

   An ensemble average of the locations of energy density peaks yields a continuous distribution, as well. Interestingly, for the random driving in Fig. 9, where the deformation function (see (34)) is symmetric about $x = l/2$, the distribution of energy density peaks $\nu_E$ is symmetric with respect to the distribution of operator evolution $\nu_O$ about $x = l/2$. We expect there may be a relation between $\nu_E$ and $\nu_O$. It will be interesting to show rigorously that the distribution of energy density peaks $\nu_E$ is continuous in the heating phase of a randomly driven CFT.

## 2.4 Type I and II exceptional points

Now we study the dynamics at type I and type II exceptional points. There are two salient features we want to highlight. One is that the entanglement entropy grows as $\sqrt{n}$, while the total energy still grows exponentially in time. The other is that the ensemble-averaged distributions of both the operator evolution $\nu_O$ and the energy-momentum density $\nu_E$ are *finite* convex combinations of Dirac measures, which is different from the heating phase where the corresponding distributions are continuous.

As we have commented before, there is an equivalence between type I and type II exceptional points, in the sense of subgroup isomorphism (see Appendix A.3 for details). Therefore, we will mainly focus on the properties of type I exceptional points. The features of time evolution at type II exceptional points can be discussed similarly and some details are in Appendix B.2. For simplicity, we only consider the protocol 1 with two driving Hamiltonians. The protocol 2 shares qualitatively the same features. The fact that the driving Hamiltonians generate only *reflection* matrices at the type I exceptional point makes it possible to compute all quantities analytically.

### 2.4.1 Sub-linear entanglement entropy growth

In this subsection, we will analytically show that the entanglement entropy grows as $\sqrt{n}$ and thus has a vanishing growth rate. Without loss of generality, we only consider the entanglement entropy growth that is caused by the chiral deformation. We will then show that the Lyapunov exponent is also zero so that the relation $\lambda_S = \lambda_L$ still holds in this case.

Let us first derive the general formula of the product of random reflection matrices, which will be useful not only for computing entanglement but also other quantities. Let $M_A'$ and $M_B'$ denote the two non-commuting reflection matrices that are generated by the two driving Hamiltonians. It is convenient to diagonalize one of them by an $M \in$ SU(1, 1), i.e., $M_A = M M_A' M^{-1}$ and $M_B = M M_B' M^{-1}$ with

$$M_A = \begin{pmatrix} i\cosh(\varphi) & e^{i\phi}\sinh(\varphi) \\ e^{-i\phi}\sinh(\varphi) & -i\cosh(\varphi) \end{pmatrix}, \quad M_B = \begin{pmatrix} i & 0 \\ 0 & -i \end{pmatrix}, \tag{45}$$

where $\varphi, \phi \in \mathbb{R}$ and a possible global minus sign is neglected. We also require $\varphi \neq 0$ in order to ensure that $M_A$ and $M_B$ are non-commuting. The random product of $M_A'$ and $M_B'$ reduces to that of $M_A$ and $M_B$. Namely, the matrix product for $n$ steps of random driving becomes

$$\Pi_n = M^{-1} \cdot M_1 M_2 \cdots M_n \cdot M,$$

where $M_j$'s are drawn from $\{M_A, M_B\}$ randomly. Since we are only interested in the limit $n \to \infty$, we can safely ignore $M$ (which has a finite norm) without changing the results qualitatively.

We assume the number of driving steps $n$ to be even $n = 2k$, and the case of odd $n$ can be analyzed similarly. Let $M_{r,s}$ denote the product of $2k$ matrices where the number of $M_A$ and $M_B$ is $k - r + s$ and $k + s - r$ respectively, with $0 \leq r, s \leq k$. The property $M_A^2 = M_B^2 = -\mathbb{I}$ allows for a simple expression

$$M_{r,s} = (-1)^k \begin{pmatrix} \cosh((k-r-s)\varphi) & ie^{i\phi}\sinh((k-r-s)\varphi) \\ -ie^{-i\phi}\sinh((k-r-s)\varphi) & \cosh((k-r-s)\varphi) \end{pmatrix}, \tag{46}$$

regardless of the positions of $M_A$ and $M_B$'s. We also need to determine the total probability $p_{r,s}$ for such terms. For simplicity, we assume $p_A = p_B = 1/2$ and have

$$p_{r,s} = \frac{1}{2^n} \cdot \binom{k}{r} \cdot \binom{k}{s} = \frac{1}{2^n} \cdot \frac{k!}{(k-r)!\,r!} \cdot \frac{k!}{(k-s)!\,s!}. \tag{47}$$

One can check the normalization $\sum_{r,s=0}^{k} p_{r,s} = 1$.

Now, we apply the above general formula to compute the entanglement entropy and Lyapunov exponent. The entanglement entropy (31) in this case becomes

$$\mathbb{E}\big(S_A(n)\big) - S_A(0) = \frac{c}{6} \sum_{r,s=0}^{k} p_{r,s} \cdot \log\Big( \cosh\big[2(k-r-s)\varphi\big] + \sin\phi \cdot \sinh\big[2(k-r-s)\varphi\big] \Big), \tag{48}$$

which can be evaluated numerically and the results are shown by the dotted lines in Fig. 10. One can see a clear $\sqrt{n}$ growth in the long time limit.

Let us confirm this numerical observation by an analytical derivation. The increment of the entanglement entropy after two more steps is

$$\mathbb{E}\big(S_A(n+2)\big) - \mathbb{E}\big(S_A(n)\big) = \frac{c}{6} \cdot \frac{1}{4} \sum_{r,s=0}^{k} p_{r,s} \cdot \log\left( 1 + \frac{1 - \sin^2\phi}{2} \cdot \frac{\cosh(4\varphi) - 1}{[f(k-r-s)]^2} \right), \tag{49}$$

where we have defined $f(x) := \cosh(2x\varphi) + \sin\phi \cdot \sinh(2x\varphi)$. For the special case $1 - \sin^2\phi = 0$, the averaged entanglement entropy will simply oscillate in time with period 2, as will be discussed in detail in Section 2.4.5. In the general case $1 - \sin^2\phi \neq 0$, we analyze the large-$\varphi$ limit to derive an analytical expression. The result turns out to be a good approximation even when $\varphi$ is small. In this case, the entanglement entropy growth is dominated by the terms with $k - r - s = 0$, the total probability of which is

$$\sum_{r,s;\,r+s=k}^{k=n/2} p_{r,s} = \frac{1}{2^n} \binom{n}{n/2}.$$

Then in the large-$\varphi$ limit, the entanglement entropy growth in (49) can be approximated by

$$\mathbb{E}\big(S_A(n+2)\big) - \mathbb{E}\big(S_A(n)\big) \approx \frac{\kappa}{2^n} \binom{n}{n/2},$$

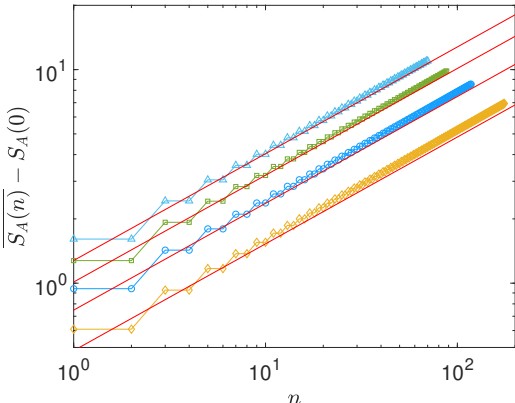

Figure 10: Growth of the averaged entanglement entropy at the type I exceptional points. The dotted data are exact results in (48) with $\phi = 0$ and $c = 1$, and the red solid lines correspond to the approximated analytical results in (51). From top to bottom, the parameters are $\varphi = 10, 8, 6,$ and 4, respectively.

where

$$\kappa = \frac{c}{24} \log \left[ 1 + \frac{1}{2}(1 - \sin^2 \phi) \cdot (\cosh(4\varphi) - 1) \right].$$

In the long time driving limit $n \to \infty$, one can use the Stirling's approximation. In particular, if $n = 4^m$ with $m \in \mathbb{Z}$, the result can be simplified to

$$\mathbb{E}\big(S_A(n+2)\big) - \mathbb{E}\big(S_A(n)\big) \approx \frac{\kappa}{2^{m+1/3}} = \frac{\kappa}{2^{1/3}} \cdot \frac{1}{\sqrt{n}}. \tag{50}$$

We then extrapolate the above expression to general $n$ and do an "integral" to have

$$\mathbb{E}\big(S_A(n)\big) - S_A(0) \approx \frac{\kappa}{2^{1/3}} \sqrt{n}, \quad n \gg 1. \tag{51}$$

In Fig. 10, we compare the analytical result (51) and the exact one (48). One can find good agreement even for small $\varphi$. In Section 2.4.3, we will provide a more general derivation of this $\sqrt{n}$ growth by utilizing the distribution of operator evolution, which justifies this growth behavior for arbitrarily small $\varphi > 0$.

Numerically, we also checked the general choice of $p_A$ and $p_B$ (where $p_A + p_B = 1$) in the driving protocol 1, as well as protocol 2 with infinitely many driving Hamiltonians at the type I exceptional point. It is found that the averaged entanglement entropy always grows as $\sqrt{n}$ in the long time limit.

Following the definition, we can find that the entanglement growth rate vanishes in this case

$$\lambda_S \propto \lim_{n \to \infty} \frac{\sqrt{n}}{n} = 0.$$

It is then natural to ask whether the Lyapunov exponent $\lambda_L$ vanishes as well. Plugging (46) into the definition of the Lyapunov exponents (9), we have

$$\lambda_L = \lim_{n \to \infty} \frac{1}{2n} \sum_{r,s=0}^{k} p_{r,s} \cdot \log \left( 2 \cosh[2(k - r - s)\varphi] \right), \tag{52}$$

where we have used the Frobenius norm. By comparing (52) and (48), one can find that their summands are exactly the same by choosing $\sin \phi = 0$ in (48). Since we have shown that

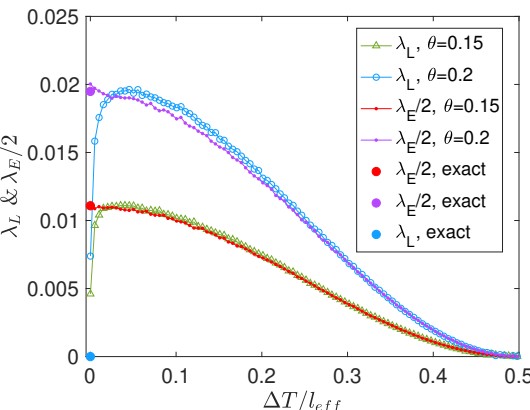

Figure 11: Distribution of $\lambda_E/2$ and $\lambda_L$ with the driving protocol 1 and with the same driving parameters in Fig. 5 (left). We also show the exact results $\lambda_L = 0$, and $\lambda_E/2 = \frac{1}{4}\log[\cosh(\varphi)]$ in (55) with $\varphi = 2\theta$ at the type I exceptional point where $\Delta T/l_{\text{eff}} = 0$.

$\mathbb{E}\big(S_A(n)\big) \propto \sqrt{n}$ for $\sin\phi \neq \pm 1$, it automatically tells us

$$\lambda_L \propto \lim_{n\to\infty} \frac{\sqrt{n}}{n} = 0\,.$$

In the heating phase where Furstenberg's criteria are satisfied, we have shown that $\lambda_L = \lambda_S > 0$. Here we show that the relation $\lambda_L = \lambda_S$ still holds at the type I exceptional points except that both vanish here.

### 2.4.2 Energy evolution

Although the entanglement entropy grows sublinearly as $\sqrt{n}$ and the Lyanpunov exponent vanishes as well, we will show that the total energy still grows exponentially in time. Without loss of generality, we only consider the chiral part of the energy.

By plugging the matrix product (46) into the general formula (30), the ensemble-averaged energy evolution is

$$\mathbb{E}\big(E(n)\big) = \frac{\pi c}{12\,l} \sum_{r,s=0}^{k} p_{r,s} \cdot \cosh\big[2(k-r-s)\varphi\big], \quad n = 2k \in 2\mathbb{Z}\,, \tag{53}$$

where the probabilities $p_{r,s}$ are given in (47). Similar to the analysis on entanglement, we check the increment of energy after another two more steps of random driving, and find that the chiral energy is exactly amplified by a factor

$$\mathbb{E}\big(E(n+2)\big) = \mathbb{E}\big(E(n)\big) \cdot \big[\cosh(\varphi)\big]^2. \tag{54}$$

Furthermore, the total energy at odd $n$ is the same as that at even $n$, i.e., $\mathbb{E}(E(n+1)) = \mathbb{E}(E(n+2))$ for $n \in 2\mathbb{Z}$. Thus, the energy growth exponentially with the rate

$$\lambda_E = \frac{1}{2}\log\big[\cosh(\varphi)\big]\,. \tag{55}$$

Since $\varphi \neq 0$, we always have $\lambda_E > 0$. Fig. 11 shows the exact value of $\lambda_E$ at the type I exceptional point, which is smoothly connected to the $\lambda_E$ in the heating phase. See also Fig. 21 for the comparison of CFT and lattice model calculations for the total energy evolution at the type I exceptional point.

### 2.4.3 Operator evolution and energy-momentum density peak distribution

In this subsection, we discuss the operator evolution and energy-momentum density and show their distinct features compared with the heating phase. We will see that the position of the operator switches between two different points instead of converging to a stable fixed point in the long time. The energy momentum density peak appears also at these two points with probability 1/2 and 1/2.

Suppose the number of driving steps is even $n = 2k$, then a typical product of random matrices $M_{r,s}$ and its probability $p_{r,s}$ are given by (46) and (47). Unless $r+s-k = 0$, different $M_{r,s}$'s share the same fixed points $z_0 = ie^{i\phi}$ and $z_1 = -ie^{-i\phi}$, and $ie^{i\phi}$ ($-ie^{i\phi}$) is the stable one if $(k-r-s)\varphi > 0$ ($< 0$). Solving the distribution of operator evolution at time $n = 2k$ amounts to understanding how a generic point on the unit circle flows after being applied by $M_{r,s}$ once. Depending on the value of $(k-r-s)\varphi$, there are three cases:

1. $(k-r-s)\varphi \gg 0$, then $M_{r,s}$ is able to map any point to $z_0$

2. $(k-r-s)\varphi \ll 0$, then any point flows to $z_1$

3. $(k-r-s)\varphi \sim 1$, applying $M_{r,s}$ once is unable to move the points by large amount.

The first two cases are equally likely to happen because the numbers of $M_{r,s}$ with positive and negative $k-r-s$ are the same, while the last one has a less probability. Their probabilities depend on the value of $\varphi$:

- Large-$\varphi$ limit. In this case, the third case happens only if $r+s = k$. As discussed in Section 2.4.1, the probability of having $r+s = k$ is approximately $\frac{1}{2^{1/3}\sqrt{n}}$. Therefore, in the large $\varphi$ limit, the distribution of operator evolution at a large but finite $n$ is

$$v_O(n) = \frac{1}{2}\left(1 - \frac{1}{2^{1/3}\sqrt{n}}\right)\delta(z-z_0) + \frac{1}{2}\left(1 - \frac{1}{2^{1/3}\sqrt{n}}\right)\delta(z-z_1) + \cdots, \quad n \gg 1, \quad (56)$$

  where $+\cdots$ represents a continuous function that is suppressed by $1/\sqrt{n}$.

- Small-$\varphi$. In this case, $M_{r,s}$ needs a large enough $|k-r-s|\varphi$ to be able to map all the points on the unit circle to its stable fixed point. The probability that this is not satisfied is asymptotically $\frac{\varphi_c}{\varphi\sqrt{n}}$ in the large-$n$ limit, where $\varphi_c$ is an $\mathcal{O}(1)$ constant whose detailed value is not important. Therefore, the distribution of operator evolution becomes

$$v_O(n) = \frac{1}{2}\left(1 - \frac{\varphi_c}{\varphi\sqrt{n}}\right)\delta(z-z_0) + \frac{1}{2}\left(1 - \frac{\varphi_c}{\varphi\sqrt{n}}\right)\delta(z-z_1) + \cdots, \quad n \gg 1, \quad (57)$$

  where $+\cdots$ is a continuous function suppressed by $1/\sqrt{n}$. As we can see, the weight of the Dirac measures becomes smaller than that in the large-$\varphi$ limit.

In both cases, the distribution of operator evolution approaches an equal weight combination of Dirac measures in the long-time limit

$$v_O = \frac{1}{2}\delta(z-z_0) + \frac{1}{2}\delta(z-z_1). \tag{58}$$

Mathematically, $v_O$ is called the *common invariant measure* of the random matrices in $G_\mu$ that is generated by the non-commuting reflection matrices (see Theorem A.6 in the appendix).

Fig. 12 shows a concrete example. The left and middle plots are the operator evolution with $\varphi = 5$ and $\varphi = 1$ driven for the same time. One can see that the distribution is indeed a combination of two Dirac measures, and the one with larger $\varphi$ has larger weight. We further

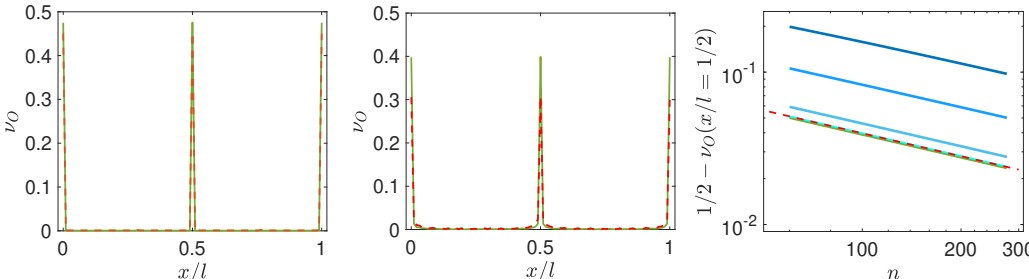

Figure 12: Distribution of operator evolution at type I exceptional point with $\varphi = 5$ (left) and $\varphi = 1$ (middle) in (46). The initial positions of operator are uniformly distributed on the line. We choose $\phi = \pi/2$ in both cases. The driving steps are $n = 4^3$ (red dashed line) and $n = 4^4$ (green solid line). Right: The scaling behavior of $1/2 - \nu_O(x/l = 1/2)$ for $\varphi = 1, 2, 3, 4, 5$ (solid lines from top to bottom). The red dashed line is the analytical result in (56).

check how the the weight of the Dirac measures deviate from $1/2$ and confirm the $1/\sqrt{n}$ for both small and large $\varphi$. The results are shown in the right plot.

The discussion on the behavior of the energy density peak is similar, except that it is related to the unstable fixed point instead of the stable one. Noticing that, at type I exceptional point, the stable and unstable fixed points are interchanged during the random driving, therefore the distribution of the energy density peaks should be the same as that of the operator evolution

$$\nu_E = \nu_O \,. \tag{59}$$

This is different from the heating phase, where $\nu_E \neq \nu_O$ in general, as seen in Fig. 9. In the real space, the above result means that there are on average two energy density peaks at $x_0$ and $x_1$ within each wavelength of deformation, which are related to $z_0$ and $z_1$ on $\partial \mathbb{D}$ by $z_{0(1)} = e^{i \frac{2\pi}{l} x_{0(1)}}$.

We conclude this subsection with a remark. The behavior of operator evolution is also related to the group walking as discussed in Appendix B.3. Fig. 29 shows that, at the exceptional point, the group walking of $\rho_n$ (which determines the operator evolution) only hits two points on $\partial \mathbb{D}$, which correspond to the two peaks in Fig. 12. This is different from the heating phase where the group walking of $\rho_n$ can hit many different possible points on $\partial \mathbb{D}$.

### 2.4.4 Physical picture of entanglement and energy growth

The above analysis on the operator evolution also provides an intuitive way to understand and a shortcut to deriving the $\sqrt{n}$ growth of the entanglement entropy. In this subsection, we give a general discussion on how it works. The additional input information we need is that the entanglement entropy comes from the excitation accumulated at the energy density peaks [42, 43]. Then we apply the same idea to analyze the energy growth and explain why it grows much faster.

The stable and unstable fixed points are interchanged in the driving process. As long as ensemble-averaged quantities are concerned, it is not necessary to distinguish them and we simply call them fixed points.

– *General discussion:*

Suppose the system has been driven for a long enough time. As we have discussed in the previous subsection, there are three configurations of the distribution of operator evolution or equivalently the (chiral) energy density peaks: (a) the peak being at the left fixed point, (b)

the peak being at the right fixed point, (c) no energy-momentum density peaks. Each of them is depicted below, where the solid blue line represents the energy density.

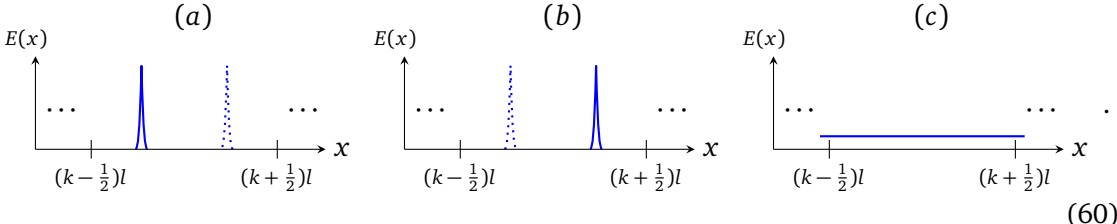

$$\tag{60}$$

The first two happens with an equal probability. In the long time limit $n \to \infty$, the probabilities of these three configurations are

$$p_{(a)} - \frac{1}{2} = p_{(b)} - \frac{1}{2} \sim \mathcal{O}\left(\frac{1}{\sqrt{n}}\right), \quad p_{(c)} \sim \mathcal{O}\left(\frac{1}{\sqrt{n}}\right). \tag{61}$$

Now let us analyze how these energy peaks move and how the entanglement entropy $S_A$ grows accordingly after another two steps of random driving. We choose our subsystem to be $A = [(k-1/2)l, (k+1/2)l]$, which includes two fixed points that can be interchanged in the evolution. If the system is in the configuration (a), there are two possibilities. One is shown in (a.1), where new EPR pairs are generated at the right fixed point (the dotted peak) with one member staying in the subsystem $A$ and the other one moving and out of the subsystem. This process increases $S_A$. The other one is shown in (a.2), where non-local EPR pairs are pumped back to the subsystem $A$, which reduces $S_A$. The above two processes happen when the dotted peak corresponds to a stable or unstable fixed point. Therefore, they have an equal probability and cancel each other completely.

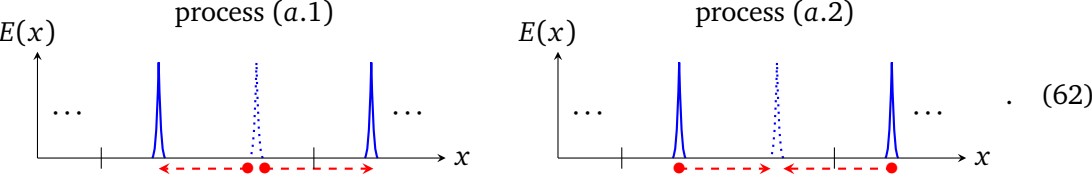

$$\tag{62}$$

As a result, the entanglement entropy growth only comes from the configuration (c), for which there are also two possibilities, shown in (c.1) and (c.2) below. Interestingly, both of them increase the entanglement entropy because there are no previously splitted EPR pairs and either driving generates new EPR pairs.

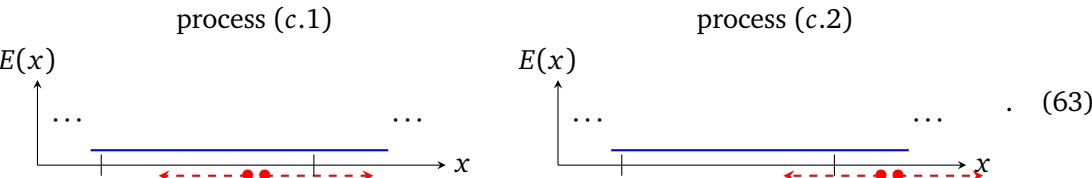

$$\tag{63}$$

Let $\kappa$ denote the amount of entanglement growth due to processes (c.1) and (c.2). Recall that the probability of configuration (c) is $p_{(c)}$ in (61), we have

$$\mathbb{E}(\Delta S_A(n)) := \mathbb{E}(S_A(n+2)) - \mathbb{E}(S_A(n)) = \kappa \cdot p_{(c)} \propto \frac{\kappa}{\sqrt{n}}. \tag{64}$$

Then one can do the integral and obtain

$$\mathbb{E}(S_A(n)) \propto \kappa \cdot \sqrt{n}. \tag{65}$$

This explains the $\sqrt{n}$ growth of entanglement from a general perspective.

Different from the entanglement, the energy growth in process (a.1) and (a.2) does not cancel, and thus is expected to be much faster. The growth behavior has to be analyzed case by case.

*– Concrete example and calculation:*

As an illustration, let us apply the above general picture to the concrete example discussed in Section 2.4.1 and Section 2.4.2. For simplicity, we set $\phi = \pi$ and take the large-$\varphi$ limit.

As we have introduced before, the system is driven by $H_A$ and $H_B$ with equal probabilities. The corresponding SU(1,1) matrices are $M_A$ and $M_B$ defined in (45). Assume that the system is already randomly driven by $n = 2k$ steps and the product of random matrices is given by $M_{r,s}$ defined in (46). The three configurations (a), (b) and (c) in (60) exactly correspond to the three cases $k - r - s > 0$, $< 0$ and $= 0$ respectively. We want to analyze how different configurations evolve in the next two steps.

Let us first analyze the configuration (a) or equivalently $k - r - s > 0$. After another two steps of random driving, the time evolution is given by $M_{r,s}(M_A M_B)$ or $M_{r,s}(M_B M_A)$.[17] They exactly correspond to process (a.1) and (a.2) and each occurs with a probability 1/4. Accordingly, the change of entanglement entropy and energy (of a unit cell) are

$$
\begin{cases}
\text{process } (a.1): & \Delta S_A \approx \dfrac{c}{3} \cdot \varphi, & \Delta E = \dfrac{\pi c}{12l} \left( E_n(r,s) \cosh(2\varphi) + f_n(r,s) \right), \\[2mm]
\text{process } (a.2): & \Delta S_A \approx -\dfrac{c}{3} \cdot \varphi, & \Delta E = \dfrac{\pi c}{12l} \left( E_n(r,s) \cosh(2\varphi) + f_n(r,s) \right),
\end{cases}
$$

where $E_n(r,s) = \cosh(2m\varphi)$, $f_n(r,s) = \sinh(2m\varphi)\sinh(2\varphi)$ and $m = k - r - s$ and $k = n/2$. By summing over different processes, the change of entanglement entropy cancel each other while the energy growth does not. The same result hold for the configuration (b) as well.

Now let us consider the configuration (c) or equivalently $k - r - s = 0$. The non-trivial evolution in two steps is also given by $M_{r,s}(M_A M_B)$ and $M_{r,s}(M_B M_A)$ with the same probability 1/4, which correspond to the process (c.1) and (c.2). In this situation, the change of entropy and energy are

$$
\begin{cases}
\text{process } (a.1): & \Delta S_A \approx \dfrac{c}{3} \cdot \varphi, & \Delta E = \dfrac{\pi c}{12l} \cosh(2\varphi), \\[2mm]
\text{process } (a.2): & \Delta S_A \approx \dfrac{c}{3} \cdot \varphi, & \Delta E = \dfrac{\pi c}{12l} \cosh(2\varphi)
\end{cases}
$$

and both of them grows after summing over all the processes.

The probabilities of the three configurations are just what we have used in obtaining the distribution of operator evolution (56). We can then sum over all the contributions and have

$$
\begin{aligned}
\mathbb{E}(S_A(n+2)) - \mathbb{E}(S_A(n)) &\approx \frac{c}{6} \cdot \varphi \cdot \frac{1}{2^{1/3}\sqrt{n}}, \\
\mathbb{E}(E(n+2)) - \mathbb{E}(E(n)) &= \mathbb{E}(E(n)) \cdot \cosh^2(\varphi),
\end{aligned}
\tag{66}
$$

the integral of which further leads to

$$
\mathbb{E}(S_A(n)) - S_A(0) \approx \frac{c \cdot \varphi}{6} \cdot \frac{1}{2^{1/3}} \cdot \sqrt{n}, \quad \mathbb{E}(E(n)) = \frac{\pi c}{12l} \cdot [\cosh(\varphi)]^n .
\tag{67}
$$

This confirms our general discussion and the analysis in previous sections.

---

[17]The other two possibilities are $M_{r,s}M_A^2$ and $M_{r,s}M_B^2$, which lead to trivial evolution.

### 2.4.5 Cases with oscillating entanglement entropy

The entanglement entropy growth in (51) holds for most choices of entanglement cuts. There are also special choices of entanglement cuts, with which the entanglement entropy simply oscillates in time.

The case that we discussed in Section 2.4.1 provides a concrete example. When we choose $\sin \phi = \pm 1$, one can check that the entanglement entropy does not grow at all after every two steps

$$\mathbb{E}\big(S_A(n+2)\big) - \mathbb{E}\big(S_A(n)\big) = 0.$$

More explicitly, we have

$$\mathbb{E}\big(S_A(2k)\big) - S_A(0) = 0, \quad \mathbb{E}\big(S_A(2k-1)\big) - S_A(0) = \pm \frac{c \cdot \varphi}{6}, \quad k \in \mathbb{Z}^+, \tag{68}$$

where $\pm$ correspond to $\sin \phi = \pm 1$ respectively, and $c$ is the central charge. See the first plot in Fig. 7 for an illustration. It is emphasized that this oscillating behavior can also be observed in lattice systems. See Fig. 19 for the comparison of CFT and lattice model calculations.

The reason behind such oscillation is the following. Physically, these special choices correspond to cutting through two chiral (or anti-chiral) energy density peaks. Then the degrees of freedom that carry quantum entanglement can flow towards and accumulate at the entanglement cuts. Intuitively, one can consider this process as dragging non-local EPR pairs to the entanglement cuts, which reduces the entanglement between the subsystem and its complement. Because of such processes, one can find that after every two steps of random driving, the growth and decrease of entanglement entropy exactly cancel with each other. This results in a period-2 oscillating behavior of entanglement entropy in (68).

## 2.5 Type III exceptional points

As defined in Section 2.2.2, at type III exceptional points, the SU(1, 1) matrices corresponding to different driving Hamiltonians share *only one* common fixed point. This makes analytical calculation difficult, and thus we investigate the time evolution behavior at type III exceptional points only numerically.

In the following, we will first list different sub-types of type III fixed points, and then study the corresponding entanglement/energy evolution and related features case by case.

### 2.5.1 Sub-types of type III exceptional points

At type III exceptional points, all the driving Hamiltonians are non-elliptic, so are the corresponding SU(1, 1) matrices. Let us consider the protocol 1 with two driving Hamiltonians $H_A$ and $H_B$. There are five different sub-types of type III exceptional points:

1. sub-type 1: Both Hamiltonians are hyperbolic. The *stable* fixed point of one hyperbolic matrix coincides with the *stable* fixed point of the other hyperbolic matrix. Their unstable fixed points are different.

2. sub-type 2: Both Hamiltonians are hyperbolic. The *stable* fixed point of one hyperbolic matrix coincides with the *unstable* fixed point of the other hyperbolic matrix. The other two fixed points do not coincide.

3. sub-type 3: Both Hamiltonians are hyperbolic. The *unstable* fixed point of one hyperbolic matrix coincides with the *unstable* fixed point of the other hyperbolic matrix. Their stable fixed points do not coincide.

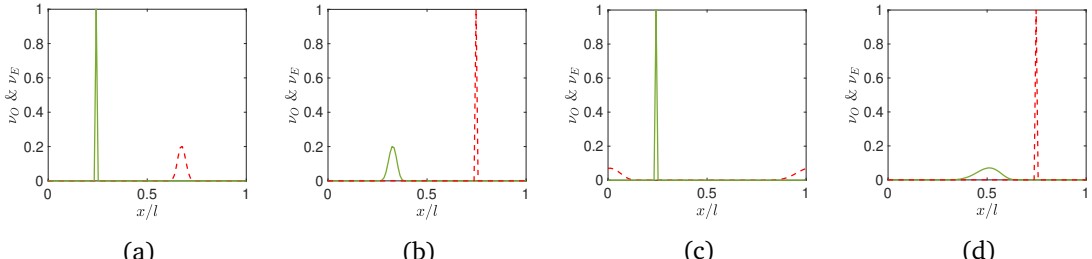

Figure 13: Distribution of the (chiral) operator evolution $\nu_O$ (green solid lines) and the distribution of (chiral) energy density peaks $\nu_E$ (red dashed lines) within one wavelength of deformation. In all the plots, we consider the random driving with two Hamiltonians $H_A$ and $H_B$ with probabilities $1/2$ and $1/2$. For $H_A$ we choose $\sigma_A^0 = \sigma_A^- = 0, \sigma_A^+ = 1$ in (5). For $H_B$, we choose (a) $\sigma_B^0 = -\sigma_B^- = 1/2, \sigma_B^+ = 1$, (b) $\sigma_B^0 = \sigma_B^- = 1/2, \sigma_B^+ = 1$, (c) $\sigma_B^0 = -1, \sigma_B^+ = 0, \sigma_B^- = 1$, and (d) $\sigma_B^0 = 1$, $\sigma_B^+ = 0, \sigma_B^- = 1$ in (5). These choices correspond to sub-type-1, -3, -4, and -5 of type III exceptional points. The driving times are $T_A/l = T_B/l = 1/50$. We take $N_{\text{sample}} = 10^6$ in the numerical calculation.

4. sub-type 4: One Hamiltonian is hyperbolic and the other is parabolic. The *stable* fixed point of the hyperbolic matrix coincides with the unique *clockwise* (or *counter-clockwise*) fixed point of the parabolic matrix. [18]

5. sub-type 5: One Hamiltonian is hyperbolic and the other is parabolic. The *unstable* fixed point of the hyperbolic matrix coincides with the unique *clockwise* (or *counter-clockwise*) fixed point of the parabolic matrix.

For more than two driving Hamiltonians, the corresponding type III exceptional points can be understood based on the above sub-types. For example, if we consider $N$ hyperbolic Hamiltonians, and all the stable fixed points of $m$ Hamiltonians coincide with all the unstable fixed points of the other $N-m$ Hamiltonians (and there are no other coincident fixed points), then the time evolution features will be similar to sub-type 2 exceptional point.

As summarized in table 1, these different sub-types of type III exceptional fixed points may have different time evolution features. We give detailed discussions on these features in the rest of this section.

### 2.5.2 Operator evolution and energy-momentum density peak distribution

Since there are common invariant measures at the type III exceptional point (see Theorem A.6), it is expected that the distributions of operator evolution and energy-momentum density peaks may have discrete measures in the long time driving limit $n \to \infty$. The main results are summarized as follows, with the concrete examples discussed later.

1. sub-type 1: The distribution of operator evolution is a Dirac measure, and the distribution of energy-momentum density peaks is continuous.

2. sub-type 2: The distributions of the operator evolution and the energy-momentum density depend on the relative strength of the stable and unstable fixed points that coincide

---

[18]Different from the fixed points of a hyperbolic matrix, which are either attractive or repelling, the fixed point of a parabolic matrix has a chirality. That is, the operator that is not initially located at the unique fixed point (see (A.2)) will flow to the fixed point in a clockwise or counter-clockwise manner.

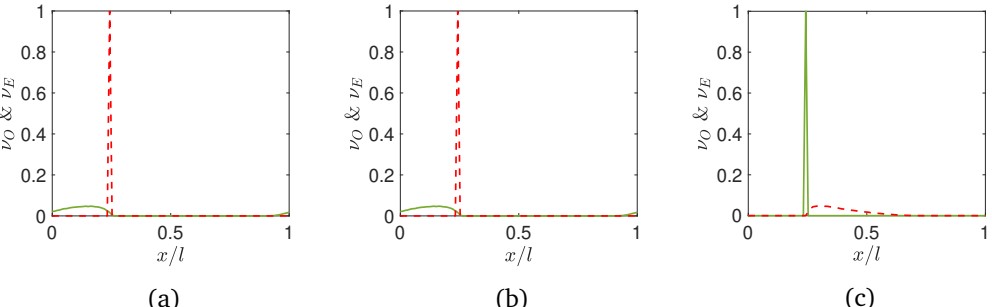

Figure 14: Distribution of the (chiral) operator evolution (green solid lines) and the (chiral) energy density peaks (red dashed lines) at sub-type-2 of type III exceptional point. We use protocol 1 with two Hamiltonians $H_A$ and $H_B$ with probabilities $1/2$ and $1/2$. The two Hamiltonians $H_A$ and $H_B$ are defined with $\sigma_A^0 = \sigma_A^- = 0, \sigma_A^+ = 1$ and $\sigma_B^0 = -\sigma_B^- = 1/2, \sigma_B^+ = -1$ (see (5) for definitions) The time durations of driving are $T_A/l = 1/40$ and (a) $T_B/l = 1/30$, (b) $T_B/l = 1/40$, (c) $T_B/l = 1/50$. We take $N_{\text{sample}} = 10^6$ in the numerical calculation.

with each other. [19] If the stable (unstable) fixed point is stronger than the unstable (stable) fixed point, then the distribution of operator evolution (energy-momentum density peaks) is a Dirac measure, and the distribution of energy-momentum density peaks (operator evolution) is continuous. If the strengths of the coincide stable and unstable fixed points are the same, then the distributions of both operator evolution and energy-momentum density peaks are continuous.

3. sub-type 3: The distribution of operator evolution is continuous, and the distribution of energy-momentum density peaks is a Dirac measure.

4. sub-type 4: The distribution of operator evolution is is a Dirac measure, and the distribution of energy-momentum density peaks is continuous.

5. sub-type 5: The distribution of operator evolution is continuous, and the distribution of energy-momentum density peaks is a Dirac measure.

In short, it is found that at sub-type 1, 3, 4, and 5 exceptional points, either the distribution of operator evolution or the distribution of energy-momentum density peaks is a Dirac measure, and the other is continuous (see Fig. 13). One way to understand it is the following. For example, at the sub-type 1 exceptional point, since the stable fixed points of both hyperbolic Hamiltonians coincide, and the operators must flow to this coincident stable fixed point in the long time limit. Therefore, the operator evolution has a Dirac measure distribution at this stable fixed point ($x/l = 1/4$ in Fig. 13 (a)). On the other hand, the locations of the energy-momentum density peaks are determined by the unstable fixed points. They are continuously distributed because the two unstable fixed points do not coincide. The other three plots in Fig. 13 can be understood similarly.

The feature of sub-type 2 exceptional point is more interesting. The distributions of operator evolution and energy-momentum density peaks depend on the relative strength of the coincident stable and unstable fixed points. As shown in Fig. 14, we choose the parameters

---

[19]Here the strength of the stable (unstable) fixed point characterizes how much an operator is attracted to (or repelled from) the fixed point in a single driving step. More precisely, the strength is determined by $|\partial f(x)/\partial x| \cdot T$ at the stable (unstable) fixed point $x = x_\bullet$ ($x_\circ$), where $f(x)$ is the deformation function in (5), and $T$ is the driving time. For a given driving Hamiltonian with stable (or unstable) fixed points, the time of driving $T$ determines strength of the fixed points). See the concrete examples in Fig. 14.

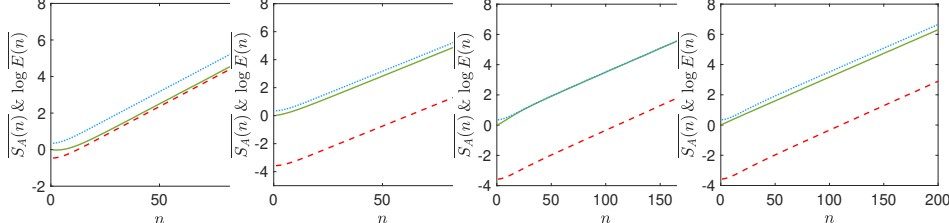

Figure 15: Averaged entanglement entropy evolution $3\mathbb{E}(S_A(n))$ (green solid lines) and the total energy evolution $\frac{1}{2}\log\mathbb{E}(E(n))$ (red dashed lines), and $\mathbb{E}(\log\|\Pi_n\|)$ (blue dotted lines) at sub-type 1, 3, 4, and 5 exceptional points (from left to right). The subsystem $A$ is chosen as $[(k-1/2)l, (k+1/2)l]$ where $k \in \mathbb{Z}$. The slopes of the above mentioned three lines correspond to $\lambda_S$, $\lambda_E$, and $\lambda_L$ respectively. The driving protocol as well as the driving parameters are the same as those in Fig. 13, but with $N_{\text{sample}} = 10^3$ here.

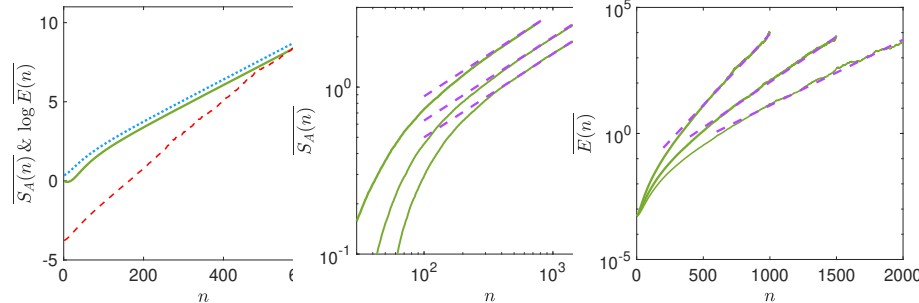

Figure 16: Left: Ensemble-averaged entanglement entropy evolution $3 \cdot \mathbb{E}(S_A(n))$ (green solid lines), the energy evolution $\frac{1}{2}\log\mathbb{E}(E(n))$ (red dashed lines), and $\mathbb{E}(\log\|\Pi_n\|)$ (blue dotted lines) at sub-type 2 exceptional point. Middle: Ensemble-averaged entanglement entropy evolution. The purple dashed lines are fitting with $y \propto x^{1/2}$. Right: Ensemble-averaged energy evolution. The purple dashed lines are fitting with $y \propto e^{\lambda x}$. The driving protocol and driving Hamiltonians are the same as Fig. 14. We choose $T_0/l = 1/40$ and $T_1/l = 1/30$ (left). In the middle and right plots, we chose $T_0/l = T_1/l = 1/40$, $1/50$, and $1/60$ (from top to bottom). In all the plots we do ensemble average over $N_{\text{sample}} = 10^4$.

such that the stable fixed point of $H_A$ ($x_{A,\bullet} = l/4 \mod l$) is coincident with the unstable fixed point ($x_{B,\circ} = l/4 \mod l$) of $H_B$, and $|\partial v_A(x)/\partial x|$ at $x_{A,\bullet}$ equals $|\partial v_B(x)/\partial x|$ at $x_{B,\circ}$. The relative strength of the fixed points is determined by the driving time $T_A$ and $T_B$. For $T_A < T_B$, the strength of the unstable fixed point is stronger. Then one can find that the distribution of energy density peaks is a Dirac measure $\delta(x - x_{A,\bullet})$ at $x_{A,\bullet} = x_{B,\circ}$. On the other hand, the distribution of operator evolution is continuous because the operator evolution cannot be stabilized at any point. For $T_A > T_B$, the result can be similarly understood, except that now the distribution of operator evolution is a Dirac measure and the energy-momentum density peaks are continuously distributed. For $T_A = T_B$, i.e., the strength of the coincident stable and unstable fixed points are the same, it is observed that the distributions of both operator evolution and energy-momentum density peaks are continuous, because neither the operator evolution nor the energy-momentum density peaks can be stabilized in this case.

### 2.5.3 Entanglement and energy evolution in different sub-types

For the sub-type 1, 3, 4, and 5 fixed points, it is found that the entanglement entropy grows linearly in time and the total energy grows exponentially in time, as shown in Fig. 15. In particular, one can observe that

$$\lambda_S = \lambda_L = \lambda_E, \tag{69}$$

where $\lambda_S$ and $\lambda_E$ are the growth rate of the entanglement entropy and the total energy. This is different from the heating phase where $\lambda_S = \lambda_L$ and $\lambda_E \geq \lambda_S$ (See, e.g., Fig. 11, and Fig. 25 in the appendix).

At the sub-type 2 exceptional point, there are in general two kinds of behaviors in the entanglement/energy evolution, as shown in Fig. 16:

1. If the strength of the coincident stable and unstable fixed points are different, then the entanglement entropy grows linearly in time and the total energy grows exponentially in time [See Fig. 16 (left)]. Similar to the heating phase, in general we have $\lambda_E \geq \lambda_S$.

2. If the strength of the coincident stable and unstable fixed points are the same, then the entanglement entropy grows as square root of time and the total energy grows exponentially in time [See Fig. 16 (middle, right)].

It is noted that for the second case above, the features of entanglement/energy evolution are the same as that at the type I and type II exceptional points. The common structure of these three types of exceptional points is that the coincident stable and unstable fixed points have the same strength.[20] As discussed in Section 2.4.4, the physical picture is that such structure will cause the cancellation of entanglement entropy growth during the random driving, which results in a sub-linear entanglement entropy growth.

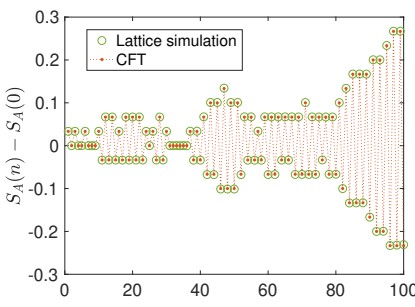 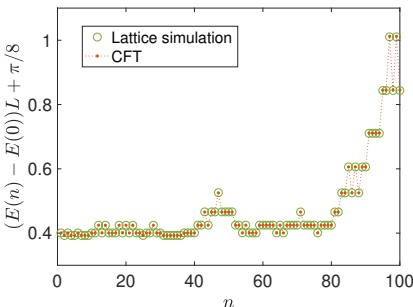

Figure 17: Comparison of the entanglement entropy (left) and the total energy evolution (right) at the type I exceptional point in lattice simulations and CFT calculations with $L = 800$. We consider a single randomly generated driving sequence. The two driving Hamiltonians are $H_0$ and $H_{\theta=0.05}$ with probability $p = 1/2$ and $1/2$. The driving times are $T_0/L = T_\theta/L_{\text{eff}}(\theta) = 1/2$.

## 3 Comparison of lattice model calculations and CFT calculations

In this section, we compare the numerics on a lattice model and the CFT calculations for both the entanglement entropy and the total energy evolution. In particular, we make the

---

[20]It is reminded that there are two pairs of coincident stable/unstable fixed points at type I/II exceptional points, but only one pair of coincident stable/unstable fixed points at type III exceptional points.

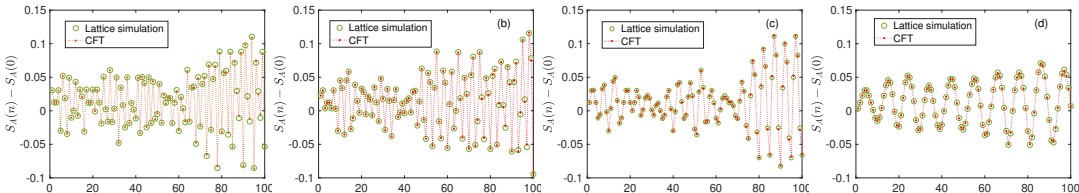

Figure 18: Comparison of lattice simulations and CFT results for the entanglement entropy evolution in the heating phase with a single random sequence. Here we choose the same random sequence as Fig. 17. We take $L = 1600$ and $T_0/L - 1/2 = T(\theta)/L_{\text{eff}}(\theta) - 1/2$ which equals (a) 0.1, (b) 0.2, (c) 0.3, and (d) 0.4.

comparison *for an arbitrary random sequence*. If the comparison agrees well for an arbitrary random sequence, then the ensemble average must also agree. [21]

The lattice model we consider is a free fermion lattice, which has finite sites $L$ with open boundary conditions. Corresponding to (1), we can deform the Hamiltonian in space, and use these deformed Hamiltonians to drive the system in time. We prepare the initial state as the ground state of the homogeneous Hamiltonian $H_0$ with half filling and open boundary conditions, where

$$H_0 = \frac{1}{2} \sum_{j=1}^{L-1} c_j^\dagger c_{j+1} + h.c. . \tag{70}$$

Here $c_j$ are fermionic operator satisfying the anticommutation relations $\{c_j, c_k\} = \{c_j^\dagger, c_k^\dagger\} = 0$, and $\{c_j, c_k^\dagger\} = \delta_{jk}$. The deformed Hamiltonian, with inhomogeneous Hamiltonian density in space, has the form

$$H_1(\theta) = \frac{1}{2} \sum_{j=1}^{L-1} f_j(\theta) c_j^\dagger c_{j+1} + h.c. , \tag{71}$$

where for simplicity we consider the 1-parameter family of deformed Hamiltonians by choosing the deformation function $f_j(\theta) = 1 - \tanh(2\theta) \cdot \cos \frac{2\pi j}{L}$. This deformation is the lattice version of the deformation in (34) by choosing $q = 1$ with open boundary conditions. As a remark, the reason we choose $q = 1$ in (71) is to maximize the wavelength of deformation and do the numerics in an efficient way. One can certainly choose a larger $q \in \mathbb{Z}^+$. In this case, one needs to take a larger $L$ to make a good comparison with the CFT calculation. For later use, one can define the effective length of the system as $L_{\text{eff}}(\theta) = L \cosh(2\theta)$, which characterizes the total time that the quasi-particle needs to travel from one end to the other end of the system. [41]

In the following calculations, we will consider the driving protocol 1 as introduced in Section 2.2.3. That is, we drive the lattice system randomly with $H_0(\theta = 0)$ and $H_1(\theta)$, with fixed time interval $T_0$ and $T_1$, respectively. The probabilities are chosen as $p_0 = p_1 = 1/2$. The phase diagram of this driven system corresponds to Fig. 3 (by replacing $l_{\text{eff}}$ with $L_{\text{eff}}$), where one can observe both heating phases and the type I exceptional point. We will compare the entanglement/energy evolution both at the exceptional point and in the heating phase in the following subsections.

---

[21]One reason we mainly focus on the time evolution in a single random sequence is that it takes a long time to perform ensemble average for a lattice system with a large system size. Nevertheless, in Fig. 19 and Fig. 21, we show the ensemble-averaged results both at the type I exceptional point and in the heating phase, but with a smaller system size. The agreement is still remarkable.

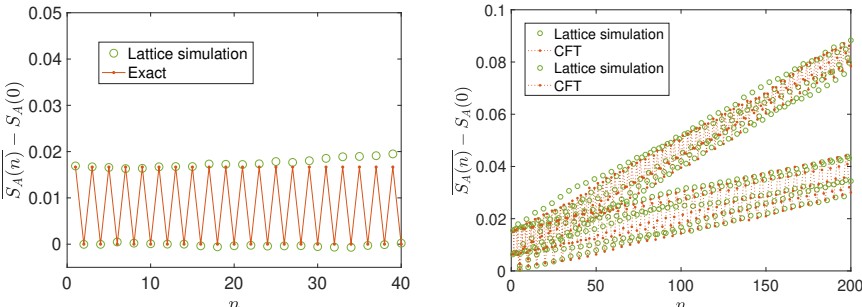

Figure 19: Comparison of lattice simulations and the analytical results/CFT results for the ensemble-averaged entanglement entropy evolution at the type I exceptional point (left) and in the heating phase (right). Left: $T_0/L - 1/2 = T(\theta)/L_{\text{eff}}(\theta) - 1/2 = 0$, $N_{\text{sample}} = 10^4$, and $L = 400$. The red solid line is the analytical CFT result in (72). Right: $T_0/L - 1/2 = T(\theta)/L_{\text{eff}}(\theta) - 1/2 = 0.1$ (top) and 0.3 (bottom), $N_{\text{sample}} = 5000$, and $L = 200$.

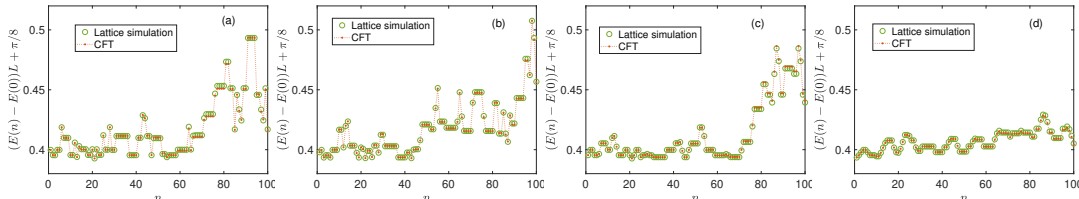

Figure 20: Comparison of lattice simulations and CFT results for the total energy evolution in the heating phase with a single random sequence. Here we choose the same random sequence and driving parameters as Fig. 18.

## 3.1 Time evolution of entanglement entropy

The entanglement entropy evolution of the free fermion lattice model can be calculated based on the Peschel's method [72]. One can refer to the appendix in Ref. [40] for the details of calculation of the entanglement entropy and correlation functions in the time-dependent driven free fermion system. Here we choose the subsystem as $A = [0, L/2]$. The corresponding CFT result of the entanglement entropy evolution can be found in (31).

The entanglement entropy evolution for a single randomly generated driving sequence is compared at the type I exceptional point in Fig. 17, and in the heating phase off the exceptional point in Fig. 18. One can find that the lattice calculations agree with the CFT calculations very well. We also checked other randomly generated driving sequences, and the agreement is also remarkable, as expected.

As discussed in Section 2.4.5, there is an interesting case at the type I exceptional point when the entanglement cuts coincide with the fixed points of operator evolution. This case can be realized in the lattice system in (70) and (71), by choosing $T_0/L = T_\theta/L_{\text{eff}}(\theta) = 1/2$ and $A = [0, L/2]$. In this case, the *ensemble-averaged* entanglement entropy will oscillate instead of increasing in time with

$$\mathbb{E}\big(S_A(2k)\big) - S_A(0) = 0, \quad \mathbb{E}\big(S_A(2k-1)\big) - S_A(0) = \frac{c \cdot \theta}{3}, \quad k \in \mathbb{Z}^+, \tag{72}$$

where $\theta$ is the deformation parameter in (71). Interestingly, this oscillating behavior can be observed in lattice systems, as seen in Fig. 19. As a remark, this oscillating behavior can not last for an arbitrarily long time in the lattice system, because the system is keeping absorb-

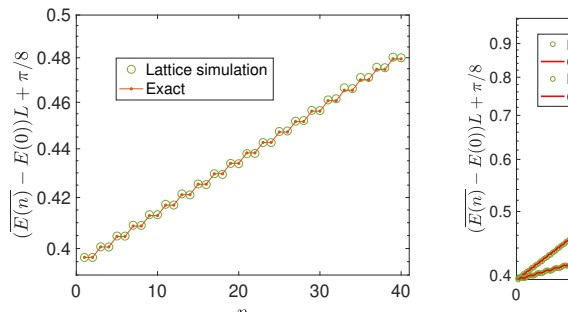
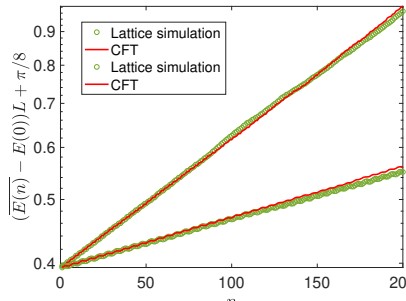

Figure 21: Comparison of lattice simulations and the analytical results/CFT results for the averaged entanglement entropy evolution at the type I exceptional point (left) and in the heating phase (right). The driving parameters are the same as Fig. 19.

ing energy (with energy growing exponentially fast). As discussed in detail in Ref. [42], the agreement between the CFT and lattice calculations will finally break when the higher energy modes (which can no longer be described by CFT) in the lattice system are involved.

In the heating phase, the ensemble-averaged entanglement entropy is shown in Fig. 19, where the numerical calculations agree with the CFT calculations. One can observe that the entanglement entropy grows linearly in time (up to oscillating features).

## 3.2 Time evolution of total energy

We also compare the time evolution of the total energy between the lattice systems and the CFT calculation. In the lattice system, the energy evolution is computed by evaluating $\langle\psi(t)|H_0|\psi(t)\rangle$, where $|\psi(t)\rangle$ is the time dependent wavefunction. In the CFT calculation, the energy evolution is evaluated through (33).

We compare the energy evolution both at the type I exceptional point in Fig. 17 (right plot), and in the heating phase in Fig. 20, for an arbitrarily generated random sequence. One can find that the agreement is remarkable.

We also compare the *ensemble-averaged* total energy evolution. For example, for the type I exceptional point, it is predicted that total energy still grows exponentially in time $n$. For concreteness, for the CFT with open boundary conditions, with the same approach in Section 2.4.2, one can obtain the analytical result of the energy evolution as follows:

$$\mathbb{E}\big(E(2k-1)\big) = \mathbb{E}\big(E(2k)\big) = \frac{\pi c}{8L}\big[\cosh(2\theta)\big]^{2k} - \frac{\pi c}{6L}, \quad \text{where } k \in \mathbb{Z}^+, \tag{73}$$

where $\theta$ is the deformation parameter in (71). The comparison of the lattice calculations and the above exact result can be found in Fig. 21 (left plot), and the agreement is remarkable. Similar to the entanglement entropy evolution, as we take a longer driving time, here the agreement between the CFT and lattice calculations will finally break down when the higher energy modes (which are no longer described by CFT) in the lattice system are involved. [42]

In the heating phase, the comparison of ensemble-averaged total energy evolution can be found in Fig. 21, where the agreement is also remarkable.

# 4 Conclusion and discussion

In this work, we have systematically studied the randomly driven CFT in (1+1) dimensions with $SL_2$ deformations and given a complete classification and characterization of all possible types of random drivings where the driving Hamiltonians are independent and identically

distributed (i.i.d). The heating phase and (different types of) exceptional points can be determined by examining whether Furstenberg's criteria are satisfied or not. In general, the exceptional points only have zero measure in the parameter space. We characterize the heating phase and different types of exceptional points by the time evolution of entanglement entropy and energy, and the distributions of operator evolution and energy-momentum density peaks, with the main features summarized in Table 1 and Section 1.2. Although we are mainly interested in the physical phenomena, we hope to emphasize that some physical properties in the randomly driven CFT can be rigorously proved or discussed based on Furstenberg's theorems and the related mathematics. For example, the linear growth of entanglement entropy and the continuous distribution of operator evolution in the heating phase can be rigorously proven, etc. We also study the entanglement/energy evolution in lattice models, and the results agree with the CFT calculations remarkably well.

Now, we give several comments and discussions in order:

1. *Comparison of periodically, quasi-periodically and randomly driven CFTs*

   In this work and the prior one [39], the general features of periodically, quasi-periodically and randomly driven CFTs with $SL_2$ deformations have been systematically studied. The possible phase diagrams under different drivings can be summarized as follows:

   | Driven CFTs | Heating phase | Non-heating phase | Critical |
   |:---:|:---:|:---:|:---:|
   | Periodic | ✓ | ✓ | ✓ |
   | Random | ✓* | × | × |
   | Fibonacci | ✓ | × | ✓ |

   $$\text{(74)}$$

   where $*$ indicates the presence of exceptional points. It is emphasized that the quasi-periodic driving in (74) is only for the Fibonacci sequence. There are certainly other types of quasi-periodic drivings, where one may observe both heating phases and non-heating phases with phase transitions. [73] In short, it is possible to have non-heating phases, where the entanglement/energy does not grow in time, in both periodically and quasi-periodically driven CFTs. On the contrary, there is no non-heating phase in the randomly driven CFTs, since the total energy still grows exponentially in time even at the exceptional points.

2. *Accidental exceptional points in randomly driven CFTs*

   In this work, we are mainly interested in the ensemble average of the time evolution of related physical quantities in a randomly driven CFT. Most recently in [53], it was found that if one considers a *single* trajectory of random driving that depends on an additional parameter, there almost surely exist the *accidental exceptional points*. More explicit, the accidental exceptional points are searched by tuning the additional parameter while keeping the random sequence unchanged. Then for each sequence, there can be a certain parameter where the Lyapunov exponent reaches zero. See Appendix A.5 for the observation of accidental exceptional points in a randomly driven CFT. We emphasize that the locations of these accidental exceptional points are sequence dependent and therefore unpredictable, and hence the name 'accidental'. From this point of view, the exceptional points studied in our work are *intrinsic*, as their locations in the parameter space can be predicted.

3. *Random drivings beyond $SL_2$ deformation*

   Recently in Refs. [43, 44], the periodically driven CFT with $SL_2$ deformations has been generalized to the case with arbitrarily smooth deformation, where the underlying algebra is the infinite-dimensional Virasoro algebra. It is still an open question on the

fate of quasi-periodically/randomly driven CFTs with general deformations. For example, in the random drivings, it is apparent that Furstenberg's theorem will no longer be applicable with the general deformations. Mathematically, we need a generalized version of Furstenberg's theorem. That is, instead of considering random sequences of Möbius transformations, one needs to consider random sequences of circle maps. See, e.g., Refs. [74, 75] for examples of the mathematical studies of random circle maps. If there is a generalized version of Furstenberg's theorem, it is interesting to further ask whether there exist generalized "exceptional points" and study the properties at these points. [22]

A closely related setup on the general random driving in $(1+1)d$ CFTs was recently studied in Ref. [76], where the initial state is chosen to be a short-range entangled gapped state (which is approximated by a regularized conformally invariant boundary state), and the entanglement evolution is dominated by the "EPR" pairs emitted from the initial state. The effect of random driving is to introduce fluctuations on the linear growth of the entanglement entropy, which was found to be related to the Kardar-Parisi-Zhang class fluctuations. It will be interesting to investigate the net effect of a general random driving by considering a CFT ground state as the initial state and study the generic features in the entanglement and energy evolution, *etc*.

4. *Ensemble averaged CFTs*

It was recently found that a simple model of gravity in two dimensions (JT gravity) is dual to a random ensemble of quantum mechanical systems [77]. One may wonder if something similar happens in higher dimensions, such as the ensemble average of random CFTs. Recent studies along this direction include the ensemble average of free CFTs over the moduli space [78, 79]. What we do in randomly driven CFT is to perform random averaging for the unitaries in the non-equilibrium dynamics, which are not replaceable by a single quantum quench. We believe the gravity dual of such randomly driven CFTs will be interesting and deserves a future study. [23]

5. *Other related works on time-dependent driven CFTs*

There are certainly other setups of time-dependent driven CFTs besides the one considered in this work. For example, instead of considering bulk driving, one can consider boundary driving as in [80–83]. Interestingly, in the setup of moving mirrors in CFTs, the non-equilibrium dynamics can also be studied based on conformal maps as well [82, 83]. We look forward a connection between our setup and the moving-mirror setup.

Moreover, there is a recent setup on periodically driven perturbed CFTs, where the time-dependent perturbation is relevant [84]. In this case, the driven system is no longer at the critical point (which is different from our setup where the system is always critical so that an exact solution in the whole parameter space exists). Although determining the phase diagram is a challenging problem in this setup, one can still approach the

---

[22]We expect there may still exist exceptional points in randomly driven CFTs with general deformations. First, the physical picture for the exceptional points in $SL_2$ deformed randomly driven CFTs is that the EPR pairs are pumped in and out of the subsystem during the random driving, which results in partial cancelation of the entanglement entropy. For example, in type I/II exceptional points, the stable and unstable fixed points for operator evolution are kept switching during the random driving. Second, with this physical picture, we can consider a random driving by evolving the system with generally deformed non-commuting Hamiltonians $H_1$ and $H_2$ randomly in time. In particular, we choose $H_1$ and $H_2$ such that they both have two fixed points (one is stable and the other is unstable) in the operator evolution. In addition, the stable (unstable) fixed point of $H_1$ coincides with the unstable (stable) fixed point of $H_2$ in real space. We expect the exceptional point (where the Lyapunov exponent is zero) may be observed with such choice of driving Hamiltonians.

[23]We thank Jie-Qiang Wu for suggesting this interesting point to us.

stable region and investigate related physical properties when the driving frequency is large. Furthermore, it is also interesting to compare our setup to the Floquet setups as studied in AdS4/CFT3 in [85, 86]. It will be interesting to ask what is the randomly driven version of the above stories.

We also want to mention the recent work [87], where the deformed Hamiltonians are used for the preparation of "a part of an infinite system" on a finite-size quantum simulator. In particular, the deformed Hamiltonian can effectively "cool' the bulk to zero temperature. In a forthcoming work, we will show such cooling effect can be exactly studied in $(1+1)d$ CFTs.

# Acknowledgements

We thank for helpful discussions with Dan Borgnia, Daniel Jafferis, Bo Han, Eslam Khalaf, Ching Hua Lee, Ivar Martin, Shinsei Ryu, Hassan Shapourian, Tsukasa Tada, Michael Widom, Jie-Qiang Wu, and Yahui Zhang. XW, RF and AV are supported by a Simons Investigator award (AV) and by the Simons Collaboration on Ultra-Quantum Matter, which is a grant from the Simons Foundation (651440, AV). AV and RF are supported by the DARPA DRINQS program (award D18AC00033). YG is supported by the the Simons Foundation through the "It from Qubit" program.

# A  Basics of Furstenberg's theorem in randomly driven CFTs

In this appendix, we introduce some basics of Furstenberg's theorem and its application in randomly driven CFTs. There are many useful review materials on Furstenberg's theorem and their applications, e.g. [51] and [52].

## A.1  Preliminaries

Fursbenberg's theorem (see Section 1.1) on random products of $SL(n, \mathbb{R})$ matrices can be applied to randomly driven CFTs, because $SU(1,1)$ is isomorphic to $SL(2, \mathbb{R})$. A quick way to see the isomorphism is through the following 1-to-1 map from $SL(2, \mathbb{R})$ to $SU(1,1)$

$$\begin{pmatrix} a & b \\ c & d \end{pmatrix} \mapsto \begin{pmatrix} \alpha & \beta \\ \beta^* & \alpha^* \end{pmatrix} = \begin{pmatrix} \frac{(a+d)+i(b-c)}{2} & \frac{(a-d)-i(b+c)}{2} \\ \frac{(a-d)+i(b+c)}{2} & \frac{(a+d)-i(b-c)}{2} \end{pmatrix} = Q \begin{pmatrix} a & b \\ c & d \end{pmatrix} Q^{-1}, \qquad (A.1)$$

where $a, b, c, d \in \mathbb{R}$ with $ad - bc = 1$ and consequently $|\alpha|^2 - |\beta|^2 = 1$. Here $Q = \frac{1}{\sqrt{2}} \begin{pmatrix} 1 & -i \\ 1 & i \end{pmatrix}$ is a unitary matrix and geometrically corresponds to the Cayley transform: $SL(2, \mathbb{R})$ group acts on upper half plane $\mathbb{H} = \{z = x + iy, x, y \in \mathbb{R}, y > 0\}$ via the linear fractional transformation (in this case, the Möbius transformation) $z \mapsto \frac{az+b}{cz+d}$ while $SU(1,1)$ group acts on the unit disk $\mathbb{D} = \{w \in \mathbb{C}, |w| < 1\}$ via the linear fractional transformation $w \mapsto \frac{\alpha w + \beta}{\beta^* w + \alpha^*}$. These two group actions are related via the Cayley transform $Q: w = \frac{z-i}{z+i}$ that maps the upper half plane $\mathbb{H}$ to the unit disk $\mathbb{D}$. It is also straightforward to extend the group actions to the boundaries $\partial \mathbb{H}$ and $\partial \mathbb{D}$. In the random products of $SL(2, \mathbb{R})$ matrices, one usually needs to consider the action of $g \in SL(2, \mathbb{R})$ on the unit vector $\vec{x} = (x_1, x_2)^T \in \mathbb{RP}^1$, with the expression $g\vec{x} = \begin{pmatrix} a & b \\ c & d \end{pmatrix} \begin{pmatrix} x_1 \\ x_2 \end{pmatrix} = \begin{pmatrix} ax_1 + bx_2 \\ cx_1 + dx_2 \end{pmatrix}$. One can first map the unit vector to the real axis as $\pi(\vec{x}) = x_1/x_2$ if $x_2 \neq 0$, and $\pi(\vec{x}) = \infty$ if $x_2 = 0$. Then $g\vec{x}$ can be mapped

to the action of $g \in \mathrm{SL}(2, \mathbb{R})$ on the boundaries $\partial \mathbb{H} := \{z = x + iy, x, y, \in \mathbb{R}, y = 0\}$ as $g \cdot z = \pi\left(g \cdot \pi^{-1}(z)\right) = \frac{az+b}{cz+d}$, which is the Möbius transformation introduced above.

It is known there are three types of $\mathrm{SU}(1,1)$ matrices as follows. Let $M \in \mathrm{SU}(1,1)$ be different from $\pm \mathbb{I}$. Then $M$ is elliptic if $\mathrm{Tr}(M) \in (-2, 2)$, parabolic if $\mathrm{Tr}(M) = \pm 2$, and hyperbolic if $\mathrm{Tr}(M) \in (-\infty, -2) \cup (2, \infty)$. If $M(z) = z$, we will say $z$ is the fixed point of $M$. On the unit circle $\partial \mathbb{D}$ where $\partial \mathbb{D} := \{z \in \mathbb{C}, |z| = 1\}$, there is one fixed point for the parabolic matrix, and two fixed points for the hyperbolic matrix. For the elliptic matrix, the two fixed points are not on the unit circle $\partial \mathbb{D}$. The distribution of fixed points for different types of $\mathrm{SU}(1,1)$ matrices can be visualized as follows:

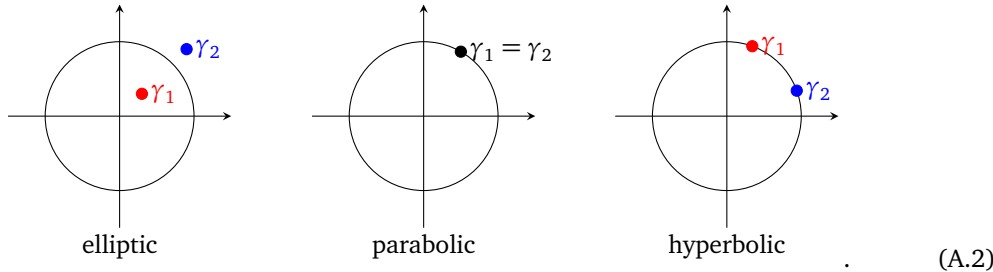

$$\text{elliptic} \qquad \text{parabolic} \qquad \text{hyperbolic} \qquad . \tag{A.2}$$

For example, if $M \in \mathrm{SU}(1,1)$ is hyperbolic, then one has $M(\gamma_{1,2}) = \gamma_{1,2}$, where $\gamma_{1,2} \in \partial \mathbb{D}$. In addition, one fixed point is stable and the other is unstable. Suppose $\gamma_1$ is stable and $\gamma_2$ is unstable, then for arbitrary $z \in \partial \mathbb{D}$ and $z \neq \gamma_2$, one has $\lim_{n \to \infty} M^n(z) \to \gamma_1$.

Moreover, if $M \in \mathrm{SU}(1,1)$ is elliptic, there exists $V \in \mathrm{SU}(1,1)$, so that there exists a similarity transformation $M = VUV^{-1}$, where $U = \mathrm{diag}(e^{i\theta}, e^{-i\theta})$ is diagonal and unitary. If $M$ is hyperbolic, there exist $V \in \mathrm{SU}(1,1)$ so that $M = V P_{x, \phi=0} V^{-1}$, where

$$P_{x,\phi} = \begin{pmatrix} \cosh(x) & e^{i\phi} \sinh(x) \\ e^{-i\phi} \sinh(x) & \cosh(x) \end{pmatrix}, \quad \text{with } x, \phi \in \mathbb{R}. \tag{A.3}$$

Note that $P_{x,\phi} P_{y,\phi} = P_{x+y,\phi}$, and $P_{x,\phi} = V P_{x,\phi=0} V^{-1}$, where $V = \mathrm{diag}(e^{i\frac{\phi}{2}}, e^{-i\frac{\phi}{2}})$.

In the following, we introduce some basic properties of the random products of $\mathrm{SL}(2, \mathbb{R})$ (and therefore $\mathrm{SU}(1,1)$) matrices. We begin with the seminal result of Furstenberg and Kesten [88].

**Theorem A.1.** *Let $Y_1, \cdots, Y_n$ be i.i.d. matrices in $\mathrm{SL}(2, \mathbb{R})$. There exist real numbers $\lambda_+$ and $\lambda_-$ such that*

$$\lim_{n \to \infty} \frac{1}{n} \log ||Y_n \cdots Y_1|| = \lambda_+ \quad \text{and} \quad \lim_{n \to \infty} \frac{1}{n} \log ||(Y_n \cdots Y_1)^{-1}||^{-1} = \lambda_-, \tag{A.4}$$

*with probability 1.*

The numbers $\lambda_+$ and $\lambda_-$ are called extremal Lyapunov exponents. For arbitrary $M \in \mathrm{SL}(2, \mathbb{R})$, since $||M|| \geq 1 \geq ||M^{-1}||^{-1}$, then we have $\lambda_+ \geq 0 \geq \lambda_-$. The extremal Lyapunov exponents $\lambda_+$ and $\lambda_-$ may be viewed as functions of the data $M_1, \cdots, M_m; p_1, \cdots, p_m$ where $M_i \in \mathrm{SL}(2, \mathbb{R})$ and $p_i$ are probabilities with $p_1 + \cdots + p_m = 1$. The probability vectors $(p_1, \cdots, p_m)$ can vary in the open simplex

$$\Delta^m = \{(p_1, \cdots, p_m) : p_1 > 0, \cdots, p_m > 0, p_1 + \cdots p_m = 1\}.$$

Then we have the following theorem [71]

**Theorem A.2.** *The extremal Lyapunov exponents $\lambda_\pm$ depend continuously on $(M_1, \cdots, M_m; p_1, \cdots, p_m) \in \mathrm{SL}(2, \mathbb{R})^m \times \Delta^m$ at all points.*

Based on the above theorem, one can prove rigorously that the distribution of Lyapunov exponents in, e.g., Fig. 5, is continuous.

Next, in Furstenberg's theorem (see Theorem 1.1), one needs to consider the strongly irreducible property, essentially, how the randomly chosen matrices act on the vectors in $\mathbb{R}^d$. Denote the projective space $P(\mathbb{R}^d)$ or $\mathbb{RP}^{d-1}$ as the set of directions (or unit vectors) in $\mathbb{R}^d$. Then given a subgroup $G_\mu \subset \text{SL}(d, \mathbb{R})$, we say that $G_\mu$ is strongly irreducible if there does not exist *a finite set* $V \in \mathbb{RP}^{d-1}$ such that $M(V) = V$ for any $M \in G_\mu$. In our randomly driven CFTs, the time evolution of operators is governed by $\text{SU}(1,1)$ matrices rather than $\text{SL}(2,\mathbb{R})$ matrices. In this case, the strongly irreducible condition can be rephrased in terms of how $M \in \text{SU}(1,1)$ acts on the points in $\partial \mathbb{D}$, as described in Section 1.1.

Now, we provide a supplementary discussion on the invariant measure. As discussed in Section 2.3.3, the invariant measures correspond to the distribution of operator evolution in the long time limit. In fact, the invariant measure is directly related to the Lyapunov exponent through the following theorem.

**Theorem A.3.** *Let $\{M_n, n \geqslant 1\}$ be a sequence of i.i.d. matrices in $\text{SL}(2, \mathbb{R})$ with the distribution $\mu$. Suppose that Furstenberg's criteria are satisfied, and if $\nu$ is the $\mu$-invariant distribution on $\mathbb{RP}^1$, then the Lyapunov exponent can be expressed as:*

$$\gamma = \int \int \log \frac{||Mx||}{||x||} d\mu(M) d\nu(\vec{x}). \tag{A.5}$$

In addition, from Theorem 2.5, it is known that if Furstenberg's criteria are satisfied, then the operator evolution converges to a Dirac measure in a single random sequence. On the contrary, the convergence of the distribution of operator evolution to a Dirac measure only implies the norm growth (not necessarily in an exponential fashion) : [51]

**Lemma A.4.** *Let $\nu \in \mathcal{M}(\partial \mathbb{D})$ be continuous, and let $\Pi_n = M_1 \cdots M_n$ be a sequence in $\text{SU}(1,1)$ such that $\Pi_n \nu \to \delta(z - z_*)$ where $z_* \in \partial \mathbb{D}$. Then $\lim_{n \to \infty} ||\Pi_n|| \to \infty$.*

In the main text, we use Furstenberg's theorem and related theorems to rigorously prove some physical properties in the heating phase, as summarized below:

1. Theorem 1.1 → Provide the criteria to determine the heating phase and exceptional points.

2. Theorem 2.3 → The entanglement entropy grows linearly in time as $\mathbb{E}(S_A(n)) \approx \frac{\lambda_L \cdot c}{3} n$.

3. Theorem 2.5 → The operator position approaches a certain stable fixed point in the long time limit in a *single* random driving sequence.

4. Theorem 2.5 and Lemma 2.4 → The ensemble-averaged distribution of operator evolution must be continuous in space.

5. Theorem A.2 → The Lyapunov exponents (and therefore the growth rate of entanglement entropy) are continuously distributed in the parameter space.

## A.2 Common invariant measures at the exceptional points

In this appendix, we study the common invariant measures at different types of exceptional points. Let us first consider the simplest case, i.e., the invariant measure of a *single* $M \in \text{SU}(1,1)$. Given $M \in \text{SU}(1,1)$ and a probability measure $\nu$ on $\partial \mathbb{D}$, we say that $\nu$ is $M$-invariant if and only if $M(\nu) = \nu$. For example, if $M \in \text{SU}(1,1)$ is hyperbolic, then

$\nu = \{\lambda\delta(z - e^{i\theta_0}) + (1 - \lambda)\delta(z - e^{i\theta_1})\big|\lambda \in [0, 1]\}$, where $e^{i\theta_0}$ and $e^{i\theta_1}$ are the two fixed points of $M$.

In the following theorem, we list the invariant measures for different types of SU(1, 1) matrices. The proof can be found in Ref. [54].

**Theorem A.5.** *Let $M \in$ SU(1, 1), then*

1. *If M is hyperbolic, the invariant measures are precisely the convex combinations of the point masses at the two fixed points of M. That is, the M-invariant measure is $\nu = \{\lambda\delta(z - e^{i\theta_0}) + (1 - \lambda)\delta(z - e^{i\theta_1})\big|\lambda \in [0, 1]\}$, where $e^{i\theta_0}$ and $e^{i\theta_1}$ are the two fixed points of M.*

2. *If M is parabolic, the unique invariant measure is the point mass at M's unique fixed point. That is, the M-invariant measure is $\nu = \delta(z - e^{i\theta_0})$, where $e^{i\theta_0}$ is the unique fixed point of M.*

3. *If M is elliptic and the eigenvalues of M are not roots of unity, then M has a unique invariant measure described as follows. If $(1, re^{i\varphi})^T$ with $r < 1$ is an eigenvector of M, the invariant measure is $P_r(\theta, -\varphi)\frac{d\theta}{2\pi}$ where $P_r$ is the Poisson kernel.* [24]

4. *If M is elliptic and the eigenvalues of M are roots of unity, let n be the smallest integer so that $M^n = \mathbb{I}$ or $-\mathbb{I}$. Let $\theta_0 = 0$, $\theta_1, \cdots, \theta_{n-1}$ be a reordering of $\{\varphi|\varphi = M^j(1), j = 0, 1, \cdots, n-1\}$ so that $0 = \theta_0 < \theta_1 < \cdots < \theta_{n-1} < 2\pi$. Let $\omega$ be an arbitrary probability measure on $[\theta_0, \theta_1)$. Then $\nu = \frac{1}{n}\sum_{j=0}^{n-1} M^j(\omega)$ is M-invariant.*

Based on the above theorem, one can further consider the *common invariant measure* of $M_A$ and $M_B$, i.e., $\mathcal{I}(M_A) \cap \mathcal{I}(M_B)$, where $\mathcal{I}(M)$ denotes the $M$-invariant measure as introduced in the above theorem. More concretely, the common invariant measure of $M_A$ and $M_B$ are determined as follows. [54]

**Theorem A.6.** *Let $M_A$, $M_B \in$ SU(1, 1) be distinct and different from $\pm\mathbb{I}$. Suppose also $M_A \neq -M_B$. Then the common invariant measure $\mathcal{I}(M_A) \cap \mathcal{I}(M_B) \neq \emptyset$ if and only if*

1. *when $M_A$ and $M_B$ are both reflection matrices. In this case $\mathcal{I}(M_A) \cap \mathcal{I}(M_B)$ is always nonempty and has a single element $\frac{1}{2}\delta(z - e^{i\theta_0}) + \frac{1}{2}\delta(z - e^{i\theta_1})$, where $e^{i\theta_0}$ and $e^{i\theta_1}$ are the two fixed points for $M_C = M_A M_B$, which is always hyperbolic.*

2. *when $M_A$ is non-elliptic and $M_B$ is elliptic, if and only if $M_A$ is hyperbolic, $M_B$ is a reflection matrix, and $M_B$ permutes the two fixed points of $M_A$. In this case, $\mathcal{I}(M_A) \cap \mathcal{I}(M_B)$ is then $\{\frac{1}{2}\delta(z - e^{i\theta_0}) + \frac{1}{2}\delta(z - e^{i\theta_1})\}$ where $e^{i\theta_0}$ and $e^{i\theta_1}$ are the two fixed points of $M_A$.*

3. *when $M_A$ and $M_B$ are both non-elliptic, if and only if $M_A$ and $M_B$ have a common fixed point. More concretely, $\mathcal{I}(M_A) \cap \mathcal{I}(M_B) \neq \emptyset$ is either $\{\delta(z - e^{i\theta_0})\}$ if $M_A$ and $M_B$ have a single common fixed point $e^{i\theta_0}$, or $\{\lambda\delta(z - e^{\theta_0}) + (1 - \lambda)\delta(z - e^{i\theta_1})\big|\lambda \in [0, 1]\}$ if $M_A$ and $M_B$ have a pair of common fixed points $e^{i\theta_0}$ and $e^{i\theta_1}$. In the later case, both $M_A$ and $M_B$ are hyperbolic and they commute with each other.*

4. *when $M_A$ and $M_B$ are both elliptic and at least one is not a reflection matrix, if and only if $M_A$ and $M_B$ commute.*

Based on the above theorem, one can classify different types of exceptional points where Furstenberg's criteria are no longer satisfied. See Section 2.2 for details. Furthermore, it turns out the common invariant measures at type I/II exceptional points correspond to the distribution of operator evolution and energy density peaks in the long driving limit $n \to \infty$ (See Section 2.4).

---

[24]The Poisson kernel is defined as $P_r(\theta, \varphi) = \frac{1-r^2}{1-2r\cos(\theta-\varphi)+r^2}$.

### A.3 Equivalence between type I and type II exceptional points

In this appendix, we show that type I and type II exceptional points as defined in Section 2.2.1 are equivalent to each other. Let us consider the case that the subgroup $G_\mu \subset \mathrm{SU}(1,1)$ is generated by two non-commuting matrices $M_A$ and $M_B$. Generalization to the case with more than two generating matrices is straightforward.

First, we show that for an arbitrary type I exceptional point, there is a corresponding type II exceptional point. Consider $M_A$ and $M_B$ as the two non-commuting reflection matrices in a type I exceptional point. The corresponding generating matrices in type II exceptional point can be chosen as $M_C$ and $M_A$ (or $M_B$), where $M_C = M_A M_B$ is always a hyperbolic matrix. Now we show that $M_A$ (or $M_B$) switches the two fixed points of $M_C$, which are denoted as $e^{i\theta_1}$ and $e^{i\theta_2}$, respectively. Here $e^{i\theta_{1,2}}$ are fixed points of $M_C$ indicates that $M_C(e^{i\theta_1}) = e^{i\theta_1}$ and $M_C(e^{i\theta_2}) = e^{i\theta_2}$. It is noted that $M_A^2 = M_B^2 = -\mathbb{I}$, based on which one can find $M_C^{-1} = M_B M_A$, $M_A M_C M_A^{-1} = M_C^{-1}$, and $M_B M_C M_B^{-1} = M_C^{-1}$. Since $e^{i\theta_{1,2}}$ are also fixed points of $M_C^{-1}$, then we have $M_C^{-1}(e^{i\theta_1}) = M_A M_C M_A^{-1}(e^{i\theta_1}) = e^{i\theta_1}$, based on which we can obtain $M_C(M_A^{-1}(e^{i\theta_1})) = M_A^{-1}(e^{i\theta_1})$. Since $M_A$ is a reflection matrix, which does not have a fixed point, then we can only have $M_A^{-1}(e^{i\theta_1}) = e^{i\theta_2}$, or $M_A(e^{i\theta_2}) = e^{i\theta_1}$. Similarly, one has $M_A(e^{i\theta_1}) = e^{i\theta_2}$. That is, $M_A$ switches the two fixed points of the hyperbolic matrix $M_C$. Therefore, the random driving with generating matrices $M_C$ and $M_A$ corresponds to the type II fixed point.

Second, we show that for an arbitrary type II exceptional point, there is a corresponding type I exceptional point. For simplicity, let us first choose the hyperbolic matrix of the form $M_C = P_{x,\phi=0}$ in (A.3), such that the two fixed points correspond to $e^{i\theta_{1,2}} = \pm 1$. The allowed reflection matrix that can permute these two fixed points can only be of the form $M_A = \begin{pmatrix} i\cosh\theta & \pm i\sinh\theta \\ \mp i\sinh\theta & -i\cosh\theta \end{pmatrix}$. In this case, one can check explicitly that $M_B := M_C \cdot M_A^{-1} = \begin{pmatrix} -i\cosh(x \mp \theta) & i\sinh(x \mp \theta) \\ -i\sinh(x \mp \theta) & i\cosh(x \mp \theta) \end{pmatrix}$ is also a reflection matrix. Next, let us consider the general hyperbolic matrix $\widetilde{M}_C$, which can be expressed as $\widetilde{M}_C = V P_{x,\phi=0} V^{-1} = V M_C V^{-1}$ according to (A.3), where $V \in \mathrm{SU}(1,1)$. Then the allowed reflection matrix that permutes the two eigenvectors of $\widetilde{M}_C$ are of the form: $\widetilde{M}_A = V \cdot M_A \cdot V^{-1}$. Then one can check that the matrix $\widetilde{M}_B := \widetilde{M}_C \cdot \widetilde{M}_A^{-1}$ are of the form $\widetilde{M}_B = V \cdot M_C M_A^{-1} \cdot V^{-1} = V \cdot M_B \cdot V^{-1}$, which is always a reflection matrix. [25] In other words, for each type II exceptional point, one can find the corresponding type I exceptional point that is generated by two reflection matrices.

Therefore, we have shown that type I and type II exceptional points are equivalent to each other, in the sense that they can be generated by the same subgroup $G_\mu \subset \mathrm{SU}(1,1)$.

### A.4 Invariant measure and operator evolution in the heating phase

In this appendix, we describe the procedures to obtain the invariant measure in the heating phase. Based on Theorem 2.5 and Lemma 2.4, this invariant measure corresponds to the distribution of operator evolution in the long time driving limit. Therefore, we just need to study the distribution of operator evolution as $n \to \infty$.

As seen in Fig. 22, one considers operators that are uniformly distributed on the circle with coordinates $e^{\frac{2\pi i x}{l}}$. Then the operators evolve under a random driving. Since Furstenberg's criteria are satisfied, the operators will finally flow to a stable fixed point. It is noted that the location of this stable fixed point is randomly distributed. Based on Theorem 2.5 and Lemma

---

[25]If $M$ is reflection matrix, then $VMV^{-1}$ where $V \in \mathrm{SU}(1,1)$ is also a reflection matrix, since $\mathrm{Tr}(VMV^{-1}) = 0$ and $(VMV^{-1})^2 = -\mathbb{I}$.

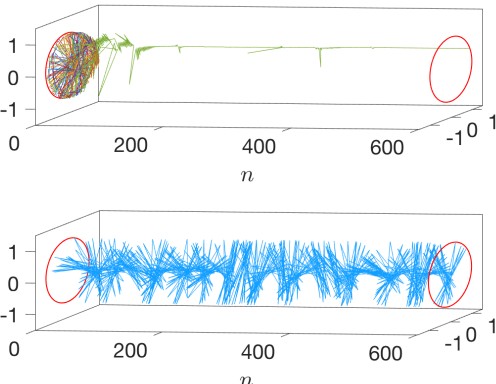

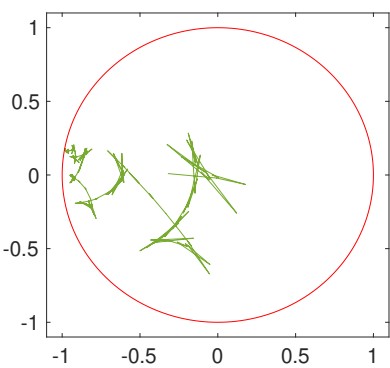

Figure 22: Left (top): Trajectories of operator evolution ($e^{i\frac{2\pi x_n}{l}}$) in a single random driving sequence. We choose 20 different initial positions homogeneously distributed on the circle. Left (bottom): The trajectory of energy density peaks $e^{i\frac{2\pi x_{\text{peak}}}{l}}$ in a single random sequence. We choose $\Delta T/l_{\text{eff}} = 0.01$. The driving protocol and other parameters are the same as those in Fig. 7. Right: A sample plot of group walking of $\rho$ in a randomly driven CFT. In the long time driving limit $n \to \infty$, $\rho$ will approach a certain stable fixed point on $\partial \mathbb{D}$ exponentially fast.

2.4, one can find the ensemble average of these stable fixed points are continuous, which are the same as the invariant measure. For example, one can refer to Fig. 9 for an ensemble average of the stable fixed points of operator evolution.

The stable fixed point of operator evolution reflects the structure of random products of SU(1, 1) matrices. This is related to property ($i$) of Theorem 2.5. One can refer to Appendix B.3 for more details.

As a comparison, we also show the trajectory of the energy density peaks in the random driving. In contrast to the operator evolution, the locations of the energy density peaks keep changing (See Fig. 22). Interestingly, the ensemble average of energy density peaks are continuous and seem to be related to the distribution of operator evolution, as seen in Fig. 9.

## A.5 Accidental exceptional points

The existence of accidental exceptional points was recently studied in Ref. [53]. In the random driving, one introduces an additional parameter. That is, by fixing the *same* random sequence, one changes the driving parameter continuously. In this case, it is shown in Ref. [53] that there almost surely exist the accidental exceptional points, where the Lyapunov exponents could drop to zero.

This phenomenon can be observed in a randomly driven CFT. As seen in Fig. 23, we drive the CFT randomly with two Hamiltonians $H(\theta = 0)$ and $H(\theta \neq 0)$ with probabilities 1/2 and 1/2. By fixing the same random sequence, and changing the driving parameter $\Delta T/l_{\text{eff}} = T_\theta/l_{\theta,\text{eff}} - 1/2$ continuously, one can observe that the Lyapunov exponents $\lambda_L$ may drop to zero at certain points. It is emphasized that the locations of these accidental exceptional points are not predicted. In other words, by choosing another random sequence, the distribution of accidental exceptional points will change accordingly. This is different from the "intrinsic" exceptional points as discussed in the main text. The *accidental* exceptional point cannot be observed after doing an ensemble average, but the "intrinsic" exceptional points can be observed with and without doing ensemble average.

Nevertheless, we checked the scaling behavior of $\lambda_L$ near those accidental exceptional

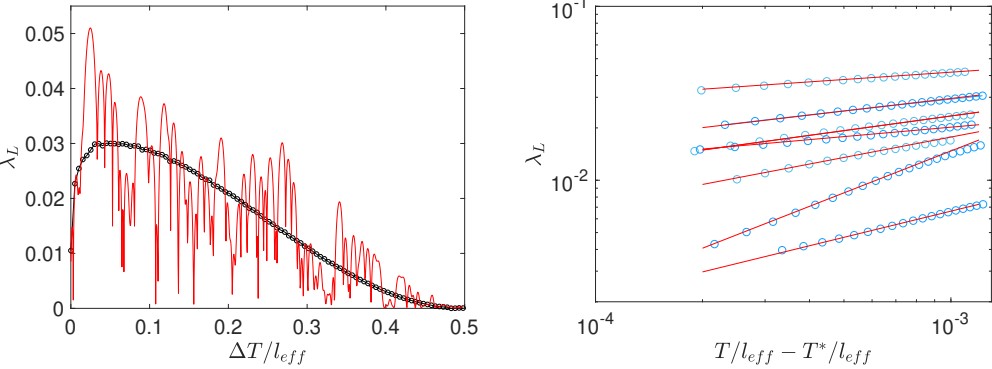

Figure 23: Left: Distribution of Lyapunov exponents obtained from ensemble average (black circles) and in a single random sequence (red solid line) where the random sequence is kept the *same* for different driving parameters $\Delta T/l_{\text{eff}}$. The driving protocol and parameters are the same as the left plot in Fig. 5, with $\theta = 0.25$. Right: A sample plot of the scaling behavior of Lyapunov exponents near different accidental exceptional points. The locations of accidental exceptional points are denoted by $T^*/l_{eff}$. Blue dots correspond to the numerical calculations and the red lines are fittings according to $y \propto x^\alpha$. From top to bottom, the values of $\alpha$ are 0.14, 0.24, 0.28, 0.18, 0.39, 0.8, and 0.5, respectively.

points (See Fig.23). It is observed that the scaling exponents may take different values near different accidental exceptional points. For example, in the sample plot in Fig.23, the exponents range from 0.14 to 0.8. It may be interesting to study these *accidental* exceptional points further in detail somewhere else. In the current work, we are mainly interested in the behaviors that can be observed even after doing an ensemble average.

# B   More on entanglement entropy evolution and others

In this appendix, we present more details on the features of entanglement entropy and energy evolution in the heating phase and at the exceptional points.

## B.1   Entanglement entropy evolution

### B.1.1   General formula

In this appendix, we give a derivation of the entanglement entropy evolution of subsystem $A = [x_1, x_2]$ in a randomly driven CFT.

We start from the two-point correlation function $\langle\Psi_n|\mathcal{O}(x_1)\mathcal{O}(x_2)|\Psi_n\rangle$, where $|\Psi_n\rangle$ is the wavefunction after $n$ steps of driving and $\mathcal{O}(x)$ is a general primary field with conformal dimension $(h, \overline{h})$. Here $\mathcal{O}(x_i)$ is defined on the spacetime cylinder. We do the computation in the imaginary time and thus use the coordinate $w = \tau + ix$. Let us consider a conformal mapping $z = e^{\frac{2\pi q w}{L}} = e^{\frac{2\pi w}{l}}$ to map the $w$-cylinder to the $q$-sheet $z$-Riemann surface (see Fig. 2), on which the operator evolution of $\mathcal{O}(z_1)$ and $\mathcal{O}(z_2)$ is determined by Eq.(23). Next, we map the $q$-sheet $z$-Riemann surface to the complex $\zeta$-plane via a conformal mapping $\zeta = z^{1/q}$, and one can obtain

$$\langle\Psi_n|\mathcal{O}(w_1,\overline{w}_1)\mathcal{O}(w_2,\overline{w}_2)|\Psi_n\rangle = \prod_{i=1,2}\left(\frac{\partial\zeta_i}{\partial w_i}\right)^h \prod_{i=1,2}\left(\frac{\partial\overline{\zeta}_i}{\partial\overline{w}_i}\right)^{\overline{h}} \langle\mathcal{O}(\zeta_1,\overline{\zeta}_1)\mathcal{O}(\zeta_2,\overline{\zeta}_2)\rangle_\zeta, \quad \text{(B.1)}$$

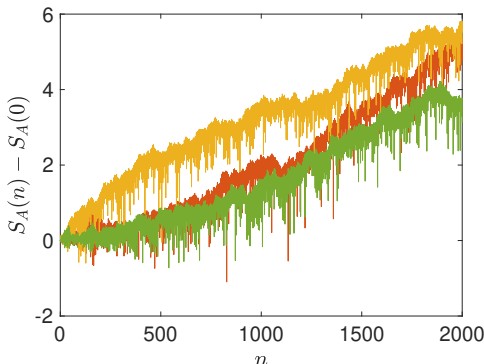

Figure 24: A sample plot of entanglement entropy evolution in the heating phase without performing ensemble average. The driving protocol and parameters are the same as Fig. 7, with $\Delta T/l_{\text{eff}} = 0.1$.

where $w_j = 0 + ix_j$. The above equation can be explicitly evaluated in terms of the SU(1, 1) matrix elements in $\Pi_n$ in (24). It is a product of the holomorphic and anti-holomorphic parts. For example, the contribution of the holomorphic part in Eq.(B.1) can be expressed as

$$
\left(\frac{2\pi}{L}\right)^{2h} \cdot \frac{z_1^h}{(\beta_n^* z_1 + \alpha_n^*)^{2h}} \cdot \frac{z_2^h}{(\beta_n^* z_2 + \alpha_n^*)^{2h}} \cdot \left(\frac{\alpha_n z_1 + \beta_n}{\beta_n^* z_1 + \alpha_n^*}\right)^{(\frac{1}{q}-1)h} \left(\frac{\alpha_n z_2 + \beta_n}{\beta_n^* z_2 + \alpha_n^*}\right)^{(\frac{1}{q}-1)h}
$$
$$
\cdot \left[\left(\frac{\alpha_n z_1 + \beta}{\beta_n^* z_1 + \alpha^*}\right)^{\frac{1}{q}} - \left(\frac{\alpha_n z_2 + \beta}{\beta_n^* z_2 + \alpha^*}\right)^{\frac{1}{q}}\right]^{-2h},
\tag{B.2}
$$

where $z_i = e^{\frac{2\pi w_i}{l}}$. The contribution of the anti-holomorphic part can be obtained by replacing $\alpha_n \to \alpha_n'$, $\beta_n \to \beta_n'$ and $z_i \to \bar{z}_i$ in the above equation. Noting that $z$ lives on a $q$-sheet Riemann surface (see Fig. 2), one should be careful when evaluating Eq.(B.2), by tracking if $z_i$ cross the branch cuts and move from one layer to another. This is subtle but important especially when the system is in a heating phase. The relative distance between $z_1$ and $z_2$ will depend on whether there are energy-momentum density peaks between them [42].

Then the $m$-th Renyi entropy evolution of subsystem $A = [x_1, x_2]$ can be obtained based on Eqs.(B.1) and (B.2), by studying the correlation function of twist operators:

$$
S_A^{(m)}(n) = \frac{1}{1-m} \log \langle \Psi_n | \mathcal{T}_m(x_1) \overline{\mathcal{T}}_m(x_2) | \Psi_n \rangle,
\tag{B.3}
$$

where the twist operators $\mathcal{T}_m$ ($\overline{\mathcal{T}}_m$) are primary operators with conformal dimensions $h = \bar{h} = \frac{c}{24}(m - \frac{1}{m})$. The entanglement entropy of subsystem $A$ can be obtained as $S_A = \lim_{m\to 1} S_A^{(m)}$. For example, let us take $A = [kl + \delta, (k+1)l + \delta]$ where $0 \leqslant \delta < l$ and $k \in \mathbb{Z}$. With this choice, the subsystem $A$ is in

$$
S_A(n) - S_A(0) = \frac{c}{3}\left(\log\left|\alpha_n \cdot e^{\frac{2\pi i\delta}{l}} + \beta_n\right|\right).
\tag{B.4}
$$

For $\delta = l/2$, the above result reduces to (25). For $\delta = 0$, the expression is also very simple, with $S_A(n) - S_A(0) = \frac{c}{3}\left(\log\left|\alpha_n + \beta_n\right|\right)$.

In Section 2.3.1 in the main text, we have proven that in the heating phase of a randomly driven CFT, for arbitrary choices of $\delta$, the entanglement entropy of subsystem $A = [kl + \delta, (k+1)l + \delta]$ will grow as $S_A(n) - S_A(0) = \frac{c}{3} \cdot \lambda_S \cdot n$ for $n \to \infty$. In particular, one has $\lambda_S = \lambda_L$. As an illustration, we give a sample plot of the entanglement entropy

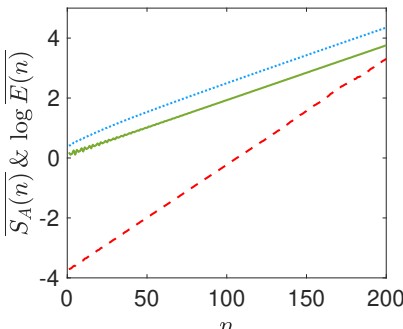
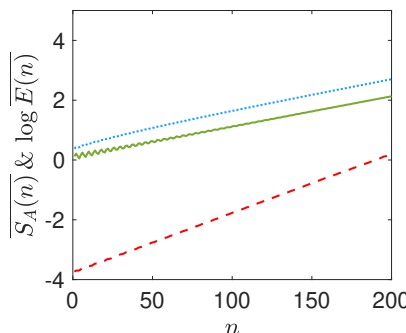

Figure 25: Averaged entanglement entropy evolution $3\mathbb{E}(S_A(n))$ (green solid lines) and the total energy evolution $\frac{1}{2}\log\mathbb{E}(E(n))$ (red dashed lines), and $\mathbb{E}(\log\|\Pi_n\|)$ (blue dotted lines) in the heating phase. The slopes of the above mentioned three lines correspond to $\lambda_S$, $\lambda_E$, and $\lambda_L$ respectively. The driving protocol as well as the driving parameters are the same as those in the left plot of Fig. 5. We choose $\Delta T/l_{eff} = 0.1$ (left) and 0.25 (right) respectively, $\theta = 0.2$, and $N_{\text{sample}} = 5 \times 10^5$ here.

evolution in a single random sequence in Fig. 24. The ensemble-averaged time evolution of the entanglement entropy, logarithmic of norm growth, and the energy of the driven system can be found in Fig. 25. One can observe that $\lambda_S = \lambda_L$ and $\lambda_E \neq \lambda_L$.

### B.1.2 Early time-evolution of entanglement entropy in the heating phase

The early time-evolution of entanglement entropy in the heating phase is also interesting. As seen in Fig.7, near the exceptional point, it takes some time for the entanglement entropy to reach a linear growth. We find the time scale before reaching a linear growth is determined by the 'distance' from the exceptional point, which is characterized by the dimensionless parameter $\Delta T/l_{\text{eff}} := T_\theta/l_{\theta,\text{eff}} - 1/2$ (See Fig.7). In the following, we will discuss the entanglement entropy evolution near the exceptional point and the trivial point respectively.

Near the exceptional points, we find that the entanglement entropy evolution can be well described by

$$\mathbb{E}(S_A(n)) - S_A(0) = \frac{c}{3}\log\left[\cosh(\lambda_L \cdot n)\right] + c_0, \tag{B.5}$$

where $c$ is the central charge, and $c_0$ is a constant. See Fig. 26 for the comparison. In the limit $n \to \infty$, one can reproduce the linear-growth result $\mathbb{E}(S_A(n)) - S_A(0) = \frac{c}{3} \cdot \lambda_L \cdot n$. One can interpret the time scale before reaching a linear growth based on Eq.(B.5). First, the linear growth can be well observed when $\lambda_L \cdot n \gg 1$. Second, the time scale $n_p$ before reaching a linear growth can be approximated by $\lambda_L \cdot n_p \sim 1$. That is, $n_p \sim 1/\lambda_L$. As we approach the exceptional point, one has $\lambda_L \propto (\Delta T/l_{\text{eff}})^{0.19}$ as studied in Fig.6. Therefore, we have $n_p \propto (\Delta T/l_{\text{eff}})^{-0.19}$. For $n \ll n_p \sim 1/\lambda_L$, the averaged entanglement entropy grows as $\mathbb{E}(S_A(n)) - S_A(0) \simeq \frac{c}{6} \cdot \lambda_L^2 \cdot n^2 + c_0$.

As a remark, the expression in (B.5) is similar to the entanglement entropy evolution after a global quantum quench in CFT. See Ref. [89] for a semi-infinite subsystem and Ref. [90] for a finite subsystem. It will be interesting to derive analytically the time evolution of entanglement entropy in the whole time region in a randomly driven CFT.

Near the trivial point, we find that the entanglement entropy evolution can be well described by

$$\mathbb{E}(S_A(n)) - S_A(0) = \frac{c}{3}\lambda_L \cdot n + c_0, \tag{B.6}$$

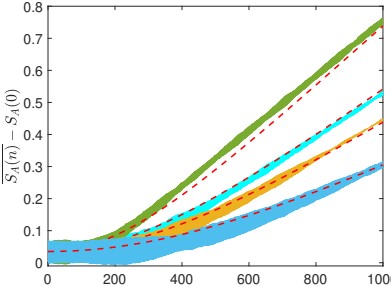 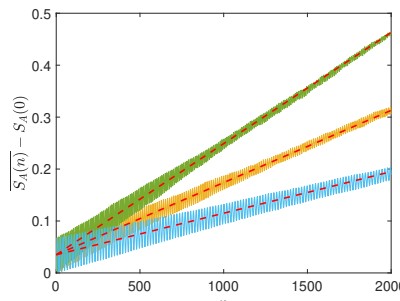

Figure 26: Left: Entanglement entropy evolution in the heating phase near the exceptional point. The red dashed lines are fitting with Eq.(B.5) by choosing different $\lambda_L$, and fixing $c = 1$ and $c_0 = 0.035$. From top to bottom, we choose $\Delta T/l_{\text{eff}} = 5 \times 10^{-5}$, $1 \times 10^{-5}$, $5 \times 10^{-6}$, and $1 \times 10^{-6}$, respectively. Right: Entanglement entropy evolution in the heating phase near the trivial point. The red dashed lines are fitting with Eq.(B.6) by choosing different $\lambda_L$, and fixing $c_0 = 0.035$. From top to bottom, we choose $\Delta T/l_{\text{eff}} = 0.4$, $0.42$, and $0.44$, respectively. Other parameters are the same as those in Fig.7.

where $c_0$ is a constant. Compared to the entanglement entropy evolution near the exceptional point, we do not observe the region $[0, n_p]$ with a slowly growing $S_A(n)$ before reaching a linear growth, as seen in Fig. 26 (right plot).

In addition, it is noted that the entanglement entropy evolution near both the exceptional point and the trivial point have some oscillating features. It is observed that these oscillating features will die out in the long time driving limit, i.e., $n \gg 1/\lambda_L$.

## B.2  Phase diagram including type II exceptional points

In this appendix, we give an example of phase diagram that contains type II exceptional points. These exceptional points form a line in the parameter space.

Let us consider the driving protocol 1 as introduced in Section 2.2.3, i.e., there are two randomly chosen Hamiltonians $H_0$ and $H_1$. To observe type II exceptional points, we require one driving Hamiltonian is elliptic (say, $H_0$) and the other is hyperbolic ($H_1$). Denoting the corresponding SU(1, 1) matrices as $M_0(T_0/l_{0,\text{eff}})$ and $M_1(T_1/l)$ respectively, then it is known that $M_0$ becomes reflection at $T_0/l_{0,\text{eff}} = 1/2$. To have type II exceptional points, it is required that the reflection matrix $M_0$ can permute the two fixed points of $M_1$. Depending on whether this permutation exist or not, one can find that the phase diagrams will be one of the two cases as follows:

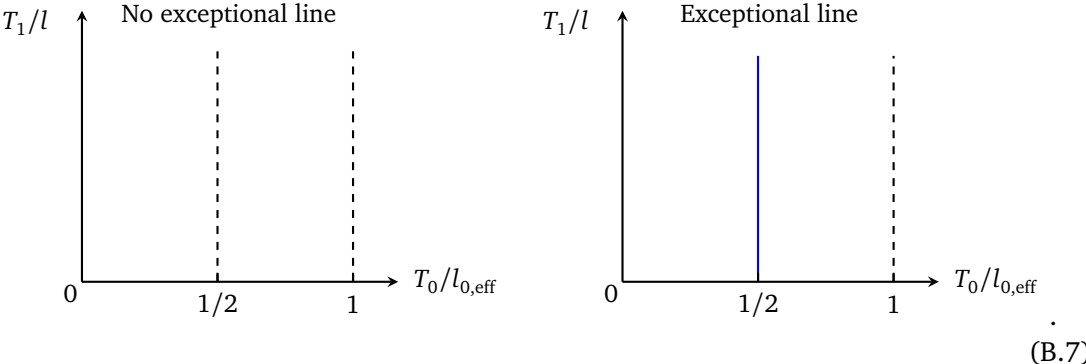

$$\tag{B.7}$$

That is, if $M_0(T_0/l_{0,\text{eff}} = 1/2)$ cannot permute the two fixed points of the hyperbolic matrix $M_1$, then there are only heating phases, as shown in the left of (B.7). On the other hand, if

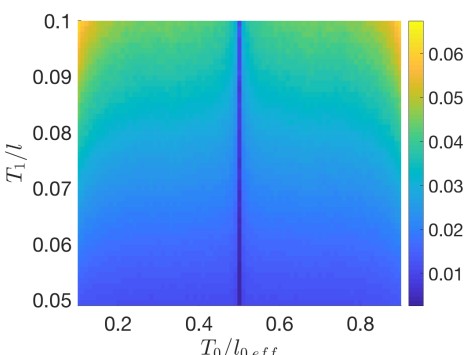 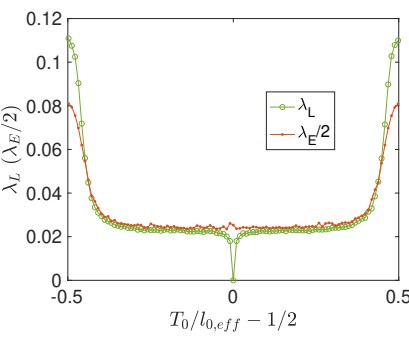

Figure 27: Left: Distribution of Lyapunov exponents $\lambda_L$. We drive the CFT randomly with $H_0$ and $H_1$. For $H_1$, we choose $\sigma^0 = \sigma^- = 0$ and $\sigma^+ = 1$ in (5). For $H_0$ we choose $\sigma^+ = \sigma^- = 0$ and $\sigma^0 = 1$ in (5). The probabilities are $p_0 = p_1 = 1/2$. See also (B.7) for the schematic plot of the phase diagram. We take $N_{\text{sample}} = 5 \times 10^3$. Right: Comparison of $\lambda_L$ and $\lambda_E/2$ along $T_1/l = 0.07$ in the left plot. $\lambda_E$ are obtained by taking $N_{\text{sample}} = 5 \times 10^4$.

$M_0(T_0/l_{0,\text{eff}} = 1/2)$ permutes the two fixed points of the hyperbolic matrix $M_1$, then there are type II exceptional points which form lines along $T_0/l_{0,\text{eff}} = n + 1/2$ where $n \in \mathbb{Z}$, as shown in the right plot of (B.7). Anywhere away from these lines will be in the heating phases where Furstenberg's criteria are satisfied.

As an illustration, we consider a concrete example with type II fixed points in the phase diagram. For $H_0$, we choose $\sigma^+ = \sigma^- = 0$ and $\sigma^0 = 1$ in (5). For $H_1$, we choose $\sigma^0 = \sigma^- = 0$ and $\sigma^+ = 1$ in (5). In this case, the hyperbolic $SU(1,1)$ matrix $M_1$ in (21) has the expression with $\alpha = \cosh\left(\frac{\pi T}{l}\right)$ and $\beta = i \sinh\left(\frac{\pi T}{l}\right)$. We choose $H_0$ and $H_1$ randomly with probabilities $p_0 = p_1 = 1/2$. The distribution of $\lambda_L$ can be found in Fig. 27, where one can observe a line of exceptional points along $T_0/l_{0,\text{eff}} = 1/2$. Similar to the type I exceptional points, the type II exceptional points can be detected by $\lambda_L$ (or $\lambda_S$) that characterize the entanglement growth, but cannot be detected by $\lambda_E$ which characterize the total energy growth (See the definition in (13)), as shown in the right plot of Fig. 27.

Now let us take a further look at the scaling behavior of $\lambda_L$, which equals $\lambda_S$, near the type II exceptional points. As shown in Fig. 28, by fixing $T_1/l$ in Fig. 27, one can find that $\lambda_L \propto (T_0/l_{0,\text{eff}} - 1/2)^\alpha$ with the fitting parameter $\alpha = 0.2$. This value is close to the fitting parameter $\alpha = 0.19$ near the type I exceptional point in Fig. 6.

Furthermore, as seen in Fig. 28 (right plot), we also check the ensemble-averaged entanglement entropy growth at the type II exceptional point. One can observe that $\mathbb{E}(S_A(n))$ grows as $\sqrt{n}$ for large $n$, which has the same feature as the type I exceptional point. This is as expected since we have shown that type I and type II exceptional points can be mapped to each other (See Appendix A.3). More concretely, along the line $T_0/l_{0,\text{eff}} = 1/2$ in (B.7), one has $M_0 = \begin{pmatrix} i & 0 \\ 0 & -i \end{pmatrix}$, and $M_1 = \begin{pmatrix} \cosh\frac{\pi T_1}{l} & i \sinh\frac{\pi T_1}{l} \\ -i \sinh\frac{\pi T_1}{l} & \cosh\frac{\pi T_1}{l} \end{pmatrix}$. Now we denote the smallest subgroup generated by $\{M_0, M_1\}$ as $G_\mu$. Then $G_\mu$ is also generated by $\{M_A, M_B\}$, where $M_A := M_1 M_0$ and $M_B := M_0$. One can find that $M_A$ and $M_B$ are the same as (45) at type I exceptional points, by identifying $\varphi = \frac{\pi T_1}{l}$ and setting $\phi = 0$. But it is noted that the detailed features of entanglement entropy evolution at type I and type II exceptional points can be quantitatively different, although both grow as $\sqrt{n}$ when $n$ is large. This is straightforward to understand because a random driving with $M_0$ and $M_1$ with probabilities $p_0$ and $p_1$ at type II exceptional point is not exactly the same as the random driving with $M_A$ and $M_B$ with certain probabilities $p_A$ and $p_B$ at type I exceptional point.

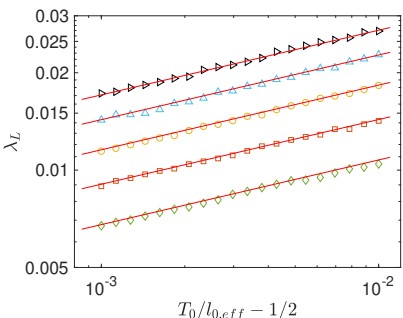 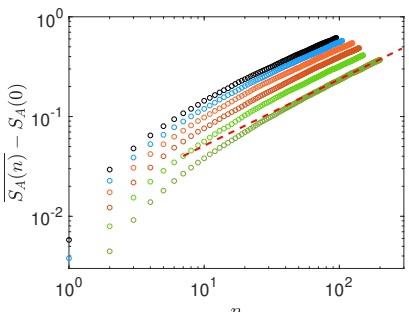

Figure 28: Left: Scaling of $\lambda_L$ near the type II exceptional point in Fig. 27. From top to bottom, we have $T_1/l = 0.09, 0.08, 0.07, 0.06$, and $0.05$ (see Fig. 27). The solid red lines are fittings with $y \propto x^{0.2}$. Right: The ensemble-averaged entanglement entropy at the type II exceptional point with $T_0/l_{0,\text{eff}} = 1/2$ and (from top to bottom) $T_1/l = 0.09, 0.08, 0.07, 0.06, 0.05$, and $0.04$. The red dashed line is a guiding line with $y \propto \sqrt{x}$.

## B.3 Group walking

In this appendix, we provide an intuitive way to understand the distribution of operator evolution in the randomly driven CFT.

As introduced in our prior work [39], the group walking in $\Pi_n = M_1 \cdot M_2 \cdots M_n$ determines the time evolution of entanglement/energy. More concretely, we consider another general form of a SU(1, 1) matrix as [26]

$$M(\rho, \zeta) = \frac{1}{N_\rho} \begin{pmatrix} \sqrt{\zeta} & -\rho^* \frac{1}{\sqrt{\zeta}} \\ -\rho \sqrt{\zeta} & \frac{1}{\sqrt{\zeta}} \end{pmatrix}, \quad \rho \in \mathbb{D}, \zeta \in \partial\mathbb{D}, \tag{B.8}$$

where $N_\rho = \sqrt{1 - |\rho|^2}$. Here we have defined the unit disk as $\mathbb{D} := \{z \in \mathbb{C}, |z| < 1\}$, and the edge of the disk as $\partial\mathbb{D} := \{z \in \mathbb{C}, |z| = 1\}$. For the group walking, we mean the evolution of the parameters $\rho$ and $\zeta$ (or their combinations) in $\Pi_n$. In this appendix, we are mainly interested in the group walking of $\rho$, which is related to Theorem 2.5 and the operator evolution.

In the property ($i$) of Theorem 2.5, it is stated that if Furstenberg's criteria are satisfied, then for any $z \in \mathbb{D}$, $M_1 M_2 \cdots M_n \cdot z$ converges to a certain $z^\bullet \in \partial\mathbb{D}$ as $n \to \infty$. This property actually tells us the behavior of group walking in SU(1, 1) matrix $\Pi_n$ in (8). By simply choosing $z = 0$, one has

$$\lim_{n \to \infty} M_1 M_2 \cdots M_n \cdot z = \lim_{n \to \infty} \Pi_n \cdot z = \lim_{n \to \infty} \frac{\beta_n}{\alpha_n^*} =: - \lim_{n \to \infty} \rho_n^* = z^\bullet \in \partial\mathbb{D}. \tag{B.9}$$

That is, in the long time driving limit, $\beta_n/\alpha_n^*$ will approach a certain stable value, which is independent of $n$.

A sample plot of the group walking of $\rho_n$ can be found in Fig. 29. In the heating phase where $\Delta T/l_{\text{eff}} \neq 0$, $\rho_n$ will approach a certain stable point at $\partial\mathbb{D}$.[27] For different random sequences, the locations of these stable points are different. As discussed near Theorem 2.5, the locations of these random stable fixed points correspond to the locations of operator evolution.

There are several interesting features we want to point out:

---

[26]More precisely, this parametrization ($\rho \in \mathbb{D}, \zeta \in \partial\mathbb{D}$) of matrix $\Pi$ only covers the SU(1, 1)/$\mathbb{Z}_2$, to obtain the full SU(1, 1) group, one need to let $\zeta$ live on the double cover of the boundary circle. However, our physical quantities are obtained from the Möbius transformation rather than the SU(1, 1) matrix directly, the former is indeed isomorphic to the $\mathbb{Z}_2$ quotient of the latter, namely SU(1, 1)/$\mathbb{Z}_2$ and agrees with our parametrization.

[27]For comparison, we plot the group walking with a finite number of driving steps. For those driving parameters with small $\lambda_L$, one needs to consider more driving steps to observe $\rho_n$ arriving at a certain fixed point at $\partial\mathbb{D}$,

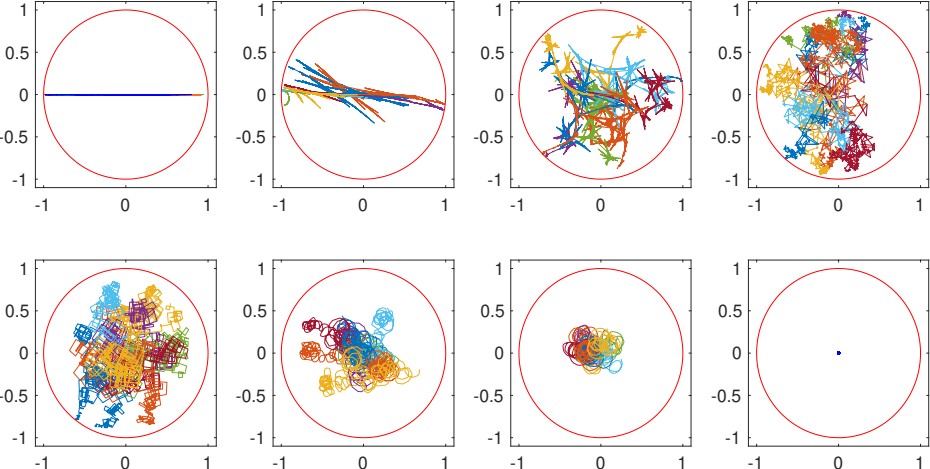

Figure 29: Trajectory of $\rho_n$ on the unit disk $\mathbb{D}$ with $n = 200$ steps of random driving. The driving protocol and parameters are the same as those in Fig. 7. We choose $\theta$ randomly distributed in $[0, 0.2]$. From left to right (and then top to bottom), we choose $\Delta T/l_{\text{eff}} := T_\theta/l_{\theta,\text{eff}} - 1/2 = 0$, $0.0005$, $0.01$, $0.1$, $0.25$, $0.4$, $0.48$, and $0.5$, respectively. The number of random samples are $N_{\text{sample}} = 10$.

1. At the type I exceptional point (where $\Delta T/l_{\text{eff}} = 0$), $\rho_n$ will walk to $\partial\mathbb{D}$ only at two points, which are $z = 1$ and $-1$ in Fig. 29. This is different from the heating phase where $\rho_n$ walks to $\partial\mathbb{D}$ randomly with a continuous distribution. The group walking feature at the exceptional point is related to the ensemble-averaged distribution of operator evolution $\nu_O = \frac{1}{2}\delta(z - e^{i\theta_0}) + \frac{1}{2}\delta(z - e^{i\theta_1})$ [See Eq.(58)]. For the parameter in Fig. 29, one has $e^{i\theta_0} = 1$ and $e^{i\theta_1} = -1$.

2. At the exceptional point (where $\Delta T/l_{\text{eff}} = 0$) and the trivial point (where $\Delta T/l_{\text{eff}} = 1/2$), although $\lambda_L = 0$ in both cases, the features of group walking are totally different. In the later case, there is no nontrivial group walking of $\rho_n$, i.e., $\rho_n$ stays at the origin.

3. For the group walking of $\rho_n$ near the exceptional point and near the trivial point, their features are different. For example, the trajectories of $\rho_n$ are approximately large arcs near the exceptional point; while they mainly circle around the origin near the trivial point. We believe these different features in group walking, which are intuitive, are related to the different scaling behaviors of $\lambda_L$ in Fig. 6.

As a remark, besides the group walking of $\rho_n$ which are related to the distribution of operator evolution, one can also consider the group walking of $\rho_n \zeta_n$ in (B.8). As discussed in [39], the group walking of $\rho_n \zeta_n$ are related to the distribution of the locations of energy-momentum density peaks.

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
