# Peer review of "Periodically, Quasi-periodically, and Randomly Driven Conformal Field Theories (II): Furstenberg's Theorem and Exceptions to Heating Phases"

_SciPost Physics, doi:SciPost Phys. 13, 082 (2022)_

## Round 1 · Referee Report · Anonymous (Referee 1) · 2022-2-3

Strengths

The paper is clearly written.
The work is mathematically elegant.

Weaknesses

It's unlikely that the models studied here will be experimentally accessible in the near future.
Randomly driven 1+1D CFTs are a rather specialized/finely-tuned set of models; it seems unlikely that the lessons learned here are applicable to many-body systems more generally.

Report

This paper examines the effect of random driving in CFTs in 1+1D. For a special set of driving perturbations, the dynamics of energy and entropy can be understood in terms of the properties of random concatenations of Mobius maps.

The authors find that, generically, quenching from the ground state results in a linear growth of entanglement in time and an exponential growth of energy density. The authors are able to obtain the growth rates of these various quantities analytically, often in terms of the Lyapunov exponent. The authors also provide a complete account of various (finely tuned) subclasses of model which can exhibit more exotic phenomena (like a $\sqrt{t}$ growth of von Neumann entropy).

This paper forges a link between the well established field of CFTs and the more recent interest in equilibration in driven quantum many-body systems. The paper is well-written (especially given its length and complexity). So, provided the following points and questions are addressed, I am happy to recommend publication.

  1. For those models with linear entropy growth, it's clear (e.g., fig 7) that the models take some time to settle into a linear growth regime. e.g., in Fig 7 the curves are not exactly straight lines (some wiggle/some possibly have curvature). What parameters determine the time scale over which these transient effects die out? Is it some measure of proximity to an exceptional point?

  2. In 1+1 D, RCFTs have an infinite tower of conserved densities (e.g., https://arxiv.org/pdf/hep-th/9412229.pdf). Do the randomly driven CFTs examined here have anything like the same structure? i.e., does the model have any (local) conserved quantities? I suspect the answer is no, but I'm not entirely sure.

  3. Related to 2. The regimes with $\sqrt{t}$ growth of entropy are intriguing. Some ergodic systems with diffusive transport have $\sqrt{t}$ growth of Renyi entropies (e.g., https://journals.aps.org/prl/pdf/10.1103/PhysRevLett.122.250602). I wonder if there is any connection to that body of work. One way of checking this is to answer Q2: There's unlikely to be a connection if you can prove that the CFT models (in particular, the relevant exceptional points with $\sqrt{t}$ entropy growth) do not have diffusive transport.

  4. Also related to 2. Do the authors agree that the cardy-calabrese quasi-particle picture is underpinned by the inherent integrability of 1D CFTs (which guarantees existence of quasi-particles)? If so I'm confused: The argument for $\sqrt{t}$ entropy growth at certain exceptional points seems to rely on the quasi-particle picture, yet it doesn't appear as though the CFT models studied are integrable (Q2)?

  5. Please make listed requested changes.

Requested changes

  1. Please respond to questions above, and if appropriate, change main text.
  2. The notation in Eq 7 is confusing. The $\cdot z$ makes it look like z is a row vector, which it isn't. I recommend defining the matrix M on a new line.
  3. Pg 3 Hamilton -> Hamiltonian
  4. pg 5 ensamble->ensemble
  5. pg 13 momenum -> momentum

---

## Round 1 · Referee Report · Bastien Lapierre (Referee 2) · 2022-5-5

Strengths

  • Interesting link between non-equilibrium dynamics and the theory of SL(2) random matrices: sublinear growth of entanglement entropy when Furstenberg criteria are violated, as well as different spatial structures in the operator evolution and energy density.

  • Great agreement with the free fermion numerics for a decent number of driving steps.

  • Complete classification of the SL(2) random drives.

  • Enlightening quasi-particle picture in Sec. 2.4.4.

Weaknesses

  • Results quite limited in their nature: heating takes over the whole "phase" diagram, and only finely-tuned values of the driving parameters can escape this fate in their entanglement properties (but they still heat-up strictly speaking, as the energy grows exponentially, see Fig. 4 b.).

  • Some technical details might confuse readers who are not already familiar with this topic. The information could certainly have been conveyed in a more concise way.

Report

In this paper the authors study the dynamics of a (1+1)d CFT subject to a random step-drive. The Hamiltonian is piecewise constant at each segment of the drive and is given by a SL(2) deformation of the usual uniform CFT Hamiltonian. It is known that the time evolution under such SL(2) deformed Hamiltonians is encoded in Möbius transformations for the (quasi-)primary fields, enabling to find the time evolution of various physical observables. In the case of a periodic drive, it has been shown in previous works that the dynamics of the problem is fully governed by the classification theory of such Möbius transformation, leading to non-heating / heating phases for elliptic / hyperbolic Möbius transformations, and the phase transition between them given by the parabolic Möbius transformation. The authors investigate the case of a random drive, under which the system is always in a heating phase. They find that although the energy generically increases exponentially, the entanglement entropy has a square root growth (instead of linear) for finely-tuned values of the driving parameters. The authors classify such particular values of the driving parameters, termed "exceptional points", using the theory of products of random SL(2) matrices (and in particular, Furstenberg theorem), as all the dynamics is encoded in such products.

The paper is clearly written, and does provide an exhaustive classification of SL(2) randomly driven CFTs. It is a step forward in the inhomogeneous Floquet CFT literature, where only deterministic (periodic and quasi-periodic) SL(2) driving sequences have already been investigated in details. While the classification is impressive, it is however important to note that the results show that the rich non-heating versus heating phase structure of the periodic drive is now absent in the random drive, and the main results are very finely-tuned as they only concern isolated points/lines in parameter space, instead of an extended region of non-zero measure. The general classification is mathematically elegant and provides a bridge between non-equilibrium quantum many-body physics and the theory of random matrices. Provided that the authors address my comments, I would gladly recommend publication in Scipost Physics.

  1. Although only SL(2) deformations are considered in this work, would you expect for general smooth deformations of the stress tensor (i.e., belonging to the group of diffeos of the circle) such exceptional points to survive?

  2. The comparison between lattice numerics and CFT is rather impressive at the exceptional point. However in Fig. 17, the value of $\theta$ is small, meaning that the driving Hamiltonian $H_{\theta}$ is a very small perturbation of $H_0$. What happens for the numerical agreement if $\theta$ is large, i.e., closer to the SSD limit? Is the agreement still as good for as many driving steps?

  3. What would be the effect of an initial thermal state at temperature $\beta^{-1}$ in the dynamics? Would one expect that for small enough initial temperatures the exceptional point would survive? Could you provide a finite-temperature lattice simulation to compare with your zero-temperature result of Fig.17? Is the quasi-particle picture presented in Sec. 2.4.4 still valid in this case?

  4. Could you provide a physical intuition for the different scaling behaviours of the Lyapunov exponent between the exceptional and the trivial points? Is the scaling near an accidental point also different from the scaling near an exceptional point? Furthermore, in the case of SL(2) periodic drives, the scaling exponent of the Lyapunov exponent when going from the heating phase to the non-heating phase is $\frac{1}{2}$ [arXiv:1805.00031]. Could you comment on the difference with the 0.2/0.19 exponent of the random case?

  5. The heating phase of periodically driven CFTs is strongly non-ergodic in the sense that the operator evolution converges to some fixed points, leading to extremely inhomogeneous growth of energy density or entanglement. Except at type I/II exceptional points where this is still the case, in the heating phase of the random drive the distribution of operators is more uniform, see the sketch at the bottom of page 9. Would you argue that in this case the heating is ergodic? Thus type I/II exceptional points would be some measure zero sets of parameter space that retain non-ergodicity even in the random setting.

  6. The SL(2) periodic driving case amounts to a quench with an effective Hamiltonian at stroboscopic times, whose SL(2) Casimir invariant determines the heating/non-heating phase. Can one also replace the dynamics of the random drive by a quench with an effective Hamiltonian? It seems that because the emergent spatial structure is washed-out by the randomness of the drive, such an effective quench description in terms of a spatially deformed CFT does not exist anymore.

Requested changes

Please respond to the questions above, and if appropriate, change the manuscript.

---

## Round 2 · Referee Report · Bastien Lapierre (Referee 2) · 2022-8-2

Report

The authors have implemented the requested changes and discussed in great details the different questions, therefore I think the paper is ready for publication in Scipost Physics.

---

## Round 2 · Author Response

Since there are figures in our reply, we attach a pdf file below.

---

## Round 2 · List of Changes

-- We added appendix B.1.2 discussing the early time evolution of the entanglement entropy in the heating phase in the updated draft.

-- We update Fig.23 and the corresponding descriptions in appendix A.5 by including a sample plot of scaling behavior of Lyapunov exponents near the accidental exceptional points.

-- We add a footnote to discuss the possibility of the existence of exceptional points in randomly driven CFTs with general deformations. It is added in footnote 22 in Sec.4 in the part Random drivings beyond SL2 deformations" (Page 45 and 46).

---

## Editorial Decision

published